# LLMs as Rules Oracles: Exploring Real-World Multimodal Reasoning in Tabletop Strategy Game Environments

**Joseph J. Peper**[1*]     **Sai Krishna Gandra**[1]     **Yunxiang Zhang**[1]

**Vaibhav Chennareddy**[1]    **Shloki Jha**[1]    **Ali Payani**[2]    **Lu Wang**[1]

[1]University of Michigan      [2]Cisco Research

## Abstract

We introduce **LudoBench**[1], a multimodal reasoning benchmark that evaluates whether vision-enabled large language models (LMs) can acquire, integrate, and reason over heterogeneous game knowledge in mainstream analog tabletop games. Unlike prior works that emphasize deep strategic mastery, LudoBench targets an initial reasoning challenge uninitiated gamers face: *correctly comprehending a new tabletop strategy game for the first time*. We examine whether, given a visual depiction of a tabletop scene and a corresponding ruleset, a model can correctly answer grounded questions about the pictured scenario. Concretely, LudoBench tests three cumulative situated game-comprehension capabilities: (1) *Environment Perception*, (2) *Heterogeneous Rules Integration*, and (3) *Short-horizon Optimization*, to progressively stress-test the foundational reasoning required for real-world game comprehension. Evaluating frontier LMs on five diverse strategy games, we find that even the strongest models achieve only ∼76% accuracy on simple environment perception tasks and fall below 13% on situated multi-step comprehension puzzles that hobbyist gamers can routinely solve. Our extensive failure analysis and knowledge-ablation experiments reveal that *models largely fail to comprehend rich cross-modal reference knowledge* and are subsequently unable to apply this knowledge to messy and unfamiliar situated environments. Our findings highlight the many steps remaining for current methods to succeed on complex multimodal reasoning in the real world. Benchmark, leaderboard, and visualizer are available at 🤗 https://huggingface.co/spaces/launch/LudoBench .

## 1 Introduction

Game reasoning is a prevalent area of AI research, spanning applications from strategic gameplay to human–AI interaction (Silver et al., 2018; Lin et al., 2024; Costarelli et al., 2024). Frontier LMs such as o1 (OpenAI, 2024c) and Gemini Pro (Google, 2025) perform well in structured, simulator- and rules-enforced digital environments, including games such as chess and Go (Zhang et al., 2025; Guo et al., 2024); however, much of this progress utilizes rules-enforced or gym-style infrastructure (Towers et al., 2024; Brockman et al., 2016) that exposes a clean environment, disallows illegal actions, and handles state progression. When this scaffold is removed, even strong models propose invalid or nonsensical moves even for well-studied games such as chess (Hwang et al., 2025), suggesting incomplete internalization of game state and rules.

With this in mind, we pose the question: ***Can LMs serve as rules oracles in complex, real-world, gaming environments?*** This question is especially pertinent to the domain of *tabletop strategy games*, which are produced by the thousands each year and played primarily in offline settings (Naghikhani, 2024), with long, multimodal, and idiosyncratic rulebooks that notoriously

---

*Correspondence to jpeper@umich.edu.

[1]*Ludo*—from Latin *lūdō*, "I play", and the root of *ludology*, the scholarly study of games.

impose substantial cognitive load on first-time players (Heron et al., 2018). For example, given only an image of a tabletop game scene and a corresponding external ruleset, *can a model acquire dense cross-modal reference knowledge*, retrieve and disambiguate the relevant clauses, parse iconography and diagrams, ground them in a cluttered spatial layout, *and apply them correctly, without any legality checker or interactive affordances*?

To explore this challenge, we introduce LUDOBENCH, a multimodal game comprehension benchmark explicitly designed to evaluate vision-capable LMs on *situated game comprehension* in conventional tabletop strategy games. Unlike traditional benchmarks centered on long-term optimization, LUDOBENCH emphasizes the ability to *learn and apply new rules*, *interpret dense, real-world game states*, and *make optimal tactical decisions*, mirroring the foundational tasks faced in the first-time player setting. Specifically, the benchmark covers three cumulative competencies: (1) *Visual Environment Perception*, which involves identifying and interpreting key objects, components, layouts, and spatial relations within a physical tabletop game, (2) *Heterogeneous Ruleset Integration*, which requires retrieving, disambiguating, and applying information from complex, multimodal rulebooks to the current game state, and, to test comprehension, (3) *Short-Horizon Optimization Puzzles*, which challenge models to, for example, find the action sequence that maximizes the player's end-of-turn score.

To summarize, our main contributions and findings are as follows:

- **We hand-curate and release LUDOBENCH**, a multimodal game comprehension benchmark of 638 QA examples from five mainstream tabletop games that are varied in rule complexity and reasoning challenges, with each example pairing a visual game state and the game ruleset with a question.

- **State-of-the-art LMs significantly under-perform humans on complex situated game comprehension:** Testing on nine frontier LMs, we find that average model performance is only ∼63% on simple perception tasks (Tier 1), ∼36% on game rules reasoning (Tier 2), and under 10% on optimization puzzles (Tier 3). In contrast, *even Tier 3 puzzles are very solvable (∼82% accuracy) for hobbyist players.*

- **LUDOBENCH surfaces diverse multimodal reasoning failures:** Despite having full access to both the visual game state and rulebook, models frequently fail at scene parsing, rule retrieval, spatial manipulation, and temporal state tracking. Interestingly, although illustrated image-based rulebooks are expected to offer richer guidance, mirroring how humans learn from demonstrations, models often perform worse with these visual inputs than with plain-text rules, frequently overfitting to examples and failing to generalize. This reveals core limitations in symbolic grounding and visual-contextual reasoning. We conduct an extensive breakdown of these with annotated examples and case studies (Fig. 6, Sec. 5, Appx. D, E, and F).

## 2 RELATED WORK

**Game Reasoning with Large Language Models.** Games have long been a central testbed for AI progress, especially in planning and decision-making. Reinforcement-learning systems such as AlphaGo and AlphaZero have demonstrated superhuman ability in perfect-information settings through extensive self-play and domain-specific optimization (Silver et al., 2016; 2018). Recent advances in LMs have enabled broad generalization across unfamiliar language, vision, and interaction tasks, often in zero-shot settings without fine-tuning (Brown et al., 2020; OpenAI, 2023; Wang et al., 2023). The shift expands the scope of game reasoning: from executing within tightly scoped, rules-enforced environments to acquiring and applying new knowledge in unfamiliar settings. Building on this perspective, we examine how foundation models adapt to scenarios requiring rapid knowledge acquisition and environment comprehension.

**Game Reasoning Benchmarking.** Prior works have studied how LMs reason in text-based or programmatic (symbolic) game environments where rules and states are explicitly serialized (Hu et al., 2025a;b; Lin et al., 2024; Costarelli et al., 2024). These settings evaluate whether models can reason over explicit rule and state representations (e.g., "You are playing tic-tac-toe. The board has X at (1,1) and O at (2,2)..."). In contrast, many real-world tabletop games present more vibrant,

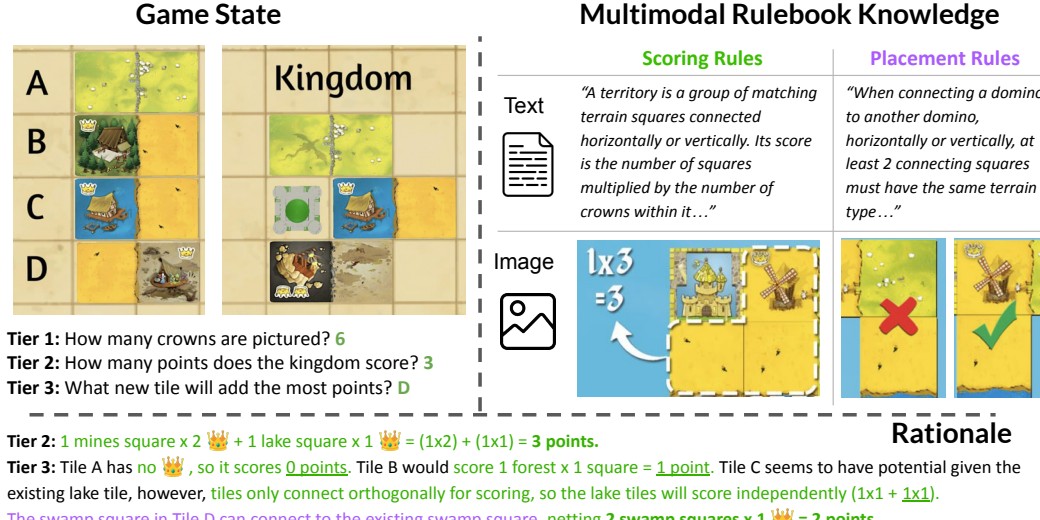

Figure 1: We illustrate examples (simplified, for presentation purposes) of questions from all three LUDOBENCH tiers. Tier 1 (*Perception*) measures the ability to extract basic visual information from the game state; Tier 2 (*Knowledge Integration*) assesses how well the model applies game rules to situated questions; Tier 3 (*Optimization*) assesses the model's capacity to solve tactical puzzles akin to those frequently made during gameplay. Due to the situated nature of our task, models must integrate heterogenous knowledge across multimodal rulebooks and game states. **Note:** Figure 6 provides extended gameplay examples and analyses of common challenges posed by LUDOBENCH.

cluttered scenes (e.g., an image of a game with dozens or hundreds of diverse units and components on a war-game map), making serialization nontrivial. Capturing these real-world nuances and the limitations they impose on current models is a core aim of LUDOBENCH.

Beyond text- and symbolic-comprehension settings, other work evaluates visual understanding by using board images or schematic diagrams for common games such as Chess, Connect Four, and Tic-Tac-Toe (Saha et al., 2024; Wang et al., 2025). These studies test whether models can align visual inputs with rule-consistent logic, but they still typically operate on simple games with highly constrained layouts and minimal iconography. In contrast, LUDOBENCH targets visually rich, real-world scenes paired with full rulebooks, requiring models to parse heterogeneous components, retrieve and disambiguate relevant rules, and make grounded decisions in unfamiliar environments.

Another related line of work probes core reasoning competencies via synthetic games with evolving rulesets: ARC tests visual abstraction over grid puzzles, while Baba Is AI, PuzzleVQA, and Algo-PuzzleVQA extend this to dynamic logic and fine-grained puzzle solving (Prize, 2025; Cloos et al., 2025; Chia et al., 2024; Ghosal et al., 2024). These tasks isolate foundational reasoning competencies; however, they are often toy-like and not necessarily indicative of performance on real-world games. A complementary thread investigates long-context decision-making in interactive environments (e.g., TextQuests and Crafter) (Phan et al., 2025; Hafner, 2022), often within rule-enforcing simulators or with memory/tracking scaffolds. More broadly, many evaluations rely on enforced structure, cleaned inputs, or oracle feedback (Costarelli et al., 2024; Lin et al., 2024; Topsakal et al., 2024; Balla et al., 2024), limiting applicability to analog tabletop settings. LUDOBENCH pairs intricate, real-world visual game states with often lengthy, heterogeneous rulebooks and situated questions, *unifying these varied directions into a single, real-world setting* and advancing a tangible application for vision-enabled LMs: acting as multimodal rules oracles for offline play.

---

[2] https://www.boardgamegeek.com

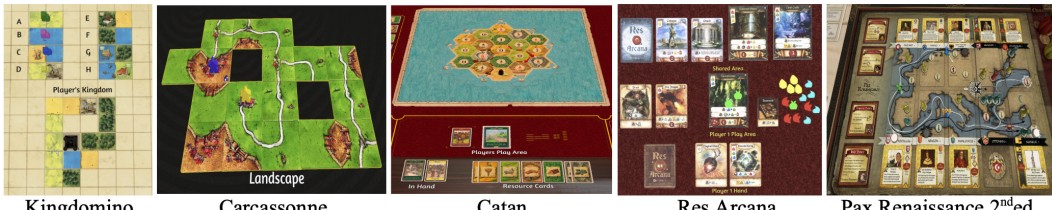

| Kingdomino | Carcassonne | Catan | Res Arcana | Pax Renaissance 2[nd]ed. |

Figure 2: Example game states from each game. Our benchmark requires models to perceive visual game states in sandbox environments, integrate rule knowledge, and perform situated reasoning.

| Game | ‖Rulebook‖ | Diff. | Unique Game Properties | # Rules | # Figs. |
|---|---|---|---|---|---|
| *Kingdomino* | 4 pg. | 1.2 | tile-laying, spatial scoring, individual player areas | 35 | 6 |
| *Carcassonne* | 8 pg. | 1.9 | shared tile-laying, dynamic board topology, position-coded roles | 39 | 30 |
| *Catan* | 16 pg. | 2.3 | network building, connectivity constraints, action chaining | 44 | 19 |
| *Res Arcana* | 12 pg. | 2.6 | card-based interactions, heavy symbol usage, card orientation, action sequencing | 112 | 31 |
| *Pax Ren. (2e)* | 44 pg. | 4.6 | shared map, private cards/tableau, large number of components, intricate ruleset | 247 | 58 |

Table 1: Descriptions of the five benchmarked games, which are diverse in ruleset length and difficulty. In addition to game difficulty, we also roughly quantify the number of atomic rule clauses and supporting rulebook figures and illustrations. Difficulty is a community-calculated[2] game complexity metric, ranging from (1 = easy, 5 = hard).

## 3 LUDOBENCH DETAILS

### 3.1 TASK FORMULATION

In this work, we explore the domain of tabletop strategy games, with an emphasis on **situated visual-language reasoning** that faithfully reflects the experience of real-world gamers during first-time gameplay.

We formulate the task as a visual comprehension task with the following inputs: **(1)** A game state $G$ depicted by one or more images, **(2)** The game's corresponding ruleset $R$ (often containing a mix of text, diagrams, iconography, and worked examples), and **(3)** A natural-language query $Q$ to probe the provided knowledge. An LM must then output an answer $A$ which resolves $Q$ given $(G, R)$.

We define three game comprehension tiers of interest, emphasizing the cumulative levels of comprehension needed to solve game reasoning tasks:

**Tier 1 – Environment Perception.** These examples probe basic perception of $G$, such as item counts, distinguishing colors, identifying spatial relations, *without requiring any rule knowledge*; accurate scene parsing skills and generic gaming familiarity are sufficient for success. The ruleset $(R)$ is not needed for solving these problems.

**Tier 2 – Heterogeneous Rules Integration.** The model must ground one or more rule clauses from $R$ within the observed visual state to assess game-specific information such as game state validity, current player score, or the valid actions a player may take.

**Tier 3 – Short-Horizon Optimization.** Beyond game state assessment and isolated comprehension, in this tier the model must suggest or predict the optimal outcome from a given game state and provided constraint expressed in $Q$. Our emphasis is on short-term optimization – such as calculating an optimal action sequence given tractable, short-horizon constraints (e.g., "what is the maximum points the player can score by the end of the current round"). Solving such problems

necessitates *comprehensive* game understanding and requires models to formulate an internal world model capable of lightweight *forward simulation*.

## 3.2 BENCHMARK DETAILS

We hand-annotate **638** QA examples in the described task format across five mainstream tabletop games—*Kingdomino*[3], *Carcassonne*[4], *Catan*[5], *Res Arcana*[6], and *Pax Renaissance (2e)*[7]—selected for diversity in component density, ruleset length, and mechanical focus (Table 1). In total, we have 100+ examples from each game and 200+ examples in each reasoning tier. All scenarios are staged in Tabletop Simulator[8], an interactive sandbox environment which affords free camera control and precise piece manipulation, enabling reproducible configurations and high-quality captures of visual game states (Appx. I).

As the first in-depth study of situated multi-modal game comprehension that pairs complex, multimodal rulesets with intricate visual tabletop game states, we prioritized fidelity and correctness through hand-crafted examples. Through this process, we find that our situated visual states are effectively reusable scaffolds for expansion; likewise, we find that rulebook worked examples can be synthesized to quickly brainstorm meaningful new questions, topics, and scenario variants. We believe lightweight human-in-the-loop curation approaches offer a strong balance of quality and scale for this domain.

| Game | Tier 1 | Tier 2 | Tier 3 |
|---|---|---|---|
| *Kingdomino* | 40 | 40 | 50 |
| *Carcassonne* | 40 | 40 | 50 |
| *Catan* | 40 | 40 | 31 |
| *Res Arcana* | 40 | 40 | 50 |
| *Pax Ren.* | 40 | 40 | 57 |

Table 2: LUDOBENCH comprises 638 questions across five games (Tier 1: 200; Tier 2: 200; Tier 3: 238).

**Example-Selection Heuristics.** We use the following heuristics to ensure high-quality examples that are both easy to produce, and have a verifiable ground-truth solution: **(1)** *Deterministic, fully observable game states.* The correct answer is derivable from what is visible in the provided game state images; hidden information and chance events are excluded. This ensures that reasoning is grounded in visible evidence and that answers are reproducible by both humans and models. **(2)** *Scoped opponent modeling.* For multi-turn optimization tasks (Tier 3), we either explicitly assume no opponent interaction, or, clearly specify the expected opponent behavior: (e.g., "on their turns, assume opponents will act to minimize your score"); this avoids ambiguity in outcomes, and allows us to focus instead on assessing rules integration. **(3)** *Single, well-defined solution.* Each example is crafted so that proper reasoning yields exactly one answer; ambiguous or probabilistic cases are avoided. This guarantees a clear evaluation target and simplifies correctness verification during evaluation.

**Annotation Process.** We conduct a three-stage annotation process to ensure our examples are high-quality. First, we task a group of two gaming experts to produce and label QA examples using the Tabletop Simulator platform. Second, we train and onboard three student annotators with hobbyist-level gaming experience to independently answers the questions (with the full rulebook available for reference). For every expert-hobbyist disagreement, we then manually discuss and resolve all conflicts to produce the final ground-truth examples. We define annotator correctness as the proportion of examples where the original answer provided by either the expert or the hobbyist exactly matches the final resolved label in the benchmark dataset. We find that the Phase 1 *expert* annotation yielded 98.3%, 97.4%, and 87.2% correctness, respectively, for the examples from Tiers 1, 2, and 3 (i.e., this percentage of examples were not relabeled in subsequent rounds). Hobbyist performance accuracy (Phase 2) was 96.7%, 89.1%, and 82.2% for Tiers 1, 2, and 3. Full details,

---

[3]https://boardgamegeek.com/boardgame/204583/kingdomino
[4]https://boardgamegeek.com/boardgame/822/carcassonne
[5]https://boardgamegeek.com/boardgame/13/catan
[6]https://boardgamegeek.com/boardgame/259925/res-arcana
[7]https://boardgamegeek.com/boardgame/378594/pax-renaissance-2nd-edition
[8]https://www.tabletopsimulator.com

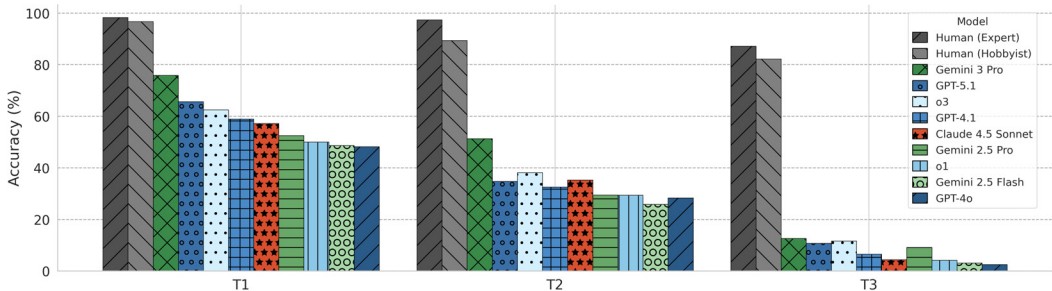

Figure 3: Overall tier-by-tier game reasoning performance, averaged over all combinations of games and rulebook modalities. Unlike humans, models struggle even with simple game state perception tasks (Tier 1), with further degradation when integrating rules knowledge into the questions (Tier 2), and single-digit accuracy on multi-step comprehension puzzles (Tier 3).

including the annotator onboarding process and a disagreement analysis case study can be found in Appendix J.

## 4 BENCHMARK EVALUATION

### 4.1 EVALUATION SETUP

We evaluate nine widely-used, multimodal-capable language models: OpenAI's **4o** and its extended-context sibling **4.1**; the reasoning-optimized variants **o1**, **o3**, and **5.1**; Google DeepMind's **Gemini Pro 3**, **Gemini 2.5 Pro**, and lower-latency **Gemini 2.5 Flash**; and Anthropic's hybrid reasoning model, **Claude 4.5 Sonnet**. Full details on prompt construction, model configurations, and the evaluation setup are provided in Appendix C.

**Prompt construction.** For caching efficiency, the rulebook knowledge $R$ is prepended at the beginning of the input, followed by the game state image(s) $G$, then the natural-language query $Q$:

```
[Start Rules]
<(optional) rulebook knowledge>
[End Rules]
<game-state image(s)>
<question + format specification>
```

All experiments are run with the *OpenCompass* VLMEVALKIT (Contributors, 2025), which unifies data parsing, batching, logging and answer extraction across models. Hyper-parameters are kept at their model defaults except as noted here; we 1) raise max_tokens to each model's maximum, 2) impose a 300 s timeout with up to five automatic retries for API-served models, and 3) omit the IMAGE-rulebook results for *Claude 4.5 Sonnet* (Anthropic, 2025) on *Pax Renaissance 2e and Catan*, whose rulebook contents exceeds its token context window.

**Evaluation Metrics:** We report the overall exact-match QA accuracy. By design, each example question specifies a lightweight semi-structured or (for longer answers) JSON-style answer schema; model outputs are first normalized by a post-processing LLM step (GPT-4o) that maps free-form text into this schema, after which exact-match is computed against the structured ground truth. To ensure reliable automated parsing during evaluation, we manually verified this LM processing step, randomly sampled 100 model predictions. We find the processing step maps outputs to the expected structure with perfect accuracy.

**Rulebook Format:** For each game we consider three variants of the ruleset knowledge $R$:

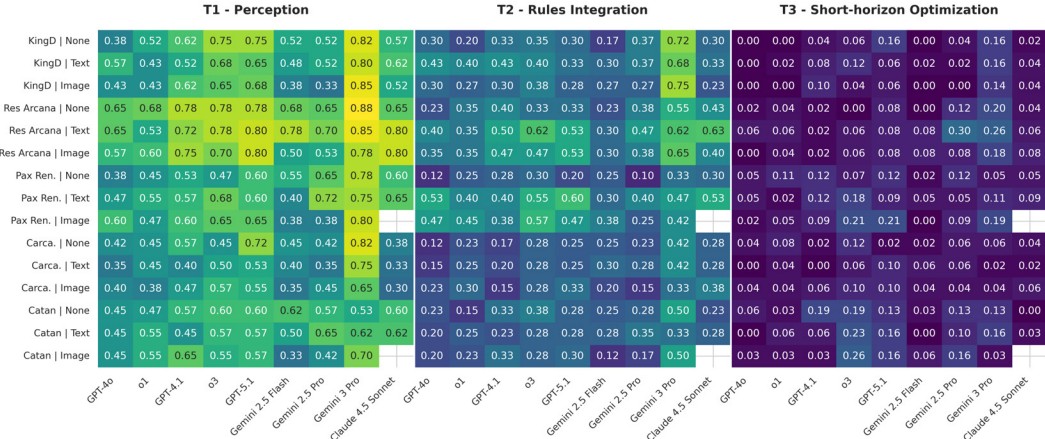

Figure 4: Accuracy breakdown. We display results of nine models on Tier 1–Tier 3 tasks under three rulebook modalities (None, Image, Text) for each game (**KingD**omino, **Res Arcana**, **Pax Ren**aissance 2e, **Carca**ssonne, **Catan**). All models achieve their highest accuracy on Tier 1 (Perception) and degrade sharply by Tier 3 (Optimization), confirming escalating and compounding task complexity. Note: the Pax Ren. and Catan image-modality rulebook did not fit within the context window of Claude 4.5 Sonnet thus we do not report these results.

- IMAGE: High-resolution screenshots of the rulebook PDF are supplied in the prompt; all pages are human-legible. This full-fidelity representation captures the same rule information provided to our annotators.

- TEXT: The text-only subset of the rulebook PDF is included in the prompt.

- NONE: Only the game state $G$ is provided; models can only use parametric knowledge.

## 4.2 OVERALL PERFORMANCE TRENDS

Figure 3 reports tier-by-tier accuracy for every model, averaged over all games and rulebook conditions. First, we observe that **even simple visual game perception tasks are challenging for frontier LMs**, with average 63% performance on the Tier 1 task. Across *all systems*, accuracy then degrades sharply, falling to 36% for Tier 2 (rules QA) and then to 8% for Tier 3 (Optimization). Of the models evaluated, we observe that Gemini Pro 3 is consistently the strongest, especially on Tier 1 and Tier 2 reasoning. GPT-5.1 and o3 are also performant. We observe a clear middle cluster – Claude 4.5 Sonnet, GPT-4.1, 4o, o1, Gemini 2.5 Pro and Gemini Flash 2.5 – with Flash 2.5 the weakest of the API-based frontier LMs. These results collectively highlight that even state-of-the-art multimodal LMs struggle to generalize across grounded reasoning tasks, indicating persistent limitations in visual parsing, rule grounding, and strategic planning.

**Tier-specific Observations.** Figure 4 provides a detailed performance across all splits. This analysis yields further takeaways, showing that:

(1) **Environment perception (T1) is not yet solved** – although some splits perform quite well (Gemini Pro 3 averages 83.6% performance on Res Arcana Tier 1 examples), Claude 4.5 Sonnet Tier 1 performance is as low as 30% *(Carcassonne | Img)*. We see frequent *failures in small-object detection and counting under partial occlusion/clutter*, yielding missed or miscounted gaming components. Models also exhibit *orientation-dependent text/icon recognition and brittle layout-aware spatial-relation parsing*, for example, when pieces sit on cards or are rotated.

(2) **Rules-comprehension questions (T2) are nearly twice as challenging as simple perception**, having the highest accuracy of 75% by GPT-o3 (King Domino | Image); while models often retrieve relevant rules correctly, we find they often misapply them due to flawed or shallow game state comprehension, often suggesting invalid moves or making critical rule misinterpretations leading to faulty score calculations.

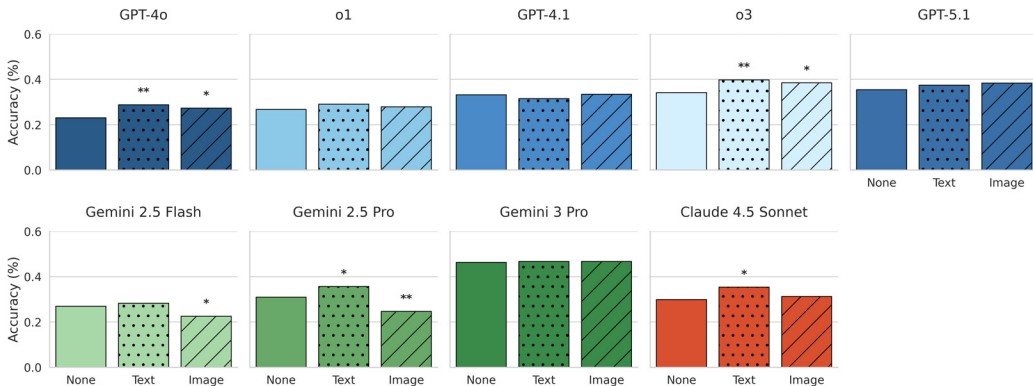

Figure 5: Rulebook modality analysis: overall accuracy for each model by rulebook modality. GPT and Claude families improve over the no-rulebook baseline with both image and text rulebooks; Gemini (except pro 3) improves with text but *degrades* with image. (Asterisks denote stat. sig. improvement vs. None; two-proportion z-test, two-sided; * $p < .05$, ** $p < .005$).

(3) **Short-horizon optimization (T3) remains virtually unsolved**, with only Gemini 3 Pro exceeding 20% performance on any split *(Res Arcana | Text)*. Figure 9 also shows that responses grow substantially longer in Tier 3, indicating *verbosity without commensurate reasoning gains*. We see a systematic limitation in models' ability to comprehend and then (validly) internally simulate even one or two steps within our complex, fully-observable gaming environments.

## 5 ANALYSES

### 5.1 INFLUENCE OF RULEBOOK MODALITY

Figure 5 aggregates model accuracy over games and tiers to isolate rule modality effects. Gemini Pro 3 performs strongly across modalities, reflecting both solid parametric knowledge and gains from text and image rulebooks. *Gemini Pro 2.5 shows a counterintuitive pattern* – decreasing with image rulebooks yet improving with textual rulesets. *All* models benefit from textual knowledge, with Gemini Pro 2.5, o3 gaining nearly 7%. By contrast, image-modality rulebooks have mixed effects: GPT models benefit, whereas both Gemini models (except Pro 3) see a decrease. These trends for GPT-4.1 and Gemini Pro 2.5 were consistent across numerous examples during manual analysis; see Appendix E for details. In contrast to humans – who greatly benefit from visual structure – current models still struggle to extract critical rule semantics from images, underscoring a need for stronger visual-document understanding.

### 5.2 BENCHMARK ORACLE SOLVABILITY

We assess benchmark head-room via plurality voting across all 27 (9 models × 3 modalities) outputs (Fig. 7). Tier 1: >90% of questions are solved by at least one system and 70% by a plurality, confirming task clarity. Tier 2: although most items remain solvable in aggregate (indicating models are capable of *some* knowledge integration), plurality accuracy slips to 50%, showing that robust rule grounding is still fragile even when the knowledge is present. Tier 3: solvability greatly decreases: 60% of Kingdomino questions and 50% overall remain unsolved; even the best plurality score (11% on Pax Ren.) leaves a large gap. For Kingdomino, we explore this further (Fig. 6, Appx. D) and see the combinatorial nature of Kingdomino (a) tile placement and (b) planning poses a challenge for all models.

### 5.3 RULE RETRIEVAL ERROR ANALYSIS

To probe where game reasoning breaks down, we analyse 30 Pax Renaissance Tier-2 questions under three input conditions—no rulebook, a text rulebook, and an image rulebook. For each, we annotate whether GPT-4o retrieves the relevant rule(s) and applies them correctly (Fig. 8). We see rule

retrieval climbs with richer input (20% → 73% → 90%), yet application accuracy lags behind (17% → 64% → 56%). In many stalled cases GPT-4o knows the rule but misreads the board (Appendix B.1) —an error images sometimes fix when the question hinges on visual cues (Appendix B.2). This reveals a dual bottleneck: richer input almost eliminates grounding errors, but faithful rule execution remains elusive. Finally, the model often over-fits to surface rulebook patterns, applying them rigidly and inappropriately, particularly on Tier-2 and Tier-3 tasks; Appendix F catalogues these hallucinations and their inductive triggers.

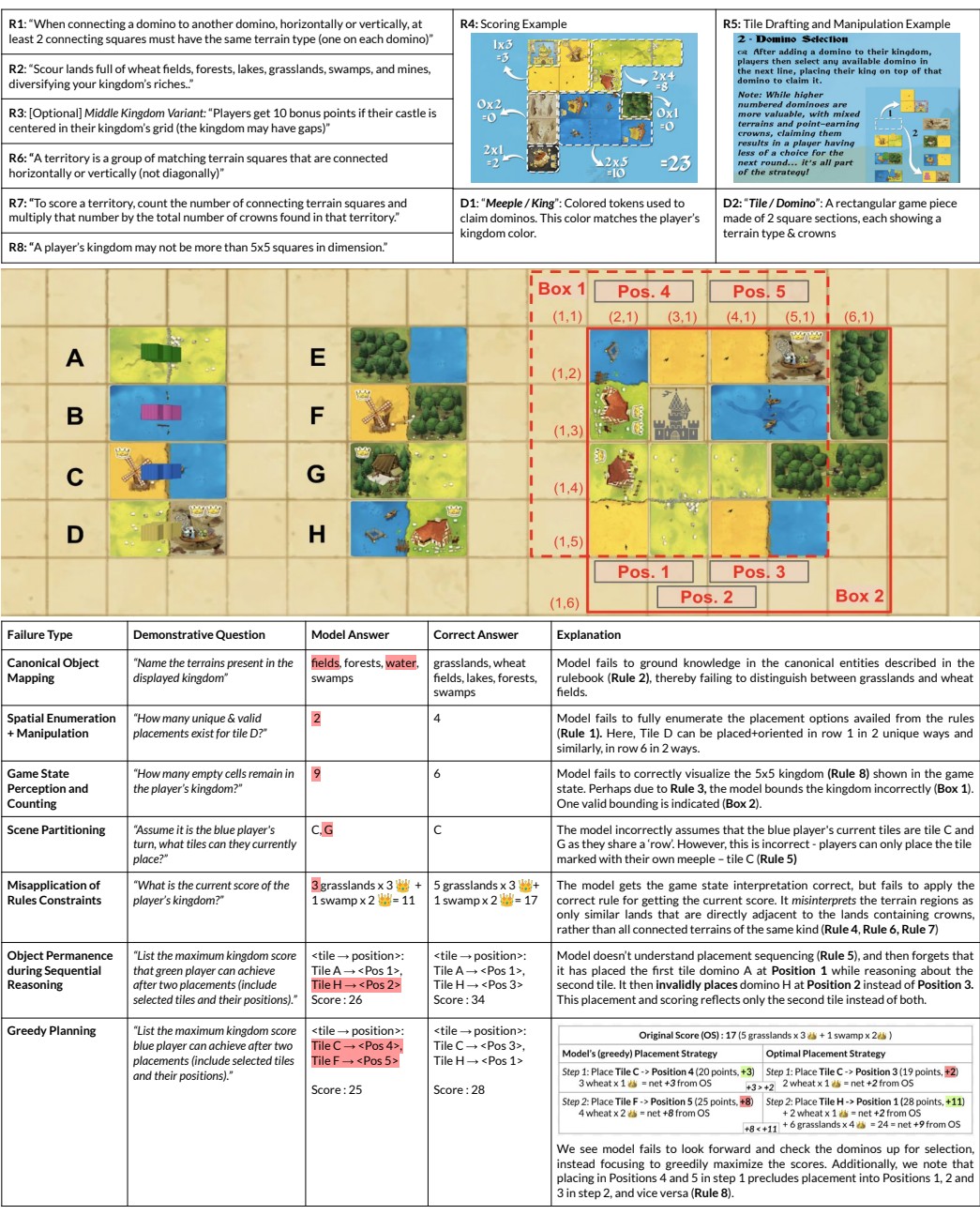

Figure 6: Illustration of common multi-modal game reasoning failures surfaced by LUDOBENCH. We provide key rule snippets from *Kingdomino* alongside a corresponding annotated game state, highlighting perception, rule application, and planning challenges. Annotation/Rulebook references are **bolded** in the explanation column. *Note: highlighted annotations and labels (e.g., position IDs, tile boxes) are used only for illustration clarity and are not part of the actual dataset inputs.*

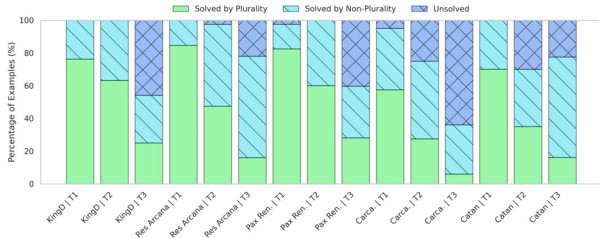

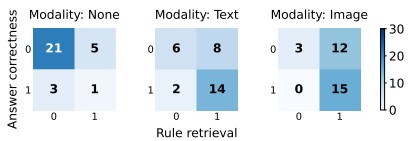

Figure 8: Error breakdown heatmaps (rule retrieval vs. answer correctness) for 30 Pax Ren. Tier-2 questions under three input modalities: no rules, text rules, and image rules.

Figure 7: Benchmark solvability across games and benchmark tiers. We aggregate 27 predictions per example (9 models × 3 rulebook input variants).

## 5.4 FURTHER ANALYSES

Due to the length of illustrated model outputs, we provide full illustrated demonstrations for each game and additional qualitative analyses in the Appendices. These materials are designed to make model behavior *auditable*: they localize errors to (i) visual state recovery, (ii) rule retrieval and alignment, and (iii) multi-step application and planning under constraints.

Across models, we observe recurring failure modes: *weak grounding* to rulebook images (icons/cards/tokens are not reliably mapped to their semantics), *brittle spatial reasoning* (localization, adjacency, and orientation errors), and *over-iteration* in multi-step settings that derails initially correct logic. With image rulebooks, models sometimes pattern-match to visually similar rulebook examples rather than using diagrams as grounding evidence. Finally, many Tier-3 failures stem from missing *intermediate state updates* when multi-action moves require updating the board before executing the next step.

Our extended analyses include:

- **Annotated Tier-3 puzzle walkthroughs** that pinpoint *where* failures occur in the reasoning pipeline (state extraction vs rule grounding vs action selection), and *how* they cascade into incorrect plans (Appx. D).

- **Rulebook modality comparison** with side-by-side outputs under *Text*, *Image*, and *None* rulebook modalities, illustrating when additional context improves rule retrieval but still fails to guarantee correct situated application (Appx. E).

- **Rulebook hallucinations and overfitting patterns survey** cataloging common error types (e.g., invented constraints, attribute transfer, and misread iconography), along with hypothesized triggers grounded in the provided state and rules (Appx. F).

- **When Claude Sonnet's "cautious" reasoning helps** analysis, highlighting cases where calibrated uncertainty and explicit verification reduce errors, as well as cases where caution does not prevent state-tracking failures (Appx. G).

- **o3 behavior analysis**, including overthinking, abstraction gaps, and brittle state tracking, which helps explain why strong general reasoning can still fail in long-horizon, component-heavy games (Appx. H).

Overall, LUDOBENCH offers a rich testbed for analyzing complex, situated multimodal game reasoning, and the appendices provide concrete, game-specific evidence to support these conclusions.

## 6 CONCLUSION

By introducing LUDOBENCH, we aim to highlight the gap between current game reasoning research and the real-world knowledge comprehension faced in the tabletop strategy games setting. This benchmark highlights the need for further model development focused on real-world situated game reasoning tasks, particularly in 'one-shot' learning scenarios where proper rule acquisition and game comprehension capabilities are paramount.

ACKNOWLEDGMENTS

This work was supported by Cisco Research. We thank David Jurgens, Rada Mihalcea, and Paramveer Dhillon for their insights and valuable discussions. We also thank the ARR and ICLR reviewers for their constructive feedback.

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

APPENDIX

## A    MODEL PREDICTION LENGTH ANALYSIS

Figure 9 compares the overall model prediction lengths across tiers. We see Perception tasks (Tier 1) are notably shorter, averaging approximately 100 tokens. In contrast, we see examples with rules integration (Tier 2) and optimization (Tier 3) notably increase prediction length, with Tier 3 outputs averaging approximately 600 tokens. However, we find that these additional prediction tokens do not simply correlate with improved performance, as evidenced in our extensive case studies.

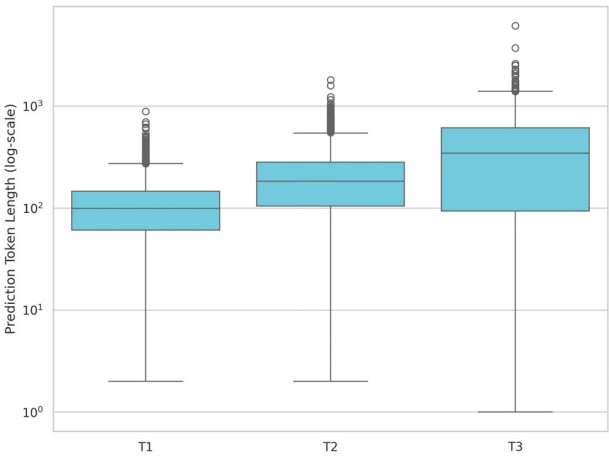

Figure 9: Prediction length vs example tier (log-scale). Models produce longer reasoning on the harder tiers, particularly tier 3, although we observe that longer rationales do not result in improved performance.

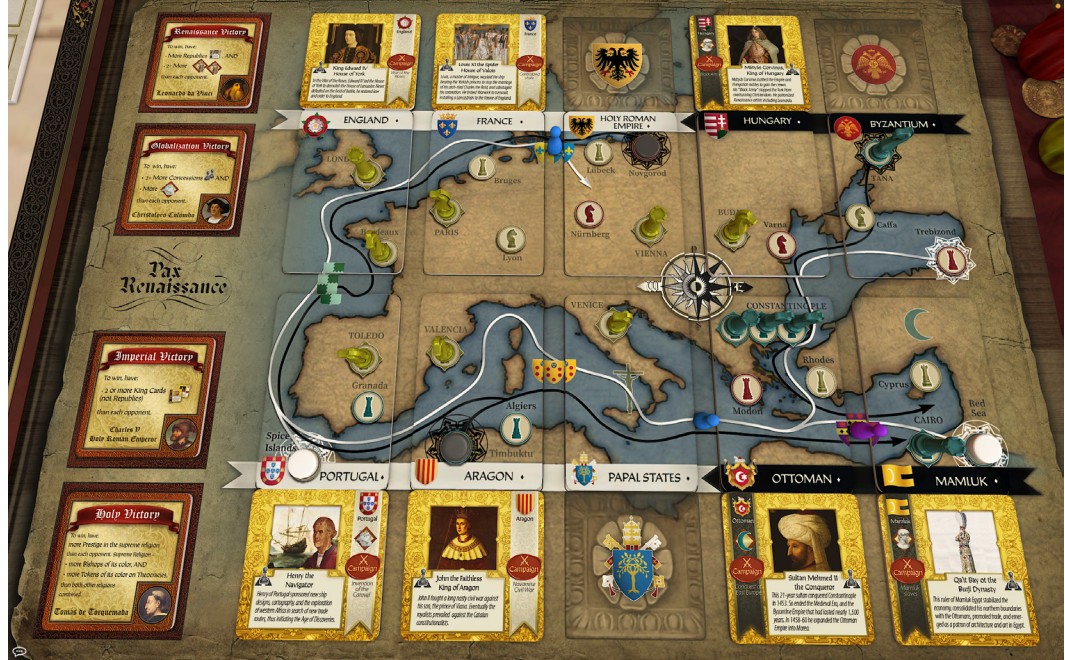

Figure 10: Game state for the example question in Appendix B.1.

## B RULES RETRIEVAL CASE STUDY

### B.1 MISINTERPRETATION OF GAME STATE DESPITE CORRECT RULE UNDERSTANDING

> **Question: How many blue concessions in total are there on the map?**
>
> **Gold Answer:** 2
> **Chain-of-Thought:**
> To determine the total number of blue concessions on the map, follow these steps:
>
> ### Step 1: Identify the blue concessions
> Blue concessions are represented by blue pawn tokens on the map. These tokens are located in the borders between empires.
>
> ### Step 2: Count the blue concessions
> From the map:
> 1. **Between England and France**: 1 blue concession.
> 2. **Between Holy Roman Empire and Hungary**: 1 blue concession.
> 3. **Between Hungary and Byzantium**: 1 blue concession.
> 4. **Between Byzantium and Mamluk**: 1 blue concession.
>
> ### Step 3: Add the counts
> 1 + 1 + 1 + 1 = **4 blue concessions**.
>
> ### Final Answer: 4

**Analysis:** The above example asks for the number of concessions (blue pawns in Figure 10). While gpt-4o correctly recognizes the rule ("Blue concessions are represented by blue pawn tokens on the map"), it fails to identify the concession on the map. This shows that *the model not only needs to understand the rule but also apply it with correct interpretation of game states*.

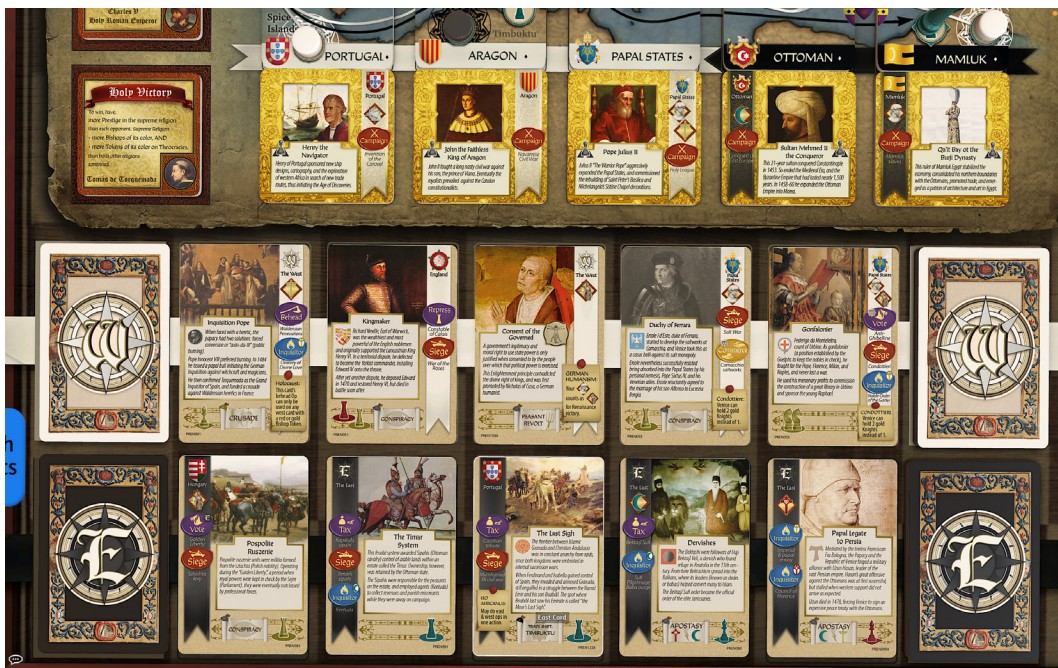

Figure 11: Game state for the example question in Appendix B.2.

## B.2 IMAGE-BASED KNOWLEDGE NEEDED INSTEAD OF TEXT

> **Question:** What is the name of the Prestige on the Duchy of Ferrara card?
>
> **Gold Answer:** Patron
> **Chain-of-Thought:**
> To determine the Prestige on the Duchy of Ferrara card, follow these steps:
> 1. **Locate the Duchy of Ferrara card**: The card is in the second row, fourth from the left.
> 2. **Identify the Prestige icon**: Prestige icons are typically represented with a gold diamond backdrop and are located in the banner section of the card.
> 3. **Read the Prestige type**: On the Duchy of Ferrara card, the Prestige is labeled as "Law."
> **Final Answer:** Law

**Analysis:** The above example asks for the name of a symbol (Prestige) on a specific card as shown in Figure 11. The rulebook shows the mapping between the name and symbol images (Figure 12). However, when converting the rulebook to text, the information in the image is lost and thus gpt-4o with text modality cannot access the necessary knowledge. This example shows that *certain questions require image-based knowledge representation rather than text*.

## C EVALUATION DETAILS

**Prompt construction.** For caching efficiency, the rulebook knowledge $R$ is prepended at the beginning of the input, followed by the game state image(s) $G$, then the natural-language query $Q$:

```
[Start Rules]
<(optional) rulebook knowledge>
[End Rules]
<game-state image(s)>
<question + format specification>
```

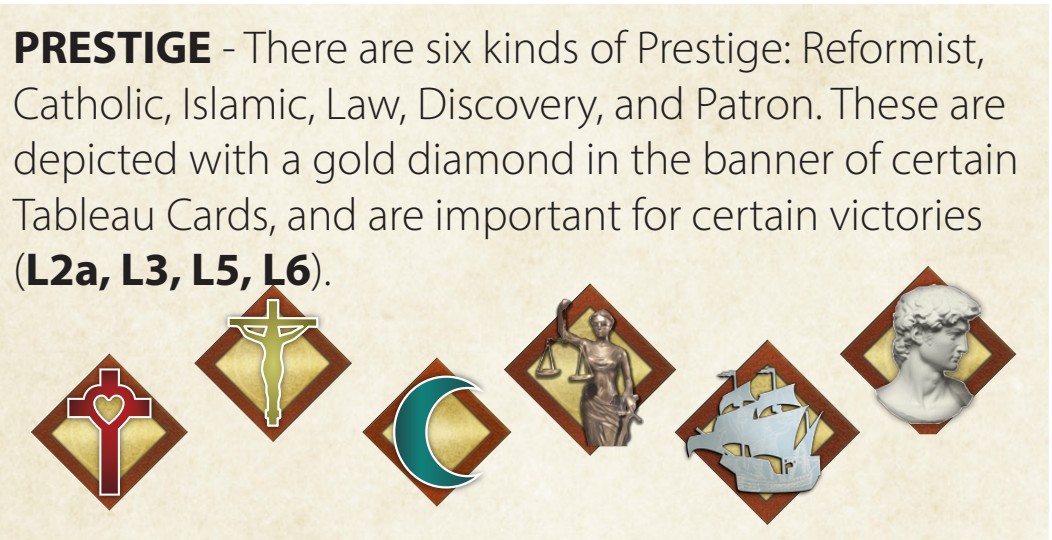

Figure 12: Picture of the relevant part in image-based rulebook to the example question in Appendix B.2.

| Model | API identifier | Version date |
|---|---|---|
| GPT-4.1 | gpt-4-1 | 2025-04-14 |
| GPT-4o | gpt-4o | 2024-11-20 |
| o1 | o1 | 2024-12-17 |
| o3 | o3 | 2025-04-16 |
| GPT-5.1 | gpt-5.1 | 2025-11-13 |
| Gemini 2.5 Pro | gemini-2.5-pro-preview-05-06 | 2025-05-06 |
| Gemini 2.5 Flash | gemini-2.5-flash-preview-04-17 | 2025-04-17 |
| Gemini 3 Pro | gemini-3-pro-preview | 2025-11-18 |
| Qwen-VL 2.5 32B | Qwen2.5-VL-32B-Instruct | 2025-02-01 |
| Claude 3.7 Sonnet | claude-3-7-sonnet-20250219 | 2025-02-19 |
| Claude 4.5 Sonnet | claude-sonnet-4-5-20250929 | 2025-09-29 |

Table 3: Exact API endpoints and version tags of evaluated models.

All experiments are run with the *OpenCompass* VLMEVALKIT (Contributors, 2025), which unifies data parsing, batching, logging and answer extraction across models. Hyper-parameters are kept at their model defaults except as noted here; we 1) raise max_tokens to each model's maximum, 2) impose a 300 s timeout with up to five automatic retries for API-served models.

Table 3 lists the API endpoints and version tags for the evaluated models: GPT-4.1 (OpenAI, 2025a), GPT-4o (OpenAI, 2024a;b), o1 (OpenAI, 2024c), o3 (OpenAI, 2025b), GPT-5.1 (OpenAI), Gemini 2.5 Pro (Google, 2025; Cloud, 2025b), Gemini 2.5 Flash (Cloud, 2025a), Gemini 3 Pro (Google Cloud), Claude 4.5 Sonnet (Anthropic, 2025) and Claude 3.7 Sonnet (Anthropic, 2025). For Qwen-VL 2.5 32B (Academy, 2025), we host the model locally, using 4 x Nvidia A40 GPUs, each with 48GB of GPU memory. Due to space constraints, we only provide Claude 3.7 and QWEN-VL analysis within the appendix sections.

**Note**: The 'reasoning_effort' parameter for o1 model is set to $Medium$ and for o3 model is set to $High$ while running these models.

## D    DETAILED ANNOTATED EXAMPLES (TIER 3)

We find that solving full problems involves nuanced strategic reasoning, including evaluating the full game state, managing resources, awareness of the visible states of opponents, and aligning

actions with long-term victory conditions. Even when the correct answer is brief (e.g., predicting a correct card, tile, or a few actions to be taken), the reasoning behind it is complex. Models must plan several steps ahead, navigate shifting board dynamics, and make context-sensitive tradeoffs. The same move may be optimal in one situation but detrimental in another depending on timing and configuration, and our dataset is designed to capture nuanced examples such as this.

Concretely, many questions involve multi-step strategies where the game state evolves with each action, e.g., placing a token, activating powers, or tracking changes to shared resources. Since intermediate states are not explicitly provided, models must internally simulate these transitions to maintain consistency. This makes the benchmark a strong test of whether language models can reason over implicit dynamics and adapt accordingly. In practice, even strong models often make subtle but meaningful mistakes, underscoring the challenge.

To showcase this complexity, we present one worked example from each of the five games in our benchmark. Each example includes:

- 
> **Ideal Reasoning Path**
>
> **Ideal Reasoning Path** – A step-by-step walk-through of how a human or ideal model would interpret the state of the game, retrieve the relevant rules, and arrive at the correct action.

> **Model Generated Output**
>
> **Model Generated Output** – A representative model's full reasoning output, highlighting intermediate thinking and reasoning, along with a final prediction.

- 
- **Side-by-Side Comparison Table** - A structured comparison of the output of the ideal model in key reasoning stages, followed by an analysis of where the model failed.

## D.1 KINGDOMINO ANNOTATION EXAMPLE

### D.1.1 QUESTION

```
What next domino should the blue player select to maximize the
score of their kingdom?  (According to Fig. 13) Assume the
game ends after placement of the pictured dominos.  Output the
following:  Domino letter, 'Previous kingdom score', eventual
'New kingdom score' after placement of all of the shown/selected
draftable dominos.  If necessary, you may take/simulate other
players' actions for them.  You should assume that the other
players will act cooperatively and will take actions to
best-support maximizing your score.  We are playing with the
harmony and middle kingdom rules.
```

### D.1.2 IDEAL REASONING PATH

> **Ideal Reasoning Path**
>
> The player begins with a score of 17 points:
>
> - Grassland region: 5 connected grassland tiles × 3 crowns = 15 points
> - Swamp region: 1 tile × 2 crowns = 2 points
>
> Total: 15 (grassland) + 2 (swamp) = 17 points.
> The game is being played with two bonus rules: **Harmony** and **Middle Kingdom**. Since only two dominos remain to be placed, it is no longer possible to complete the full $5 \times 5$ grid [Box 2] required for the Harmony bonus, so the model should ignore that.

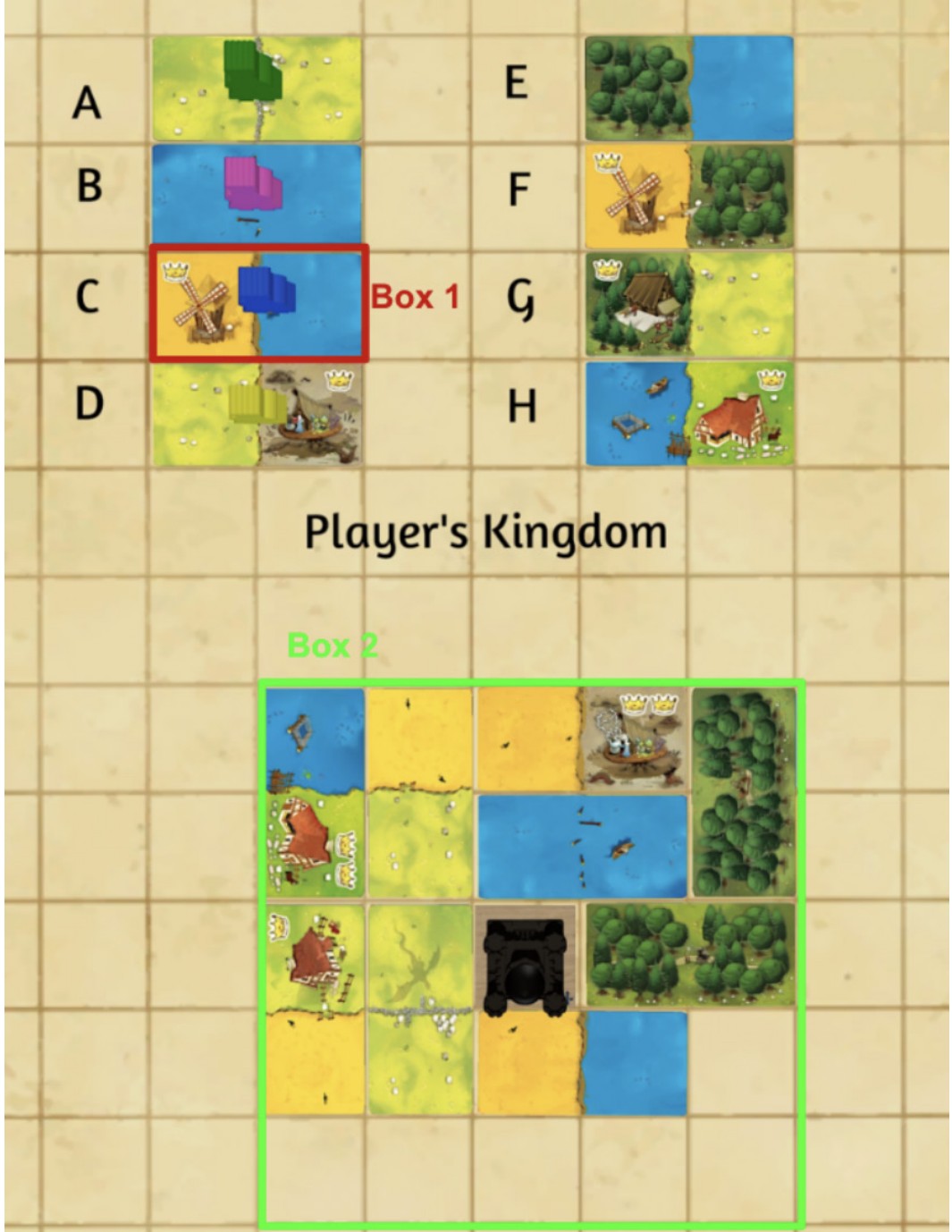

Figure 13: Game State with Annotations: KingDomino Detailed Annotated Example

However, the **Middle Kingdom** bonus (+10 points) is still achievable. To earn this bonus, the final kingdom must be centered around the castle. This can be done by placing the remaining dominos below the current bottom row, making the layout symmetrical [Fig. 13, Box 2].

Next, since we're analyzing for the **Blue player**, **Domino C** has already been selected in the previous round [Box 1]. Although the Blue player is drafting third, the other players (Green

and Pink) are playing cooperatively, so Blue is free to choose the best available domino from the remaining options: **E**, **F**, **G**, or **H**.

**Domino Evaluations:**

- **Domino E**: Adds no crowns and doesn't improve existing terrains.
- **Domino F**: Extends the wheat field region to 3 tiles with 2 crowns.
- **Domino G**: Expands the forest to 5 connected tiles with 1 crown.
- **Domino H**: Increases the grassland region from 5 tiles × 3 crowns to 6 tiles × 4 crowns, maximizing the grassland score.

**Final Score Calculation with Domino H:**

- Grassland region: 6 tiles × 4 crowns = 24 points
- Swamp region: 1 tile × 2 crowns = 2 points
- Fields region: 2 tiles × 1 crown = 2 points
- Middle Kingdom bonus: +10 points

**Total Final Score: 38 points**, the maximum achievable under the given constraints.

### D.1.3 MODEL GENERATED OUTPUT

---

**GPTo1_Azure (Modality: Image) Output**

**Step-by-step reasoning (abbreviated):**

1. Identify which two squares in the player's $5 \times 5$ grid remain unfilled.
2. Check each of the draftable dominos (A through H) to see:
   - Whether it can legally be placed to fill those two empty squares (maintaining the $5 \times 5$).
   - How many new points that placement would yield (including territory-multiplication by crowns).
   - Whether placing it achieves both Harmony (+5 points for no gaps in the $5 \times 5$) and maintains a centered Castle (+10 points for Middle Kingdom).
3. Compare the resulting final scores.

Under the (generous) assumption that the other players' selections do not block us and that domino G can be fit so as to extend both a forest and a farmland region (each gaining a crown), **domino G proves best**. It both completes the $5 \times 5$ (Harmony) and keeps the Castle centered (Middle Kingdom), thereby maximizing the final total.

**Final Answer:**
Domino letter: **G**, Previous kingdom score: **36**, New kingdom score: **57**

---

### D.1.4 MODEL REASONING BREAKDOWN

For the game of *KingDomino*, we conduct a detailed reasoning analysis by decomposing the model's decision process into six stages: understanding the game state, evaluating available dominos, analyzing possible placements, planning the optimal move, reflecting on strategic trade-offs, and finalizing the decision. This structured evaluation allows us to capture both valid model behaviors—such as correctly applying crown multiplication—and critical failure modes.

Common errors in the model's reasoning include miscalculating the starting score, hallucinating achievable bonuses like Harmony when they are not actually possible, and incorrectly splitting dominos during placement. The model also tends to over-prioritize bonus conditions at the expense of overall region optimization. These mistakes lead to an overestimated final score and selection of an invalid move.

A complete step-by-step comparison of the model's reasoning versus the ideal solution path for *KingDomino* is presented in Table 4.

## D.2 RES ARCANA ANNOTATION EXAMPLE

### D.2.1 QUESTION

```
What is the maximum guaranteed number of victory points the player
could have after 3 more actions. (According to Fig.   14) Assume
all other players have passed.  Output the answer as:  Previous
amount:  <num>, New amount:  <num>
```

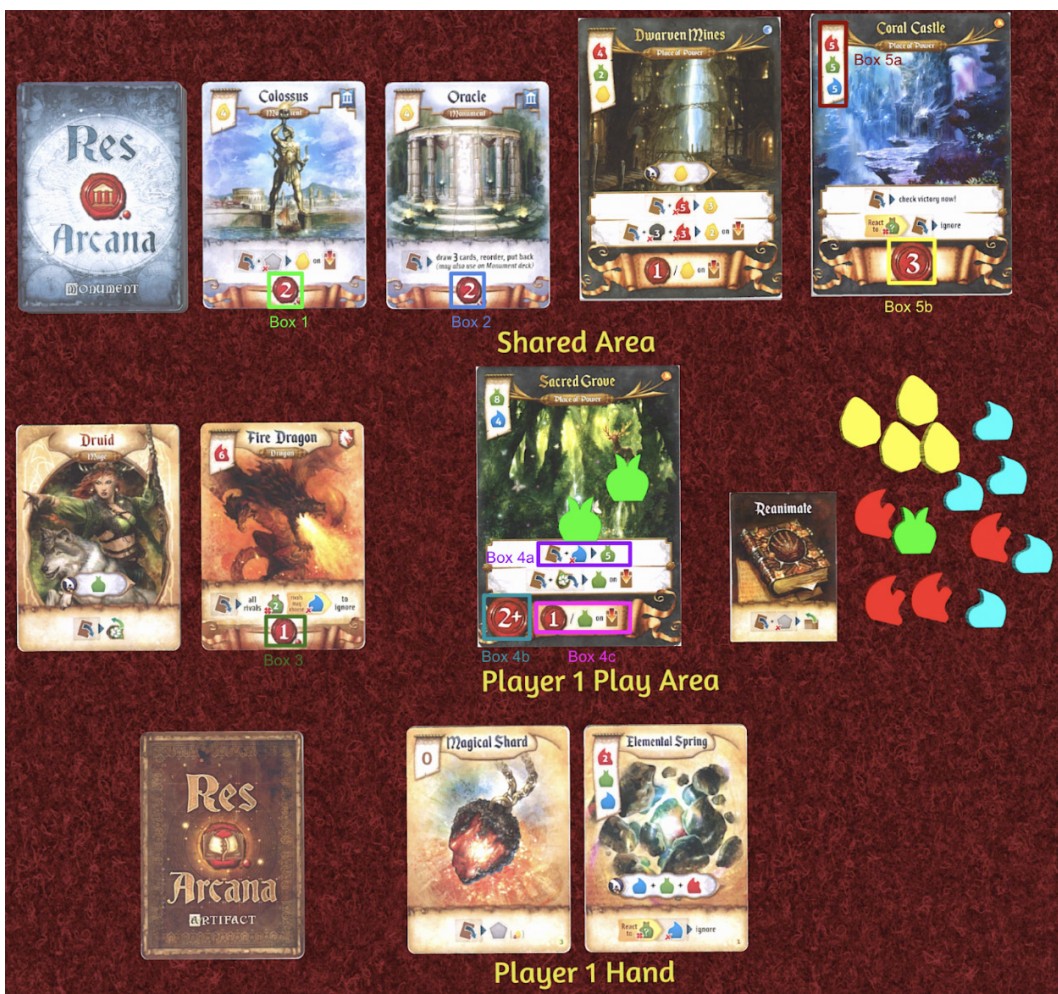

Figure 14: Game State with Annotations: Res Arcana Detailed Annotated Example

### D.2.2 IDEAL REASONING PATH

> **Ideal Reasoning Path**
>
> The player begins with 5 victory points (VP). This total comes from three sources:
>   - 1 VP from the Fire Dragon card [Fig.  14 Box 3],
>   - 2 VP from the Sacred Grove card [Box 4b], and

> - 2 bonus VP from the two Life essences stored on Sacred Grove, according to the condition shown [Box 4b].
>
> The goal is to find the maximum number of VP the player can reach in three additional actions, assuming other players don't interfere.
> For the first action, the model should recognize that the player can spend 4 gold essences to purchase one of the face-up monument cards, either the Colossus [Box 1] or the Oracle [Box 2], each of which provides +2 VP.
> Next, for the remaining two actions, the player can work toward buying the Coral Castle which is worth +3 VP [Box 5b]. However, the player currently lacks the full set of essences needed for this purchase: 5 Elan, 5 Life, and 5 Water [Box 5a]. These essences which are short can be gathered in these actions:
>
> - Use the top ability of the Sacred Grove card [Box 4a] to spend 1 Water and gain 5 Life, raising the essence pool to 6 Life, 4 Water, and 4 Elan.
> - Discard any card from hand to gain 1 Water and 1 Elan, which completes the required essences for the Coral Castle.
>
> By following this strategy, the player adds +2 VP from the monument and +3 VP from the Coral Castle, reaching a final total of 10 VP.

### D.2.3 MODEL GENERATED OUTPUT

> **GPTo1_Azure (Modality: Text) Output**
>
> **Step-by-step reasoning:**
>
> 1. From the image, the player's current scoring comes entirely from *Fire Dragon* (1 VP) plus the *Sacred Grove* with three Life on it (2 base VP + 3 for the Life = 5 VP). **Total so far = 6 VP.**
>
> 2. To claim a face-up monument (worth 2 VP) requires paying 4 Gold. The player has 2 Gold in pool and 2 artifacts in hand (*Magical Shard*, *Elemental Spring*). Each can be discarded (in two separate actions) for +1 Gold, giving exactly 4 Gold.
>
> 3. The third action can then be used to claim one of the 2 VP monuments from the center.
>
> Because we have exactly three actions (two discards plus the claim) and no time left to add more Life to the Grove, this yields a guaranteed increase of +2 VP, from 6 up to 8.
>
> **Final Answer:**
> Previous amount: **6**, New amount: **8**

### D.2.4 MODEL REASONING BREAKDOWN

Similar to the *KingDomino* analysis, we decompose the model's reasoning process for *Res Arcana* into six cognitive stages: understanding the current game state, evaluating resources, analyzing scoring routes, planning actions, reflecting on alternatives, and concluding with the final VP calculation. This structured approach allows us to examine not just the model's final prediction, but also how it arrives at that conclusion through intermediate steps.

In this case, the model demonstrates partial understanding of the discard mechanics and action sequencing but makes critical perceptual errors. It misreads the number of Life essences on the *Sacred Grove*, leading to an incorrect initial VP estimate, and assumes fewer Gold resources than available, prompting unnecessary discards. Although the model's plan is internally consistent given its assumptions, it fails to consider a higher-value strategy that combines the *Sacred Grove* with the *Coral Castle*, resulting in a suboptimal 8 VP outcome instead of the achievable 10. The detailed step-by-step comparison for *Res Arcana* is provided in Table 5.

### D.3  PAX RENAISSANCE 2E ANNOTATION EXAMPLE

#### D.3.1  QUESTION

```
Given the provided game state, how can the blue player win the
game in 2 actions?  (According to Figure  15)
```

#### D.3.2  IDEAL REASONING PATH

> **Ideal Reasoning Path**
>
> Ideally the model should realize that the blue player wants to win in the next 2 actions, one of which must be reserved for declaring victory. So within one action the blue player should gain one of the active victory conditions.
> From the game state the model should identify the fact that all victory conditions are active. [Fig.  15 Box 1]
> The current game state of the blue player is:
>
> - England Kingdom card [Box 10] (card is located in England [Box 10a], and has the Campaign operation [Box 10b])
> - Cornish Tin [Box 11] (card is functional for England [Box 11a], possesses a Law Prestige [Box 11b], and has the Vote & Commerce operations [Box 11c])
> - Tolfa Alum [Box 12] (card is functional for Papal States [Box 12a], and has the Vote & Commerce operations [Box 12b])
> - Almeida Armada [Box 13] (card functional for Portugal [Box 13a], possesses a Discover Prestige [Box 13b], and has the Repress & Corsair operations [Box 13c])
> - 3 Florins [Box 14]
> - 2 Blue Concessions (Each located on the borders of England-France [Box 2] and France-HRE [Box 3])
>
> The current game state of the purple player is:
>
> - Holy Roman Empire kingdom card [Box 5] (card is located in HRE [Box 5a], and has the Campaign operation [Box 5b])
> - Spanish Tercio [Box 6] (card is functional for the West region [Box 6a], and has the Siege operation [Box 6b])
> - Isfendiyarid dynasty [Box 7] (card is functional for Byzantium [Box 7a], and has the Repress & Corsair operations [Box 7b])
> - Byzantium kingdom card [Box 8] (card is located in Byzantium [Box 8a], and has the Campaign operation [Box 8b])
> - 4 Florins [Box 9]
> - 1 Purple Concession (Located on the borders of Ottoman-Mamluk [Box 4])
>
> Examining the game state, the model should realise that in order to achieve an "Imperial Victory", the blue player needs at least 2 more king cards than all other opponents (in this case, Purple), which means they need to get 3 total (ie, 2 more) king cards to win. The player cannot do this in one action.
> For the "Holy Victory", the minimum condition of requiring a Bishop in supreme religion is not satisfied. The player cannot attain "Renaissance Victory" in one action by getting more republics and 2 more law prestige than the opponent. The blue player already has a Discover Prestige (more than opponent) and needs one more concession (blue pawn) on the map to attain "Globalization Victory".
> In order to place one more concession, one action is sufficient: Playing the West Tableau Ops (ie, activating all of the cards on the left side of their personal board) and triggering the "Vote" operation on Cornish Tin results in a victory because the majority (To win a Vote operation, the number of concessions blue player has around the England kingdom should be more than any other players.

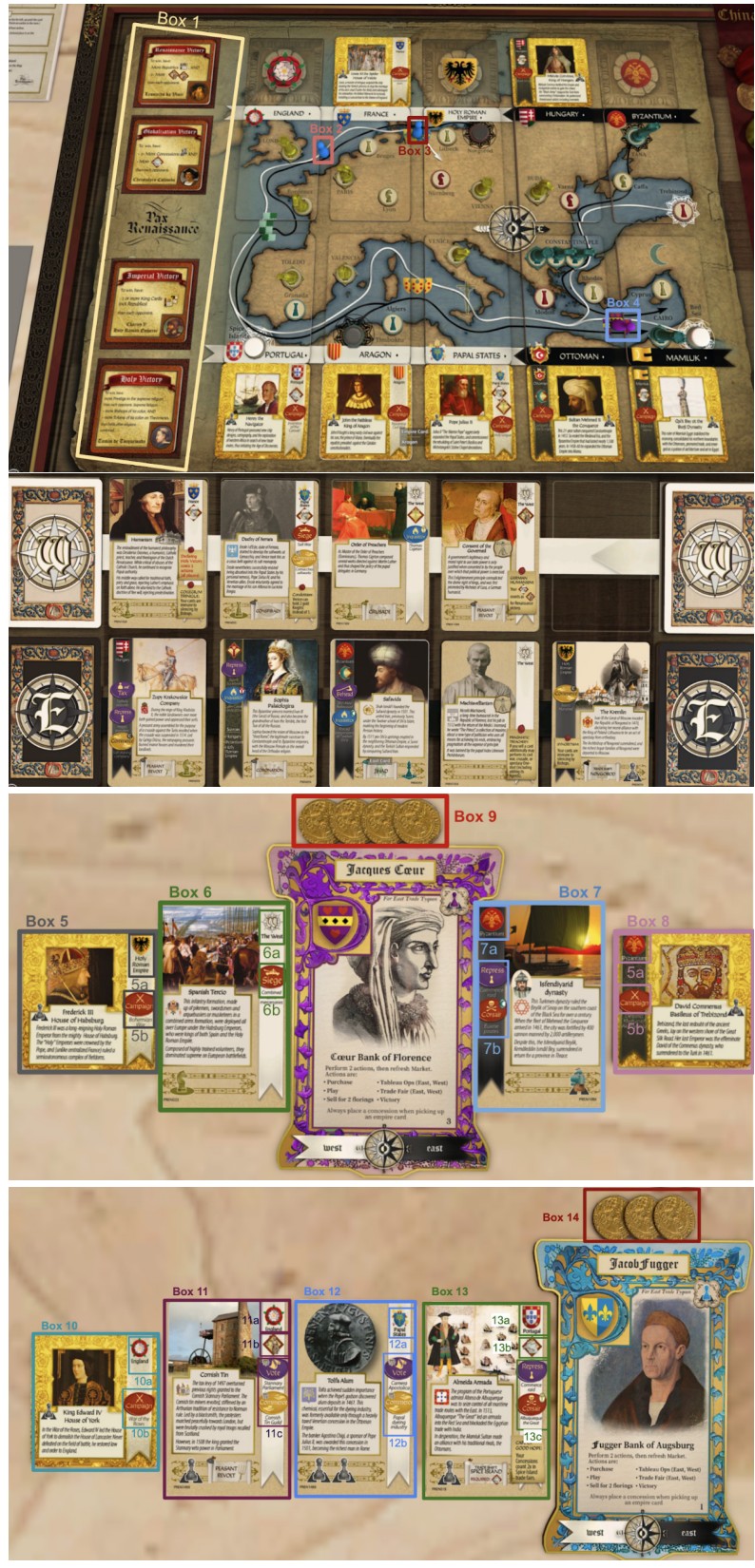

Figure 15: Game State with Annotations: Pax Renaissance Detailed Annotated Example (From the top: 1) Annotated Shared Board State, 2) Annotated Cards for Purchase, 3) Annotated Purple Player's Game State, 4) Annotated Blue Player's Game State)

> In this case the only concession surrounding England is that of the blue player). Hence, there would be a regime change in England, that gives the blue player one more concession (A byproduct of successful vote is a regime change, and this is accompanied by gaining a concession). Then at last the model should ideally say that the blue player declares Globalization victory.

### D.3.3 MODEL GENERATED OUTPUT

---

**GPT4O_AZURE (txt) Output**

**Step-by-Step Solution:**

1. **Analyze the Victory Conditions**:

   The blue player needs to meet the conditions for one of the active Victory Squares. Based on the game state:

   - **Imperial Victory** is active, requiring the player to have at least 2 more King cards than each opponent.

   - The blue player currently has **King Edward IV** in their Tableau, and no other Kings. To win, they need to acquire at least 2 more King cards.

2. **Identify Available Actions**:

   - The blue player can use the **Vote** operation on the **Tolfa Alum** card in their West Tableau to flip the Papal States Empire Square from a Republic to a King. This will add one King card to their Tableau.

   - The blue player can use the **Campaign** operation on **King Edward IV** to conquer Portugal, adding another King card to their Tableau.

3. **Florin Management**:

   The blue player has 3 Florins available.

   The **Vote** operation costs 1 Florin (to repress the opponent's Concession in the Papal States Border).

   The **Campaign** operation costs 1 Florin per Knight used in the attack.

   The blue player has enough Florins to execute both actions.

4. **Execute Actions**:

   - First, use the **Vote** operation to flip the Papal States to a King.
   - Second, use the **Campaign** operation to conquer Portugal.

5. **Declare Victory**: After acquiring 2 additional King cards (Papal States and Portugal), the blue player will have 3 King cards in total, meeting the conditions for **Imperial Victory**.

---

In the above output, we have highlighted the performance of different reasoning sub-tasks.

- Green indicates correct rule retrieval
- Blue indicates correct game state analysis.
- Red indicates the inability to analyze game state and apply rules accordingly.

Model Prediction Analysis: While the model correctly identifies that the "Imperial Victory" condition is active, it fails to check for other active victory conditions that are equally relevant in the current game state. Ideally the model should evaluate all possible applicable paths to victory before deciding on optimal action, especially when the board contains multiple overlapping win conditions. This failure to reason over alternate options highlights the lack of comprehensive evaluation – a core component of strategic reasoning.

More critically, although the model correctly notes that the blue player posses the king card, it misapplies the imperial victory rule. This condition requires the player to have at least two more kings than opponents, and in the given state, the purple player holds 2 king cards. This means

the blue player actually requires 3 more king cards to win via this route – something the model overlooked entirely. This suggests not only incomplete reading of the rule but also failure to integrate the opponent's state into its reasoning. The model's errors continue during action planning. This can be seen in these cases:

- The model notices the Papal States card but assumes the "vote" will trigger a republic effect. In reality the region is still on the throne, so voting merely moves the card to the tableau. The board position is read correctly, yet the relevant rule is applied incorrectly.

- The model correctly checks it has enough florins to launch a campaign but ignores that defenders equal attackers, making success uncertain. Here the rule (cost) is used properly, but the spatial parity on the map is missed.

Overall, the model fails to recognize that it is one action away from victory – does not choose that action. This highlights how a seemingly short and straightforward question (e.g. "What should blue do next to win") actually demands a multi-step, condition-aware reasoning and continuous tracking of a dynamic game state – a task that proves difficult even for state-of-art language models. These kinds of errors reinforce our broader argument: while the surface form of the answers may appear simple, the underlying reason required is anything but this.

### D.3.4 MODEL REASONING BREAKDOWN

In the Pax Renaissance case study, we find that while the model is capable of retrieving individual rules and game components—such as identifying victory conditions, tracking resource costs, and recognizing specific actions—it fails to integrate these elements into a valid multi-step plan. The model incorrectly prioritizes Imperial Victory, overlooking other feasible victory paths like Globalization Victory. Additionally, it fails to incorporate opponent information, ignores turn-order constraints, and miscalculates action sequencing, leading to overconfident but invalid predictions.

This highlights a broader pattern where models struggle not with isolated rule retrieval, but with coordinating reasoning across interconnected steps. The inability to simulate intermediate game states and adjust strategy accordingly results in hallucinated outcomes and suboptimal planning.

A detailed step-by-step breakdown of these failure modes is provided in Table 6.

### D.4 CATAN ANNOTATION EXAMPLE

### D.4.1 QUESTION

Assume: 1) no-inter player trading occurs, 2) there are no development cards left to draw, 3) in-hand VP cards count as points. What is the maximum number of lumber resources the blue player can get by the end of this turn, given the player rolls a 5 (According to **Figure 16**)? Return your answer in the following format: [prev_cards: , new_cards: ]

### D.4.2 IDEAL REASONING PATH

> **Ideal Reasoning Path**
>
> Ideally, the model should enumerate all possible ways the player could gain lumber this turn, rule out non-applicable options under the given constraints, and continuously update the game state as each valid action is applied.
> From Fig. 16,
> The current blue player has:
>
> - 3 knight cards [Box 5], and the largest army card [Box 6] on the players table
>
> - 2 lumber [Box 7], 1 wool [Box 8], 1 grain [Box 9], and 2 ore [Box 10] resource cards from the resource pool section.

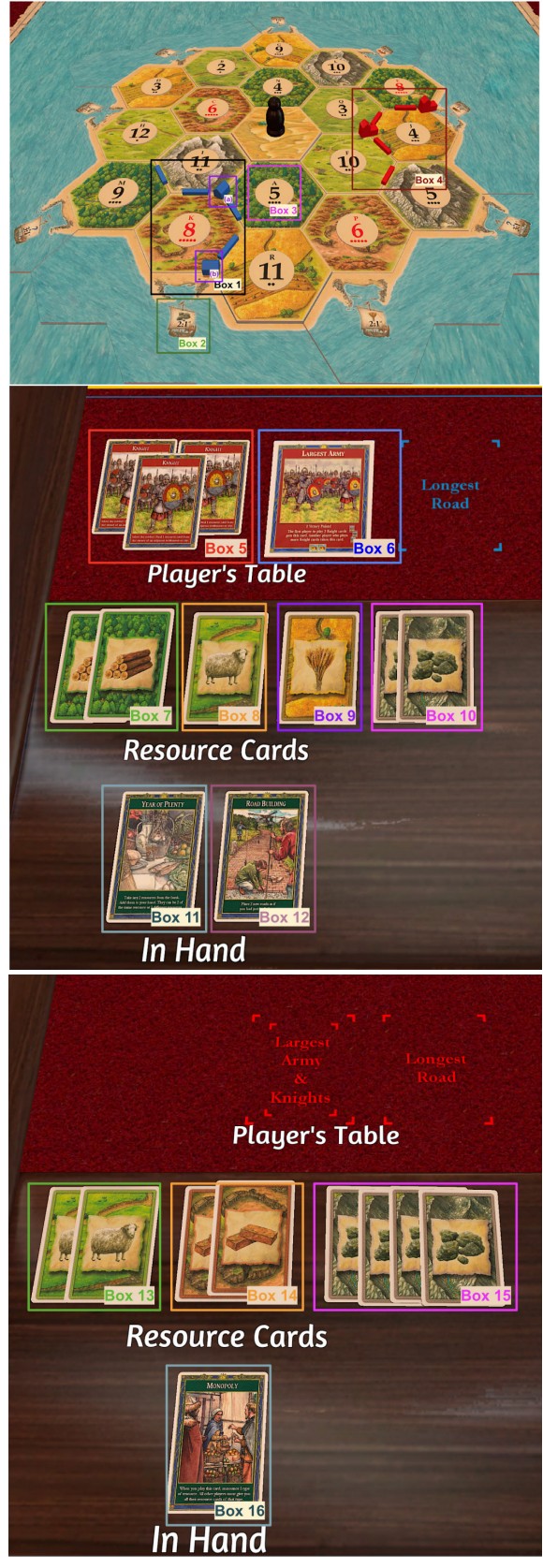

Figure 16: Game State with Annotations: Catan Detailed Annotated Example (From the top: 1) Annotated Shared Board State, 2) Annotated Blue Player's Game State, 4) Annotated Red Player's Game State)

> - Has one year of plenty [Box 11], and one road building card [Box 12] In hand development cards.
> - From the board state; blue player has 2 settlements [Box 1 (a), (b)], and 4 roads [Box 1].
>
> The current red player has:
>
> - 2 sheep [Box 13], 2 Brick [Box 14], 4 Ore [Box 15] resource cards from the resource pool section
> - Has one monopoly development card [Box 16] in the Hand. From the board state; red player has 2 cities and 4 roads [Box 4]
>
> By the rules of the game the player can gain resource cards either from trading with other player (but given as not allowed in the question restrictions), from harbor trade (3:1 or 2:1), or trade from central resource pool for (4:1), or by using Year of Plenty development cards. According to the given game state, the blue player has 2 lumber resources, and also the blue player has 1 settlement on the harbor (ore 2 : 1) [Box 1 (b)], and also one settlement in the intersection of hexes 5-forest, 11-mountains, and 8-hills. In the current round, it's given that the player rolls a 5. According to the rules, the blue player would gain 1 lumber resource from the 5-forest for the one settlement present on the intersection. So the current total number of lumber resources would be 3.
>
> In order to get the maximum number of lumber resources, the blue player has to do the following in the current turn: Use the harbor trade (ore 2:1):
>
> - Harbor Trade: trade 2 ore resource cards for one resource card of lumber type. Giving +1 lumber resource card
> - Use the development card Year of Plenty: playing this development card the player can get 2 lumber resource cards from the central resource pool. Giving +2 resource cards.
>
> After this there is no more trade possible and no cards that can be played in this turn in order to get more lumber resources.
>
> So, therefore, the maximum number of lumber resources the blue player can get by the end of this turn: 1 from the die roll, 1 from harbor trade, and 2 from the development card. So the total number of lumber resources with the blue player would be 6 lumber cards.

### D.4.3 MODEL GENERATED OUTPUT

> **GEMINI_3_PRO (img) Output**
>
> To find the maximum net number of lumber resources the blue player can gain by the end of the turn, we analyze the game state step by step:
>
> 1. **Current Resources (Start of Turn):**
> Looking at the *Player's Table* image for the blue player (indicated by the blue-bordered development cards *Year of Plenty* and *Road Building*), we count the resource cards in hand:
>
> - **Lumber (Wood):** 2 cards
> - **Wool (Sheep):** 1 card
> - **Grain (Wheat):** 1 card
> - **Ore (Rock):** 2 cards
> - **Brick:** 0 cards
> - **Total:** 6 cards
>
> Thus, `prev_cards` (Lumber) $= 2$.
>
> 2. **Resource Production (Dice Roll $= 5$):**
> We examine the board to see whether the blue player has any settlements or cities adjacent to a terrain hex marked with the number $5$.
>
> - **Blue's settlements:**

> - One settlement is on the coast at the 3:1 harbor, adjacent to the 9-Forest.
> - The other settlement is inland at the intersection of 9-Forest, 11-Mountain, and 8-Hills.
> - **Hexes with number** 5:
>   - 5-**Forest:** Located in the inner ring. Its corners touch the Desert, 11-Mountain, 10-Pasture, 6-Hills, 11-Fields, and 5-Mountain.
>     The blue settlement on the 11-Mountain corner (shared with 9-Forest and 8-Hills) is **not** adjacent to the 5-Forest.
>   - 5-**Mountain:** Located on the outer ring (bottom right). Blue has no pieces near this hex.
>
> Therefore, the blue player receives 0 resources from the dice roll.
> 3. **Trading and Development Cards:**
>   - **Trading:** Trading with other players is not allowed. Blue can trade with the bank and has a settlement on a generic 3:1 harbor (visible on the 9-Forest coast), which allows trading 3 identical resources for 1 resource of any choice.
>     Current hand: 2 Lumber, 1 Wool, 1 Grain, 2 Ore.
>     Since Blue does not have 3 of any resource, **no bank trades are possible initially**.
>   - **Development cards:** Blue has *Year of Plenty* and *Road Building* available to play.
>     - *Year of Plenty:* Take any 2 resource cards from the bank.
>     - *Road Building:* Place 2 roads (does not grant resources).
>
> 4. **Maximizing Lumber:**
> To maximize lumber, the optimal action is to play the *Year of Plenty* card.
>   - **Action:** Play *Year of Plenty* and choose 2 Lumber cards.
>   - **New hand:** 4 Lumber, 1 Wool, 1 Grain, 2 Ore.
>   - **Further actions:** Can Blue obtain more lumber after this?
>     - Blue still does not have 3 of any non-lumber resource (1 Wool, 1 Grain, 2 Ore), so harbor trading cannot produce an additional lumber.
>     - Using *Year of Plenty* to take resources for trading (e.g., taking 2 Ore to reach 4 Ore and then trading 3 Ore for 1 Lumber) yields a net gain of only 1 Lumber (ending at 3 Lumber), which is worse than directly taking 2 Lumber (ending at 4 Lumber).
>
> Thus, new_cards (Lumber) = 4.
> **Final Answer:**
>
> $$[\texttt{prev\_cards} : 2, \texttt{new\_cards} : 4]$$

In the above output, we have highlighted the performance of different reasoning sub-tasks.

- Green indicates correct rule retrieval
- Blue indicates correct game state analysis.
- Red indicates the inability to analyze game state and apply rules accordingly.

Model Prediction Analysis: The model correctly recovers key parts of the blue player's private state: especially the resource cards and in-hand cards; but fails to reconstruct the full public board state. In particular, it misses critical spatial details such as where settlements are placed and which harbors are available, leading to an incomplete and partially incorrect representation of the game.

This breakdown is especially evident when the model reasons about resources gained from a dice roll. While it understands the rule that players collect resources from adjacent hexes, it fails to identify that a blue settlement is located at the intersection adjacent to the "5-forest" hex, and therefore does not award the corresponding lumber. In other words, the rule is retrieved correctly, but the necessary board evidence (settlement placement at the relevant intersection) is not detected or integrated.

The same pattern continues in its exploration of alternative resource-acquisition paths. The model recognizes that there are multiple mechanisms to obtain resources, but because its inferred game state is already incorrect, it prematurely stops checking other viable lines: most notably the harbor trade route (2 ore : 1 harbor trade), [Box 2] and the settlement located on the harbor [Box 1(b)]. Since the harbor and settlement are not detected in the first place, the model never fully considers this path as a legitimate option.

Overall, the model enumerates plausible reasoning routes, but the failure to fully ground them in an accurate game state prevents correct optimization. As a result, it produces a partially correct answer and fails to identify the maximum lumber that the blue player can obtain under the true board configuration.

### D.4.4 MODEL REASONING BREAKDOWN

In the Catan case study, we observe that while the model can correctly recover *private, text-like/card-like* information: such as the blue player's resource cards and development cards; it struggles to ground its reasoning in the *shared, spatial* board state. Specifically, the model fails to reliably detect board-grounded settlement placements and their adjacency to key board elements (e.g., harbors), which leads to incorrect downstream inferences about what actions and resource-gain opportunities are actually available.

This weakness becomes most apparent during turn-start resolution: although the model retrieves and applies the resource-production rule, missing settlement placement causes an incorrect resource update (failing to add the additional resource the player should receive). The same grounding error cascades into path evaluation: the model correctly follows the stated constraint (no player-to-player trading), but misreads a board-dependent condition for trading and consequently discards a valid option.

As a result, the model's planning process prunes feasible paths early, converging on a suboptimal, partially correct sequence instead of the true optimal multi-step plan. Overall, this case highlights a broader pattern where models can retrieve isolated rules and components, but fail to integrate them into a consistent plan when successful reasoning requires accurate spatial state reconstruction and iterative state updates. A detailed breakdown of these failure modes is provided in Table 7

### D.5 CARCASSONNE ANNOTATION EXAMPLE

#### D.5.1 QUESTION

```
Considering the situation that the yellow player has two turns
one after the other, and that the game ends after these 2 turns.
Which tiles should the yellow player choose from the available
tile options (A, B, C) so as to get the maximum score by the
end of the game (According to Figure 17)?. Assume the players
all have multiple meeples in their private supplies. Return the
answer in the following format: [total_points_turn = <the maximum
points by the end of two turns, before end-of-game scoring>,
total_points_final = <maximum number of points at the end of
game>]
```

#### D.5.2 IDEAL REASONING PATH

> **Ideal Reasoning Path**
>
> Ideally, the model should recognize that the game ends after two consecutive turns by the yellow player (per the given constraint), correctly reconstruct the current game state, and enumerate the different legal ways to place the available tiles and meeples to maximize the yellow player's total score by game end.
> Given game state (As shown in Fig. 17):

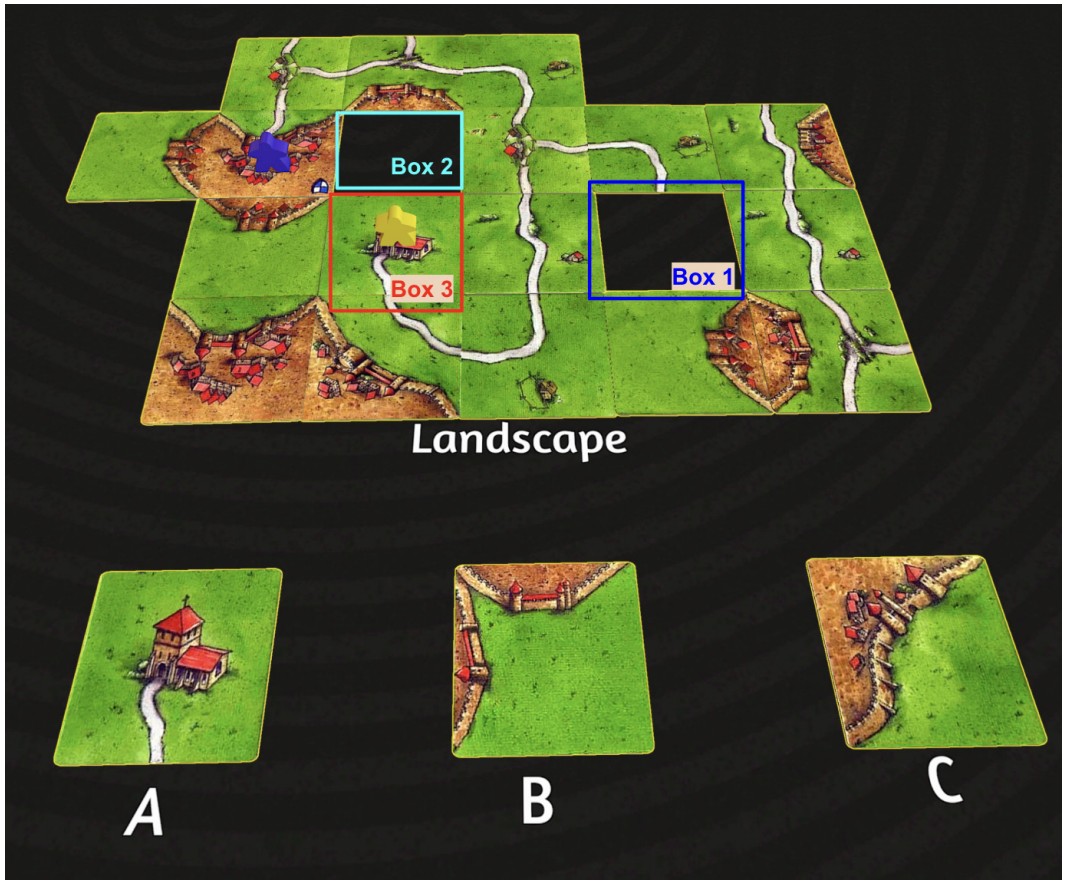

Figure 17: Game State with Annotations: Carcassonne Detailed Annotated Example

- Yellow player has one meeple (monk) on the incomplete monastery [Box 3], which is surrounded by 8 tiles.
- Blue player has one meeple (knight) on an incomplete city spanning 3 tiles.
- There are 2 void spaces: [Box 1] and [Box 2].

Scoring rules:

- Each completed monastery scores 1 point per tile in the feature.
- Each tile in a completed city scores 2 points per tile, and each coat of arms in the city scores an additional 2 points.
- At game end, farmers are scored based on completed cities: a farmer scores

$$(\# \text{ completed cities}) \times 3.$$

Placement constraints: Tiles may be placed as long as they continue the landscape (matching adjacent edges). From the given options, [Box 2] can be filled with Tile B or Tile C, and [Box 1] can be filled with Tile A.

Optimal plan to maximize yellow's score:

- Turn 1: Place Tile B in [Box 2] so that it fits the landscape (city edges match). Then place a yellow meeple in the field as a farmer. This placement completes the monastery feature, so yellow scores 1 point per tile in the monastery:

$$9 \text{ points.}$$

The monk meeple is returned to the pool.

- **Turn 2:** Place Tile A in [Box 1] so that it matches the surrounding landscape, then place a yellow meeple on the monastery as a monk. Since the monastery is immediately complete, yellow again scores 1 point per tile:

$$9 \text{ points.}$$

  The monk meeple is returned to the pool.

- **End of Game:** Score remaining yellow meeples. The only remaining meeple is the farmer placed in Turn 1. This farmer is adjacent to 3 completed cities (two completed after placing Tile B in [Box 2], and one completed city below [Box 1]). Hence the farmer scores:

$$3 \times 3 = 9 \text{ points.}$$

Therefore, the total points gained during the final two turns are:

$$9 \text{ (Turn 1)} + 9 \text{ (Turn 2)} = 18.$$

Including the farmer score at game end, the yellow player's total gain is:

$$18 + 9 = 27 \text{ points.}$$

### D.5.3 MODEL GENERATED OUTPUT

**GPT_5.1_OpenAI (img) Output**

**Final Answer:** [total_points_turn = 9, total_points_final = 13]

**Step-by-step reasoning:**

1. **Current situation**
   - Yellow has a single meeple on an incomplete 3-tile road.
   - There are two empty spaces in the interior, but none of the available tiles (A, B, C) legally fit those gaps.
   - So Yellow must place their tiles on the outer edge of the landscape.
   - Only Yellow's road and any new features Yellow creates will score for them.

2. **Best use of tile A (monastery with road)**
   - Tile A can be placed so that:
     - Its road does **not** connect to Yellow's existing road (otherwise we'd only add 1 more road tile and still leave it incomplete).
     - The monastery is adjacent to **5 tiles** by the time the game ends (including those already there and those Yellow adds with their second tile).
     - Yellow places a meeple as a **monk** on this monastery.
     - Because the monastery is not yet fully surrounded, it does **not** score during the turn.
     - At game end, an incomplete monastery scores 1 point for each surrounding tile plus the monastery tile itself:
       * 5 adjacent tiles + 1 (the monastery tile) = **6 points**.

3. **Best use of tile B or C**
   - Tile B or C can be placed so they complete a **small 2-tile city** (using one pre-existing city tile on the edge).
   - Yellow places a meeple as a **knight** on their city segment.
   - The city is completed immediately, so during the turn Yellow scores:
     - 2 tiles × 2 points per tile = **4 points**.
     - The knight is then returned to their supply and does not participate in end-game scoring.

4. **Yellow's existing road**
   - The road with Yellow's meeple remains incomplete; it consists of 3 tiles.
   - At game end, an incomplete road is worth 1 point per tile:
     - 3 tiles = **3 points**.

5. **Total scoring**
   - **During the two turns**
     - Completed 2-tile city: 4 points.
     - No other feature completes during the turns.
     - So: `total_points_turn` = 4 (city) + 0 (monastery, road) = **4**.
   - **End-of-game scoring**
     - Incomplete monastery created with tile A: 6 points.
     - Incomplete 3-tile road with Yellow's original meeple: 3 points.
     - So end-game adds: 6 + 3 = **9 points**.
   - **Grand total**
     - 4 (during turns) + 9 (final scoring) = **13 points**.

Among all ways to use two of the tiles A, B, and C, this gives Yellow the maximum total of **13 points** by the end of the game.

In the above output, we have highlighted the performance of different reasoning sub-tasks.

- Green indicates correct rule retrieval
- Blue indicates correct game state analysis.
- Red indicates the inability to analyze game state and apply rules accordingly.

Model Prediction Analysis: The model fails to reconstruct the game state accurately. While it notices a yellow meeple on the landscape, it misidentifies the feature it occupies: the meeple is actually placed on a monastery, but the model instead treats it as being on an incomplete road spanning multiple tiles.

It also detects gaps/voids in the current landscape but does not recognize that the candidate tiles can be legally placed into those openings. Instead, it incorrectly claims that placement is only possible along the border, indicating weak application of placement constraints and a failure to explore rotations and alternative fit configurations.

Finally, although the model recalls end-game scoring rules for incomplete features (e.g., monastery scoring), it does not integrate key state constraints when proposing actions. It assumes that matching the landscape and placing a meeple is sufficient, without checking whether the target feature is already occupied/contested by another meeple, leading it to propose options (e.g., Tile C) that cannot achieve the stated points under the actual board configuration.

Overall, the errors come from incomplete state grounding: the model retrieves relevant rules, but misreads feature types, overlooks legal placements, and fails to account for competing pieces, leading to confident but invalid recommendations.

### D.5.4 MODEL REASONING BREAKDOWN

In this Carcassonne case study, we observe that the model can detect salient pieces in the scene (e.g., the presence of a yellow meeple and the existence of interior voids), but fails to correctly ground those observations in the underlying feature structure and placement constraints. In particular, it misclassifies the feature the yellow meeple occupies, treating it as an incomplete road rather than a monastery, and does not reliably apply tile-placement rules to enumerate legal placements, including rotations that would allow tiles to fit into existing gaps.

This weakness is most apparent in the model's placement reasoning. Although it identifies empty spaces in the landscape, it incorrectly concludes that none of the candidate tiles can legally fit those

voids and therefore restricts placement to the outer perimeter. This indicates an incomplete search over valid placements and rotations, which prematurely eliminates feasible configurations and prevents the model from considering higher-scoring continuations that depend on filling interior gaps.

As a result, the model's end-game planning becomes suboptimal. While it gestures at end-of-game scoring, incomplete rule/constraint retrieval and early board-state misreads lead it to discard stronger scoring lines and settle on a valid but non-optimal placement. Overall, this case highlights a broader pattern where models can reference individual rules and components, but struggle to integrate them into a coherent multi-step plan when success requires accurate feature recognition, exhaustive legality checking (including rotations), and consistent tracking of ownership/constraints. A detailed breakdown of these failure modes is provided in Table 8.

| Reasoning Step | Reasoning Summary | Model Output Summary | Takeaways |
|---|---|---|---|
| <Understand> | Identify game state: current score, active bonuses, remaining dominos | Previous kingdom score: 36. Model assumes Harmony is achievable. Recognises Middle Kingdom is achievable. | Incorrect starting score (should be 17). Harmony is not possible with 2 dominos left. Correctly identifies that Middle Kingdom is still in play. |
| <Evaluate> | Compare scoring potential of each available domino (E–H) | Evaluates dominos A–H instead of limiting to draftable dominos E–H. No detailed per-domino simulation. | Ignores turn-order constraints. Fails to restrict options. Shows attempt at comparative evaluation, but lacks granularity. |
| <Analyze> | Assess best domino placement and score gain | Claims Domino G yields both Harmony (+5) and Middle Kingdom (+10). Correctly applies crown-multiplication scoring logic. Assumes cooperative drafting (no blocking). | Harmony is not achievable. Scoring formula is correctly applied. Appropriately assumes no interference from other players. |
| <Plan> | Simulate placement path to maximize total score | Breaks Domino G into two halves and places them in different locations. Predicts high score (57) from invalid placement. | Shows a fundamental misunderstanding of domino integrity (they must be placed as one unit). Overestimates final score. |
| <Reflect> | Strategic considerations: spatial balance, bonuses, and synergy | Focuses only on achieving both bonuses, ignores terrain synergy or true region expansion. | Overemphasis on bonus conditions. Misses opportunity to expand higher-value terrain (grassland). |
| <Conclude> | Final domino choice and score | Selects Domino G, Final Score: 57 | Final prediction is factually incorrect due to hallucinated bonuses and illegal placement logic. |

Table 4: Demonstration of reasoning process for the game environment: `KingDomino`. This process requires cross-modal reasoning to: 1) accurately represent the game state by combining graphical game state and rules knowledge, and 2) devise a valid and optimal solution. Red indicates unfactual/invalid outputs. Orange indicates incomplete/missing information. Green indicates valid and relevant reasoning. Existing methods struggle to produce valid action and planning logic to solve optimization problems.

| Reasoning Step | Reasoning Summary | Model Output Summary | Takeaways |
|---|---|---|---|
| \<Understand\> | Read the current victory point total from cards and on-component essences. | The player's current score comes from Fire Dragon: 1 VP and Sacred Grove: 2 base VP + 3 On-Component Life = 5 VP. Total: 6 VP. | Misreads the number of Life essences on Sacred Grove (should be 2), overestimating initial VP. |
| \<Evaluate\> | Assess resource availability for next actions. | To reach 4 Gold for a face-up monument, notes 2 Gold in pool, discards both cards in hand for +1 gold each. | Incorrectly reads Gold count; player already has 4. Discarding is unnecessary. |
| \<Analyze\> | Choose efficient scoring route. | Uses discard mechanic to suggest 2 VP monument purchase in 3rd action. | Valid application of discard rule. Reasoning is sound, though based on incorrect assumptions. |
| \<Plan\> | Lay out a concrete sequence of actions. | Proposes two discards followed by monument claim to reach 8 VP. | Sequence is logically structured and executable given the model's assumptions. |
| \<Reflect\> | Consider if higher-scoring paths exist. | Does not explore the Sacred Grove power + Coral Castle option. | Misses a more efficient 3-action plan that yields 10 VP. |
| \<Conclude\> | Calculate final VP after proposed actions. | Final score: 6 + 2 = 8. | Based on faulty VP estimate; true max is 10 if resource powers and discard are used strategically. |

Table 5: Demonstration of reasoning process for game environment: **Res Arcana**. This process requires cross-modal reasoning to 1. accurately represent the game state by combining the graphical game state *and* rules knowledge information, and 2. devise a valid and optimal solution to the given question. Red indicates unfactual/invalid outputs. Orange indicates incomplete/missing information. Green indicates valid and relevant reasoning. Existing methods struggle to produce valid action and planning logic to solve optimization problems.

| Reasoning Summary | Correct Reasoning | Model Prediction | Analysis | General Takeaways |
|---|---|---|---|---|
| Identify victory conditions + Game state | Should identify that all 4 victory conditions are in active state. Recognize that the player should gain victory in 2 actions, one reserved for declaring victory. Understand the game state and what cards each player has properly. | 1. **Analyze the Victory Conditions**: Imperial Victory is active, requiring the player to have at least 2 more King cards than each opponent. | Model correctly identifies Imperial Victory is active but fails to recognize other 3 victory conditions; ignores opponents' state; forgets to reserve one action to declare victory. | Models often fail at complete game state evaluation. |
| Check all possible victory paths + consider which is feasible | Evaluate all 4 victory conditions (Imperial, Holy, Renaissance, Globalization). Conclude that only Globalization is possible given the game state and constraints. | The blue player currently has King Edward IV in their Tableau, and no other Kings. To win, they need to acquire at least 2 more King cards. 4. Execute Actions: flip Papal States to a King (Vote); conquer Portugal (Campaign). 5. Declare Victory: with Papal States and Portugal, blue has 3 Kings total. | Model sticks with one path and doesn't consider alternatives. | Highlights lack of comprehensive evaluation |
| Optimal choice selection | For Globalization Victory, blue needs ≥2 more concessions than each opponent plus more Discover prestige; already has prestige edge; needs one more concession. | | Thinks 2 Kings is enough for "Imperial Victory" (actual rule: 2 more Kings than any other player); overlooks opponents' state. | Incomplete rule retrieval; fails to integrate opponent game state. |
| Action Planning | Choose the optimal path: West tableau ops → vote via *Cornish Tin* to flip kingdom to republic (King is on blue's card) placing a concession, then declare Globalization Victory. | 2. Identify Available Actions: - Vote on Tolfa Alum to flip the Papal States from Republic to King (adds a King). - Campaign on King Edward IV to conquer Portugal (adds another King). 3. Florin Management: blue has 3 Florins. Vote costs 1 Florin (to repress opponent's Concession in the Papal States Border). Campaign costs 1 Florin per Knight used. Blue has enough Florins for both. | Notices Papal States and regime change *(but misses that the card is on the throne)*. Notes enough Florins for "campaign" but ignores defenders=attackers, so outcome is uncertain. | Fails to integrate rule dependencies with current state; fails to check feasibility (rules vs state). |

Table 6: Demonstration of model reasoning over strategic game scenarios. This table outlines the reasoning steps required to solve a complex decision making task in a game setting of `Pax Renaissance 2e`. Reasoning Summary defines the goal at each stage. Correct Reasoning describes the ideal inference or action. Model Prediction shows the actual output. Analysis highlights deviation from expected behavior. General Takeaway reflects broader failures. Green indicates correct rule retrieval. Blue indicates correct game state analysis. Red indicates failure to apply game state and rules correctly. The table reveals that while models may retrieve relevant elements in isolation, they often fail to coordinate reasoning across steps to reach a valid optimal solution.

| Reasoning Summary | Correct Reasoning | Model Prediction | Analysis | General Takeaways |
|---|---|---|---|---|
| Identifying the game state | The model should accurately extract both the player's current resources and the public board state, including settlement locations and available harbor trades. | **1. Current Resources (Start of Turn):** Looking at the "Player's Table" image for the blue player (indicated by the blue-bordered development cards Year of Plenty and Road Building), we count the resource cards in hand: **Lumber (Wood):** 2 cards **Wool (Sheep):** 1 card **Grain (Wheat):** 1 card **Ore (Rock):** 2 cards **Brick:** 0 cards **Total:** 6 cards Thus, `prev_cards` (Lumber) = 2. | The model accurately identifies the player's resource cards and in-hand cards. It fails to detect the board-grounded settlement placements (blue settlements) and does not identify the adjacent harbor (and corresponding trade option) connected to those settlements. | The model can recover private, text-like information (cards in hand), but struggles with spatial grounding in the shared board state, which then undermines downstream reasoning about available actions and trades. |
| Turn start rule grounding and resource collection | The model should detect the blue settlement at the intersection adjacent to the 5–forest hex and correctly infer that it yields lumber on a roll of 5. | **2. Resource Production (Dice Roll = 5):** We examine the board to see if the blue player has any settlements or cities adjacent to a terrain hex marked with the number **5**. **Blue's Settlements:** One settlement is on the coast at the 3:1 harbor, adjacent to the **9-Forest**. The other settlement is inland at the intersection of **9-Forest**, **11-Mountain**, and **8-Hills**. **Hexes with Number 5:** **5-Forest:** Located in the inner ring. Its corners touch the Desert, 11-Mountain, 10-Pasture, 6-Hills, 11-Fields, and 5-Mountain. The blue settlement on the 11-Mountain is at the corner shared with 9-Forest and 8-Hills, which is **not** adjacent to the 5-Forest. **5-Mountain:** Located on the outer ring (bottom right). Blue has no pieces near this hex. **Result:** The blue player receives **0 resources** from the dice roll. | While the model applies the initial resource production rule correctly, it misses the relevant settlement placement and therefore fails to infer that the player should gain one additional resource. | The model can retrieve the relevant rule, but weak state grounding (missing settlement placement) causes an incorrect resource update and breaks downstream reasoning. |
| Check all possible resource gain paths + consider which ones are feasible | The model should enumerate all valid ways to obtain resource cards: player-to-player trading, 4:1 bank trade, harbor-based trade, and development cards (e.g., Year of Plenty), and correctly exclude player-to-player trading because it is disallowed by the constraint in the question. | **3. Trading and Development Cards:** **Trading:** The player is not allowed to trade with other players. They can trade with the bank. Blue has a settlement on a generic **3:1 Harbor** (visible on the 9-Forest coast). This allows trading 3 identical resources for 1 of any choice. Current hand: 2 Lumber, 1 Wool, 1 Grain, 2 Ore. Blue does not have 3 of any resource, so no bank trades are possible initially. **Development Cards:** Blue has **Year of Plenty** and **Road Building** face up (available to play). **Year of Plenty:** Allows the player to take any 2 resource cards from the bank. **Road Building:** Allows placing 2 roads (does not grant resources). | While the model correctly identifies the constraint in the question, it misidentifies the settlement's harbor trade condition and, as a result, wrongly discards that resource-gain path. | The model follows the stated constraints, but a misread of the board-dependent condition makes it discard a valid path. |
| Action Planning | The model should identify the full optimal sequence: make one harbor trade by giving away 2 ore to gain 1 lumber, then play *Year of Plenty* to gain 2 additional lumber. | **4. Maximizing Lumber:** To maximize the lumber count, the optimal move is to play the **Year of Plenty** card. **Action:** Play Year of Plenty and choose **2 Lumber** cards. **New Hand:** 4 Lumber, 1 Wool, 1 Grain, 2 Ore. | Because it fails to fully understand the board state and incorrectly removes valid options, the model converges on only a partially correct path. | The model's reasoning degrades when early state-reading errors prune valid options, leading it to settle on a suboptimal, partially correct solution. |

Table 7: Model reasoning breakdown for a turn-level resource maximization task in **Catan**. Reasoning Summary defines the goal at each stage. Correct Reasoning describes the ideal inference or action. Model Prediction shows the actual output. Analysis highlights deviation from expected behavior. General Takeaway reflects broader failures. Green indicates correct rule retrieval. Blue indicates correct game state analysis. Red indicates failure to apply game state and rules correctly. The table shows that small state-reading errors early on cascade into missed valid options, leading the model to a partially correct, suboptimal plan.

| Reasoning Summary | Correct Reasoning | Model Prediction | Analysis | General Takeaways |
|---|---|---|---|---|
| Identification of gamestate + possible placement of tiles | The model should correctly identify which features the yellow meeples occupy, and systematically evaluate all legal tile placements (including rotations) that satisfy the landscape placement rules. | 1. **Current situation** Yellow has a single meeple on an incomplete 3-tile road. There are two empty spaces in the interior, but none of the available tiles (A, B, C) legally fit those gaps. So Yellow must place their tiles on the outer edge of the landscape. Only Yellow's road and any new features Yellow creates will score for them. | While the model detects a yellow meeple on the landscape, it misclassifies the feature it occupies: treating it as an incomplete road rather than a monastery. In addition, it fails to explore valid tile placements, especially rotations that allow tiles to fit into existing void spaces. | The model notices key pieces, but poor feature recognition and incomplete search over legal placements (including tile rotations) lead to invalid downstream reasoning. |
| Action planning + End of Game Scoring | The model should enumerate all viable scoring paths to maximize end-game points: recognize that the incomplete monastery can be completed in one move, identify that a new monastery placed into an already-surrounded void would score immediately, and also consider the final-round advantage of placing a farmer to gain additional points at game end. | 2. **Best use of tile A (monastery with road)** Tile A can be placed so that: Its road does **not** connect to Yellow's existing road (otherwise we'd only add 1 more road tile and still leave it incomplete). The monastery is adjacent to **5 tiles** by the time the game ends (including those already there and those Yellow adds with their second tile). Yellow places a meeple as a **monk** on this monastery. Because the monastery is not yet fully surrounded, it does **not** score during the turn. At game end, an incomplete monastery scores 1 point for each surrounding tile plus the monastery tile itself: 5 adjacent tiles + 1 (the monastery tile) = **6 points**. 3. **Best use of tile B or C** Tile B or C can be placed so they complete a **small 2-tile city** (using one pre-existing city tile on the edge). Yellow places a meeple as a **knight** on their city segment. The city is completed immediately, so during the turn Yellow scores: 2 tiles × 2 points per tile = **4 points**. The knight is then returned to their supply and does not participate in end-game scoring. 4. **Yellow's existing road** The road with Yellow's meeple remains incomplete; it consists of 3 tiles. At game end, an incomplete road is worth 1 point per tile: 3 tiles = **3 points**. | The model finds a valid perimeter placement for Tile A, but it is suboptimal because it only pursues partial end-of-game points, driven by the mistaken belief that existing incomplete features cannot be completed. Although it retrieves the correct scoring rules for both incomplete and completed features, it fails to apply meeple-placement/ownership constraints: it claims Tile B or Tile C can complete a city for 4 points, but only Tile B achieves this, since Tile C would leave the city unclaimable by yellow due to an existing blue knight. | Although the model gestures at end-of-game scoring, incomplete rule/constraint retrieval and early misreads of the board state cause it to discard stronger options and settle on a valid but suboptimal placement. |

Table 8: Model reasoning breakdown for a two-turn end-game scoring task in **Carcassonne**. Reasoning Summary defines the goal at each stage. Correct Reasoning describes the ideal inference or action. Model Prediction shows the actual output. Analysis highlights deviation from expected behavior. General Takeaway reflects broader failures. Green indicates correct rule retrieval. Blue indicates correct game state analysis. Red indicates failure to apply game state and rules correctly. This table shows that early state and constraint misreads cascade into missed legal placements and ownership checks, yielding a valid but suboptimal end-game plan.

## E  CROSS-MODALITY PERFORMANCE PATTERNS

To better understand how different input modalities affect model performances, we conduct a comparative analysis across two multimodal models - GPT 4.1 (Azure), and Gemini 2.5 Pro. As shown in Figure 5, each model exhibits distinct performance patterns when exposed to None, Image, and Text rulebook formats. However, aggregate accuracy alone does not reveal the full reasoning dynamics behind these results. To provide a deeper and more faithful interpretation of these trends, we include one detailed example per model, illustrating how each modality shapes the model's reasoning, rule interpretation, and final output. These examples were selected to reflect a broader modality pattern we observed across the dataset, highlighting the model strengths and failure modes in a controlled setting.

For each example, we present a side-by-side reasoning comparison against the ideal output.

In each of the tables, the first column outlines the ideal reasoning path, describing how an optimal agent would combine game state understanding with correct rule retrieval to arrive at the solution. The next three columns present outputs generated by the model under different input modalities: None, Image, and Text. The final column provides an analysis comparing the performance of each modality in terms of interpretation of the state of the game, application of rules, and action planning.

Color coding used:

- Green: Correct rule retrieval and correct application to game state
- Blue: Correct game state understanding but incomplete or flawed rule application
- Red: Incorrect reasoning, hallucinated actions, or failure to integrate the state of the game with the rules

Findings from the modality comparison tables can be summarized as follows:

- **For GPT 4.1 Azure**: We observe that the text modality performs best, correctly integrating game state and the rules; image retrieves rules but misinterprets the game state; None performs the weakest, failing to ground either component effectively. For more in-depth analysis, see Fig. 18.
- **For Gemini 2.5 Pro**: The Gemini model's text modality consistently outperforms the others by accurately integrating game state and rules, while None performs moderately with partial rule and state recognition, and Image struggles the most, failing in both critical rule application and accurate game state interpretation. For more detailed analysis, see Fig. 19 and Fig. 20 *(NOTE: Due to the length of the model outputs, the complete comparison could not be accommodated in a single table and has therefore been split across two consecutive tables.)*

| Example 851: Pax Ren. 2e - Tier 3 question. | | | | |
|---|---|---|---|---|
| Is the prediction correct ? | ✗ | ✗ | ✅ | |
| Correct Reasoning | GPT41_Azure - None | GPT41_Azure - Image | GPT41_Azure - Text | Analysis |
| Observing the given game state, all 4 victory conditions are active. In order to win in 2 actions, the best possible victory that we can look at is Renaissance Victory (In order to gain this victory the blue player must have more republics than each opponent and 2 or more law prestige than each of the opponents) . This is because the blue player already has 2 more law prestiges than the opponent (who has none). All that the blue player needs to do is have one republic, that makes the player achieve all the conditions for Renaissance Victory (Note that the other player has no Republics yet, so gaining just one republic would satisfy the condition).

This can be done in the following manner:
1: Play the West Tableau Ops, perform the vote operation on the Tolfa Alum card (Papal States). Since there are no repressed units it would not cost any florins, and also since there is only one concession on Papal states border, it being blue makes the blue player win the vote. This results in a Regime change and blue player gets to put one more concession on the map, anywhere on the border of Papal states. This would result in flipping the kingdom card to a Republic making blue player have maximum number of republics compared to other players. This gives the player a Renaissance Victory. Now due to the placement of the new concession, The blue player would also have 2 more concessions than the other player and also more Discover prestige than the other players, making blue player even go for Globalization Victory.
2: Declare Renaissance Victory or Globalization Victory.
This way the blue player can win this game in 2 actions. | Let's analyze the board and the blue player's position:

- The blue player (Jacob Fugger) has 3 florins, 9 cubes, and the Fugger Bank of Augsburg.
- Their tableau includes: Cornish Tin, Consent of the Governed, Pope Julius II, Tolfa Alum, and Almeida Armada.
- The current victory cards available are: Renaissance, Globalization, Imperial, and Holy.
- The Renaissance Victory requires: 2+ more Republics than Theocracies, and 2+ more Concessions than any other player.

Let's check the requirements for Renaissance Victory:
- Blue has 2 Republics (Cornish Tin, Consent of the Governed) and 0 Theocracies.
- Purple has 0 Republics and 0 Theocracies.
- Blue has 3 Concessions (France, Papal States, Portugal).
- Purple has 2 Concessions (Holy Roman Empire, Byzantium).

So, blue already meets the Renaissance Victory requirements.

Now, let's check if blue can trigger a victory check:
- Blue can use "West Tableau Ops" to use Tolfa Alum's "Vote" operation to trigger a victory check.

Therefore, the two actions are:
1. Use "West Tableau Ops" with Tolfa Alum to trigger "Vote" (Victory Check).
2. Declare "Renaissance Victory". | Let's analyze the current board state and the blue player's position:

### Victory Conditions
- **Renaissance Victory**: Requires (1) more Republics than each opponent and (2) at least two more Law Prestige than each opponent.
- The Renaissance Victory card is active.
- Blue (Fugger) has a Republic (Consent of the Governed in tableau) and at least two Law Prestige (Consent of the Governed, Cornish Tin).
- No other player has a Republic in their tableau (as seen in the player boards).
- Blue has 3 Florins, enough to take two actions.

### Step-by-step Solution

#### Step 1: Declare Victory
- Blue already meets the conditions for Renaissance Victory.
- The rules allow a player to use an action to declare victory if the conditions are met.

#### Step 2: No other action is needed
- Since the victory can be declared as the first action, only one action is required.

**Note:** If the question strictly requires two actions, blue could take any legal action first (such as a West Tableau Ops or Trade Fair), then declare victory as the second action. But as per the current state, blue can win immediately in one action. | Let's analyze the current board state and the blue player's position:

### Blue Player's Tableau (Jacob Fugger)
- Cards: Cornish Tin, Consent of the Governed, Pope Julius II (King of Papal States), Tolfa Alum, Almeida Armada
- Florins: 3
- Pawns: 10 (all available)
- Papal States: Controlled as a King (not a Republic)
- Victory Squares: Renaissance, Globalization, Imperial, Holy (all visible)
- To win with Renaissance Victory: "Have more Republics than each opponent AND at least 2 more Law Prestige than each opponent."

#### Step 1: Can Blue Win Renaissance Victory?
- Blue currently controls 0 Republics (Papal States is a King).
- To win, Blue must have more Republics than the opponent (Purple has 0 Republics, so 1 is enough).
- Blue must have at least 2 more Law Prestige than the opponent.
- Blue has: Cornish Tin (Law), Consent of the Governed (Law), Tolfa Alum (Law) = 3 Law Prestige.
- Purple has: None visible in tableau.
- So, if Blue flips Papal States to a Republic, Blue will have 1 Republic and 3 Law Prestige, which is enough.

#### Step 2: How to Flip Papal States to a Republic?
- Use a Strawman Vote Op (Vote against your own King) to flip Papal States to a Republic.
- Tolfa Alum has a Vote Op (West, Papal States).
- Cost: Pay 1 Florin per Repressed Token on Papal States (none visible), so cost is 0.
- Blue has 3 Florins, so can pay.

#### Step 3: Action Sequence
- Action 1: Use Tolfa Alum's Vote Op to flip Papal States to a Republic (Strawman Regime Change).
- Action 2: Use Victory action to declare Renaissance Victory.

#### Step 4: Florin Change
- Vote costs 0 Florins (no Repressed Tokens).
- Victory action costs 0 Florins. | The initial game state is partially identified by the *none* and *image* modalities, and fully, accurately identified by the *text* modality.

As expected, the *none* modality fails to retrieve relevant rules, resulting in weak or incoherent reasoning.

The *image* modality shows a better rule retrieval, but struggles to correctly interpret the game state, leading to incorrect final plan.

In contrast, the *text* modality demonstrates strong understanding of both rules and states, applying each of them appropriately to reach the correct conclusion. While it exhibits a minor miscalculation, it however does not affect the final answer. |

Figure 18: Modality-wise reasoning comparison (GPT4.1 Azure).

| Example 548 res arc tier 3 question. | | | | |
|---|---|---|---|---|
| **Is the prediction correct ?** | ✗ | ✗ | ✓ | |
| **Correct Reasoning** | **GeminiPro2-5 - None** | **GeminiPro2-5 - Image** | **GeminiPro2-5 - Text** | **Analysis** |
| In the given situation, The amount of victory points present are 2 from sacrificial pill and 2 (1x2 calm) from sunken reef. So there are a total of 4 victory points present.

We can add vp either by using action on the mermaid card and placing calm onto sunken reef, or by using the second action on Sacrificial pil and gaining gold essences to buy a monument (2 actions), or by gaining 3 life essences and placing one death essence onto pil using first action (3 actions).

Of these, in one action only one possibility can be done in order to get more vp. 1: Perform action on mermaid card by using one calm essence and place the calm essence onto the sunken reef (adds 1 vp).
As a result the final new vp: 4+1 = 5. | The player's current total Victory Points (VP) are:
* Sunken Reef: 1 VP
* Sacrificial Pit: 2 VP (base value, as it has 0 Death essences on it currently)
* Total current VP = 1 + 2 = 3 VP.

The player has the following essences in their general pool: 6 Fire, 1 Calm, 1 Death.

Let's evaluate the VP gain from possible single actions:

1. **Activate Sacrificial Pit's ability:**
* Ability: Pay 3 of any essence + 1 Death to gain 1 Death essence on Sacrificial Pit.
* Cost: The player has 1 Death and 6 Fire + 1 Calm = 7 "any" essences. They can pay this cost (e.g., 3 Fire + 1 Death).
* Effect: 1 Death essence is placed on Sacrificial Pit.
* VP change: Sacrificial Pit's VP is 2 (base) + 1 VP per Death essence. It goes from 2 VP to 2 + 1 = 3 VP.
* Net VP gain from this action = +1 VP.
* New total VP = 3 (initial) + 1 = 4 VP.

2. **Play Elemental Spring from hand:**
* Cost: 2 Fire, 2 Life, 2 Calm.
* VP value of Elemental Spring: 1 VP.
* Resource generation:
* Player has 6 Fire, 1 Calm, 1 Death. Needs 2 Life and 1 more Calm.
* Gain 2 Life: Use Druid's ability twice (Pay 1 Fire -> 1 Life, twice). Spends 2 Fire. Player has 4 Fire, 1 Calm, 1 Death, 2 Life.
* Gain 1 Calm: Use Mermaid's ability once (Pay 1 Fire -> 1 Calm). Spends 1 Fire. Player has 3 Fire, 2 Calm, 1 Death, 2 Life.
* Player now has enough resources (3F, 2C, 2L, 1D) to pay the cost (2F, 2L, 2C).
* Action: Play Elemental Spring.
* Net VP gain from this action = +1 VP (from Elemental Spring itself).
* New total VP = 3 (initial) + 1 = 4 VP. | The player's current Victory Points (VP) are from their Places of Power in play:
1. Sunken Reef: 1 VP
2. Sacrificial Pit: 1 VP
Total current VP = 1 + 1 = 2 VP. This is the "Previous amount".

We need to find the single action that provides the maximum guaranteed net VP gain. All other players have passed, so the current player will take one action and then pass. They will not be the first to pass, so they won't get the First Player token (1 VP).

Let's evaluate possible actions:

1. **Place an artifact from hand:**
* **Magical Shard:** Cost 0. VP value of card: 0. VP gained from this action: 0.
* **Elemental Spring:** Cost 2 of any essence (no Gold/Death). Player has 4 Elan, 3 Life, 2 Calm, 1 Death, so can afford it (e.g., pay 2 Elan). VP value of card: 0. VP gained from this action: 0.

2. **Claim a monument or Place of Power:**
* **Temple (Monument):** Cost 4 Gold. VP: 2. Player has 0 Gold. Cannot afford.
* **Solomon's Mine (Monument):** Cost 4 Gold. VP: 1. Player has 0 Gold. Cannot afford.
* No other Places of Power are available to claim from the center.

3. **Discard a card for 1 Gold or any 2 other essences:**
* Discarding Magical Shard or Elemental Spring would yield resources, not VP. VP gained from this action: 0. | The player's current Victory Points (VP) need to be calculated first. Player's components contributing to VP:
1. **Sunken Reef (Place of Power):**
* VP: "1 per Calm on it".
* There are 2 Calm essences on Sunken Reef.
* VP from Sunken Reef = 2 VP.
2. **Sacrificial Pit (Place of Power):**
* VP: "1 per Death on it + 2".
* There are 0 Death essences on Sacrificial Pit.
* VP from Sacrificial Pit = 0 + 2 = 2 VP.

Total current VP (Previous amount) = 2 (from Sunken Reef) + 2 (from Sacrificial Pit) = 4 VP.

Now, let's analyze the possible actions the player can take and their impact on VP:

**Player's Essences:** 6 Elan, 1 Calm, 1 Death.
**Player's Hand:** Magical Shard, Elemental Spring.
**Player's Components in Play (all straightened):** Druid (Mage), Reanimate (Magic Item), Mermaid (Artifact), Sunken Reef, Sacrificial Pit.

Possible Actions:

1. **Place an artifact:**
* **Magical Shard:** Cost 0. No inherent VP. Placing it does not change VP.
* **Elemental Spring:** Cost 2 Elan, 1 Life, 1 Calm. Player has 6 Elan, 1 Calm, but 0 Life. Cannot afford.

2. **Claim a monument or Place of Power:**
* **Temple (Monument):** Cost 4 Gold. VP: 2. Player has 0 Gold. Cannot afford.
* **Solomon's Mine (Monument):** Cost 4 Gold. VP: 1. Player has 0 Gold. Cannot afford.

3. **Discard a card for 1 Gold or 2 other essences:**
* This action provides essences, not VP. No change in VP. | The *None* modality partially retrieves the game state but incorrectly estimates the victory points.

The *image* modality performs similarly in terms of incomplete state recognition, but further compounds the issue by severely miscalculating victory points and failing to interpret the current game configuration.

In contrast the *text* modality demonstrates the strongest understanding: it correctly identifies the victory points and accurately interprets the game state with only a minor oversight

In terms of rule retrieval, the *image* modality retrieves only basic mechanics (eg. placement cost, passing) and fails to identify or correctly interpret rules associated with key cards in play.

Surprisingly the *None* modality does a better job in this regard, it identifies correct paths and relevant rules for important cards, though it misses the crucial constraint that only one action is permitted in the question.

The *text* modality performs the best: it understands the major cards more effectively than none. While it makes a few errors, which do not seem to affect this question, It crucially respects the one-action constraint and offers more coherent, strategically grounded reasoning. |

Figure 19: Modality-wise reasoning comparison, Part 1 (Gemini 2.5 Pro).

| Example 548 Res Arcana - Tier 3 question. | | | | |
|---|---|---|---|---|
| Is the prediction correct ? | ✗ | ✗ | ✓ | |
| Correct Reasoning | GeminiPro2-5 - None | GeminiPro2-5 - Image | GeminiPro2-5 - Text | Analysis |
| | 4. **Claim Solomon's Mine from Shared Area:**
* Cost: 4 Gold.
* VP value of Solomon's Mine: 1 VP.
* Resource generation: Same as for Temple, player can get 4 Gold.
* Action: Claim Solomon's Mine.
* Net VP gain from this action = +1 VP.
* New total VP = 3 (initial) + 1 = 4 VP.

Comparing the net VP gains from a single action:
* Activate Sacrificial Pit: +1 VP
* Play Elemental Spring: +1 VP
* Claim Temple: +2 VP
* Claim Solomon's Mine: +1 VP

The maximum guaranteed net VP gain from one action is +2 VP, achieved by claiming Temple.
The player's total VP before this action is 3.
The player's total VP after this action (and necessary setup actions for gold) will be 3 + 2 = 5.

Previous amount (total VP before the action): 3
New amount (total VP after the action): 5

Final Answer:
Previous amount: 3, New amount: 5 | 4. **Use a power on one of your straightened components:**
* **Druid (Mage):** Power: Turn -> Straighten one Creature artifact. Player has no Creature artifacts in play. VP gained: 0.
* **Sunken Reef (Place of Power, straightened):**
* Power 1: Turn -> Gain 1 Elan and 1 Life. VP gained: 0.
* Power 2: Turn -> Gain 1 Gold. VP gained: 0.
* **Sacrificial Pit (Place of Power, straightened):**
* Power 1: Turn, Destroy one of your Creatures -> Gain 2 of any essence (no Gold/Death) AND 2 Gold. Player has no Creature artifacts to destroy. Cannot use. VP gained: 0.
* Power 2: Turn -> Pay 2 Elan, 1 Life -> Gain 2 VP.
* Player has 4 Elan and 3 Life. Can afford the cost.
* Essences after payment: 4-2=2 Elan, 3-1=2 Life, 2 Calm, 1 Death.
* VP gained from this action: +2 VP.

5. **Pass:**
* If the player chooses to pass as their action, they are not the first to pass, so they do not gain the 1 VP from the First Player token. VP gained from this action: 0.

Comparing the VP gained from each possible action:
* Placing Magical Shard: 0 VP
* Placing Elemental Spring: 0 VP
* Claiming a Monument: Cannot afford.
* Discarding a card: 0 VP
* Using Druid power: 0 VP
* Using Sunken Reef power (either): 0 VP
* Using Sacrificial Pit power 1: Cannot use.
* Using Sacrificial Pit power 2: +2 VP
* Passing: 0 VP

The maximum guaranteed net VP gain from a single action is 2 VP, by using the second power of the Sacrificial Pit.

The player's total VP before this action is 2.
After performing the action (Use Sacrificial Pit power 2), the player gains 2 VP.
The player's new total VP will be 2 (previous) + 2 (gained) = 4 VP.

So, the "Previous amount" is 2, and the "New amount" is 4.

Final Answer:
Previous amount: 2
New amount: 4 | 4. **Use a power on a straightened component:**
* **Druid (Mage):** Power: Turn + 1 Life -> Straighten a Creature artifact. Player has 0 Life. Cannot use.
* **Reanimate (Magic Item):** Power: Turn + 1 Death -> Gain 1 Life. Player has 1 Death. Can use. Gains Life, no VP change.
* **Mermaid (Artifact - Creature):** Power: Turn + 1 Calm/Life/Gold from pool -> Put the essence spent on one of their components.
* Player has 1 Calm. Can spend 1 Calm.
* Cost: Turn Mermaid, 1 Calm.
* Effect: Put 1 Calm on a component. If placed on Sunken Reef, Sunken Reef's VP increases by 1 (from 2 to 3).
* Net VP gain = +1. New total VP = 4 + 1 = 5.
* **Sunken Reef (Place of Power):**
* Power 1: Turn + 2 Life + 2 Elan -> Put 1 Calm on Sunken Reef. Player has 0 Life. Cannot afford.
* Power 2: 1 Gold -> Put 1 Calm on Sunken Reef. Player has 0 Gold. Cannot afford.
* **Sacrificial Pit (Place of Power):**
* Power 1: Turn + 3 Elan -> Put 1 Death on Sacrificial Pit. Player has 6 Elan. Can use.
* Cost: Turn Sacrificial Pit, 3 Elan.
* Effect: Put 1 Death on Sacrificial Pit. Sacrificial Pit's VP increases by 1 (from 2 to 3, as it becomes 1 Death + 2 base).
* Net VP gain = +1. New total VP = 4 + 1 = 5.
* Power 2: Turn + Destroy one of your Dragons or Creatures -> Gain (Cost of destroyed artifact) in any essence + 1 Gold. Player has Mermaid (Creature). This gains essences, not VP directly. Destroying Mermaid doesn't lose VP as Mermaid itself has no VP.
* Power 3: 1 Death -> Put 1 Death on Sacrificial Pit. Player has 1 Death. Can use.
* Cost: 1 Death.
* Effect: Put 1 Death on Sacrificial Pit. Sacrificial Pit's VP increases by 1 (from 2 to 3).
* Net VP gain = +1. New total VP = 4 + 1 = 5.

5. **Pass:**
* This action ends the player's turn for the round. It does not directly grant VP. Since other players have already passed, the player would not become the First Player by passing now.

Comparing the actions that grant VP:
* Using Mermaid's power to put 1 Calm on Sunken Reef: Net VP gain = 1. Total VP = 5.
* Using Sacrificial Pit's Power 1 (Turn, 3 Elan): Net VP gain = 1. Total VP = 5.
* Using Sacrificial Pit's Power 3 (1 Death): Net VP gain = 1. Total VP = 5.

The maximum guaranteed net number of VP the player can gain in one action is 1.
Previous amount of VP = 4.
New amount of VP = 4 + 1 = 5.

Final Answer:
Previous amount: 4
New amount: 5 | |

Figure 20: Modality-wise reasoning comparison, Part 2 (Gemini 2.5 Pro).

## F    HALLUCINATION PATTERNS AND POSSIBLE CAUSES

In this section, we examine qualitative instances where models generate hallucinated outputs – i.e. confident but factually incorrect statements that deviate from both the rule book and the game state. By analyzing such failures we aim to better understand the kinds of inductive error language models make when integrating visual and textual context.

We identify several frequently observed hallucination patterns and hypothesize potential triggers underlying these errors, highlighting deeper challenges in grounded multimodal reasoning. These findings are summarized in Figures 21, 22, 23, where each highlighted region (with number) corresponds to a numbered item in the adjacent column. This column provides possible explanations for the hallucinations, along with excerpts from the model outputs or images that illustrate the observed behavior.

Note: We also evaluate QWEN2.5-VL (32B) to assess performance on a strong open-source vision-language model. Due to space constraints, we do not include its full quantitative results in the main paper. In this section, we include more Tier-3 examples from QWEN2.5-VL primarily because its outputs are more manageable in length for qualitative illustration; importantly, the hallucination trends discussed here are consistent across the models we evaluate.

| Model output | Possible reason for hallucination |
|---|---|
| **Model: Qwen2.5-VL-32B-Instruct (txt)**
**Question ID: 857** | |

To determine how the blue player can win the game in 2 actions, let's analyze the game state and the victory conditions for each type of victory. We'll focus on the blue player's resources, tableau, and available actions.

### Step 1: Analyze the Blue Player's Current State
- **Florins**: The blue player has 5 florins.
- **Tableau**: The blue player has the following cards in their tableau:
- **Pope Julius II**: An Empire card for the Papal States.
- **Sultan Mehmed II the Conqueror**: An Empire card for the Ottoman Empire.
- **Qa'it Bay of the Burji Dynasty**: An Empire card for the Mamluk Empire. [1]
- **Victory Conditions**:
- **Imperial Victory**: Requires 2 or more King Cards (not Republics) than each opponent.
- **Globalization Victory**: Requires at least 2 more Concessions than each opponent and more Discovery Prestige than each opponent.
- **Renaissance Victory**: Requires more Republics than each opponent and at least 2 more Law Prestige than each opponent.
- **Holy Victory**: Requires more Prestige in the supreme religion than each opponent, more Bishops of its color, and more Tokens of its color in Theocracies than both other religions combined.

### Step 2: Evaluate Possible Actions
The blue player has the following actions available:
1. **Purchase**: Buy a card from the market.
2. **Play**: Play a card from their hand into their tableau.
3. **Sell**: Sell a card from their hand or tableau.
4. **West Tableau Ops**: Activate operations from the West tableau.
5. **East Tableau Ops**: Activate operations from the East tableau.
6. **West Trade Fair**: Convene a West trade fair.
7. **East Trade Fair**: Convene an East trade fair.
8. **Victory**: Declare victory if the conditions for a victory are met.

---

1. The cards that have been given out by the model are not present in the player's game state. The possible explanation:
- In the future logic, the model tries to get 2 kings for getting the Imperial Victory. Due to the misinterpretation that Imperial victory needs 2 king cards, the model states that it would perform a "Vote" operation, which as shown later, shown with the Papal States example (Pope Julius II), therefore assuming card to be present.
- So, trying to fit this into the logic model assumes that there are three king cards: Papal States, Ottoman, and Mamluk.
- The possible explanation for these models to select Ottoman, and Mamluk is because of the order in which they are present in the rule book.
- The following are a few screenshots from the rule book where the three kingdoms are generally often mentioned in various examples (either all 3 together, or any 2 of them paired) which might have been the reason for the model to take these 3 names.

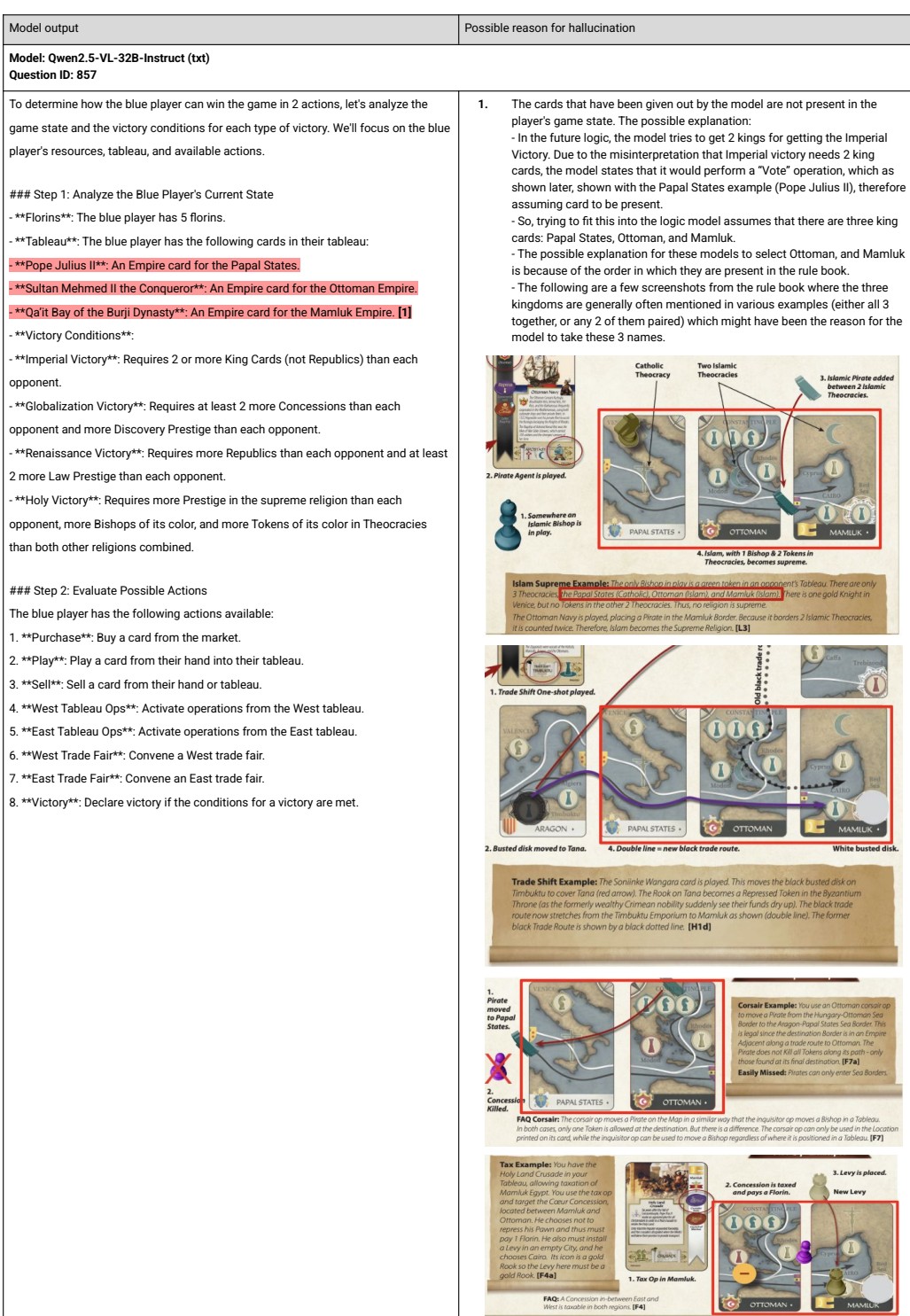

Figure 21: Hallucination and Possible Reasoning Part 1.

| Model output | Possible reason for hallucination |
|---|---|
| **Model: Qwen2.5-VL-32B-Instruct (txt)**
**Question ID: 857** | |

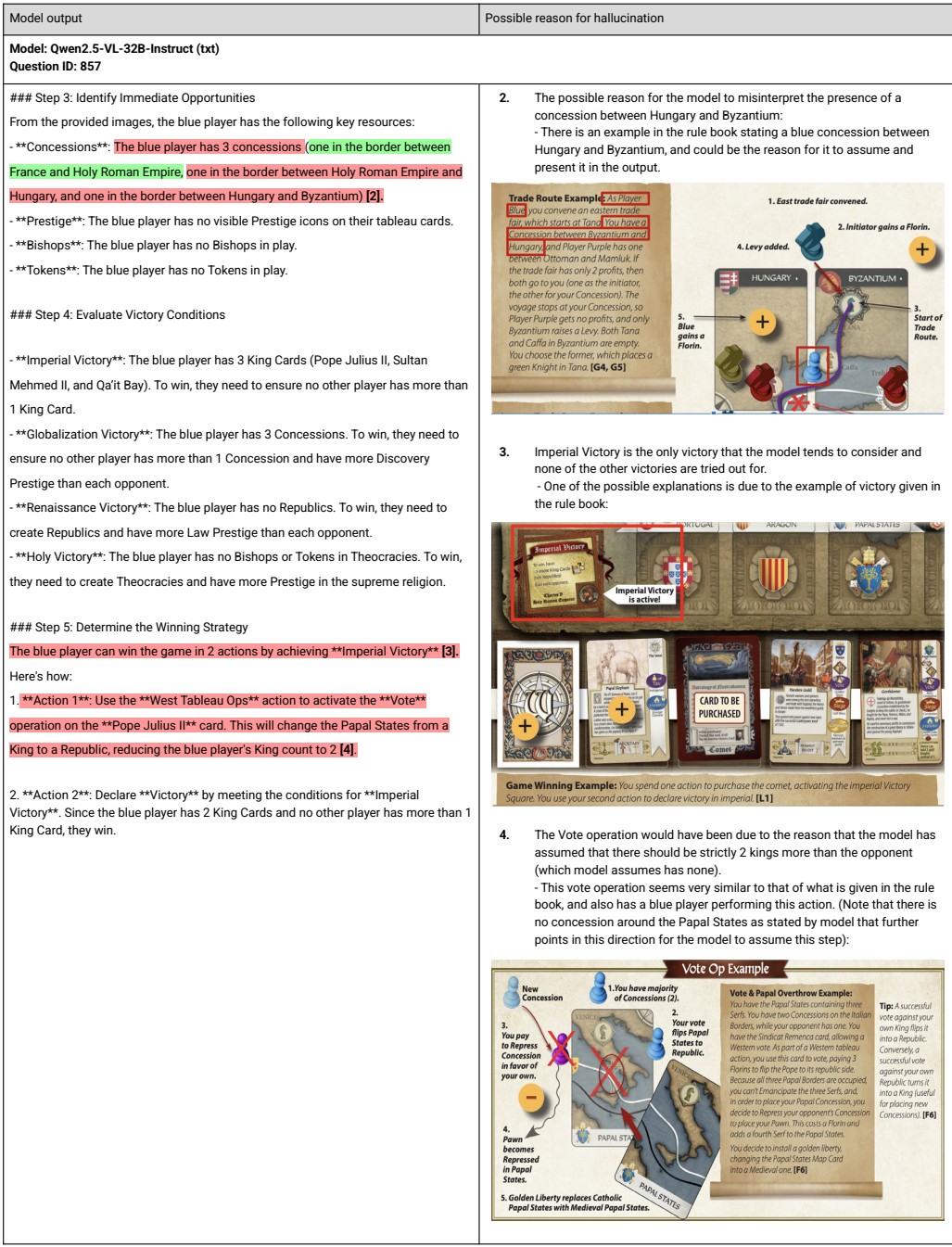

### Step 3: Identify Immediate Opportunities

From the provided images, the blue player has the following key resources:

- **Concessions**: The blue player has 3 concessions (one in the border between France and Holy Roman Empire, one in the border between Holy Roman Empire and Hungary, and one in the border between Hungary and Byzantium) [2].
- **Prestige**: The blue player has no visible Prestige icons on their tableau cards.
- **Bishops**: The blue player has no Bishops in play.
- **Tokens**: The blue player has no Tokens in play.

### Step 4: Evaluate Victory Conditions

- **Imperial Victory**: The blue player has 3 King Cards (Pope Julius II, Sultan Mehmed II, and Qa'it Bay). To win, they need to ensure no other player has more than 1 King Card.
- **Globalization Victory**: The blue player has 3 Concessions. To win, they need to ensure no other player has more than 1 Concession and have more Discovery Prestige than each opponent.
- **Renaissance Victory**: The blue player has no Republics. To win, they need to create Republics and have more Law Prestige than each opponent.
- **Holy Victory**: The blue player has no Bishops or Tokens in Theocracies. To win, they need to create Theocracies and have more Prestige in the supreme religion.

### Step 5: Determine the Winning Strategy

The blue player can win the game in 2 actions by achieving **Imperial Victory** [3].
Here's how:

1. **Action 1**: Use the **West Tableau Ops** action to activate the **Vote** operation on the **Pope Julius II** card. This will change the Papal States from a King to a Republic, reducing the blue player's King count to 2 [4].

2. **Action 2**: Declare **Victory** by meeting the conditions for **Imperial Victory**. Since the blue player has 2 King Cards and no other player has more than 1 King Card, they win.

---

2. The possible reason for the model to misinterpret the presence of a concession between Hungary and Byzantium:
   - There is an example in the rule book stating a blue concession between Hungary and Byzantium, and could be the reason for it to assume and present it in the output.

3. Imperial Victory is the only victory that the model tends to consider and none of the other victories are tried out for.
   - One of the possible explanations is due to the example of victory given in the rule book:

4. The Vote operation would have been due to the reason that the model has assumed that there should be strictly 2 kings more than the opponent (which model assumes has none).
   - This vote operation seems very similar to that of what is given in the rule book, and also has a blue player performing this action. (Note that there is no concession around the Papal States as stated by model that further points in this direction for the model to assume this step):

Figure 22: Hallucination and Possible Reasoning Part 2.

| Model output | Possible reason for hallucination |
|---|---|
| **Model: GPT41_Azure (txt)**

**Question ID: 857** | |
| Let's analyze the current board state and the blue player's position:
### Blue Player's Tableau (Jacob Fugger)
- King Edward IV (England, King)
- Cornish Tin (Vote, Commerce, Peasant Revolt)
- Tolfa Alum (Vote, Commerce)
- Almeida Armada (Repress, Commerce, Corsair, Discovery Prestige)
- 3 Florins

### Victory Squares
- Renaissance Victory: More Republics than each opponent AND at least 2 more Law Prestige than each opponent.
- Globalization Victory: At least 2 more Concessions than each opponent AND more Discovery Prestige than each opponent.
- Imperial Victory: At least 2 more King cards than each opponent (3 in 2p).
- Holy Victory: More Prestige in the Supreme Religion than each opponent.

### Board State
- Blue has 1 King (England), no Republics.
- Purple has 1 King (HRE), no Republics.
- No Republics on the board.
- Blue has 3 Concessions (Lisbon, Bordeaux, London).
- Purple has 1 Concession (Alexandria).
- Blue has 1 Discovery Prestige (Almeida Armada).
- Purple has 0 Discovery Prestige.
- Law Prestige: Blue has 0, Purple has 0.

#### Analysis

- Blue cannot win Renaissance Victory (no Republics, no Law Prestige).
- Blue cannot win Holy Victory (no religious supremacy).
- Blue cannot win Imperial Victory (only 1 King, needs 3 more than opponent in 2p).
- Blue CAN win Globalization Victory:
- Blue has 3 Concessions, Purple has 1 → Blue has 2 more.
- Blue has 1 Discovery Prestige, Purple has 0 → Blue has more.

#### What is needed?
- Blue must declare Globalization Victory.

#### Can Blue do this in 2 actions?
Yes, if Globalization Victory is already active. But in the image, the only active Victory Square is Imperial Victory (red side up). Blue must first activate Globalization Victory by purchasing a Comet card.**[5]**

#### Where are the Comet cards?
- In the West Market, the leftmost facedown card is a Comet (visible by the unique back).
- Blue has 3 Florins, and the leftmost card costs 1 Florin.

#### Step-by-step plan:
1. **Purchase** the leftmost West Market card (Comet) for 1 Florin. This immediately activates a Victory Square of your choice. Blue chooses Globalization Victory.
2. **Victory**: Declare Globalization Victory (Blue has 2 more Concessions and more Discovery Prestige than Purple).

#### Florin change:
- Action 1: -1 Florin (purchase Comet)
- Action 2: 0 Florin (declare victory) | 5. Even though the entire thought process of the model finding the solution is completely to the point, The model however considers that Globalization victory is not active and the only active victory is Imperial victory, and says the model needs to purchase a comet card for activation of another victory.
- This might be due to the rulebook stating the only active victory is Imperial victory.

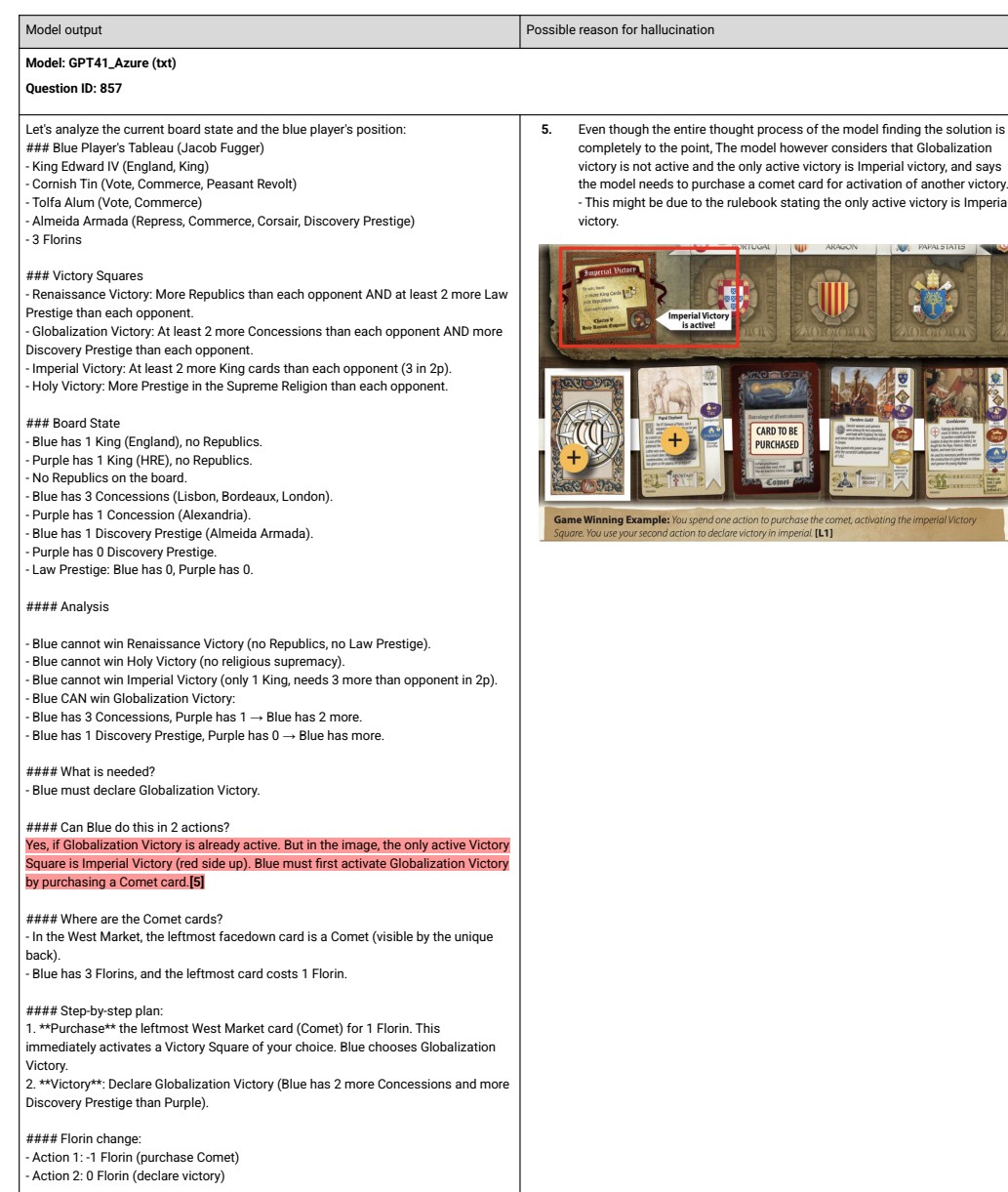 |

Figure 23: Hallucination and Possible Reasoning Part 3.

## G CLAUDE OUTPUT ANALYSIS

We run experiments with both Claude 3.7 Sonnet and Claude 4.5 Sonnet. Due to space constraints, we omit the full Claude 3.7 results from the main paper; however, we observe that its modality- and tier-wise trends, as well as its qualitative solution patterns, closely mirror those of Claude 4.5 Sonnet.

Evaluating on Claude's Sonnet models, we find demonstrates consistent gains across modalities, with highest performance observed when provided with text formatted rule book input, followed closely by image based input. The model performs least effectively in the absence of explicit rulebook context (None modality), highlighting its resilience for external knowledge sources while reasoning. When broken down by the game and tier, Claude achieves decent tier 1 results across all games, particularly excelling in Res Arcana with text and image input. Tier 2 performance, while lower, still shows reasonable comprehension of rule based tasks. As expected, Tier 3 drops significantly across the board, reflecting the inherent complexity of short-horizon optimization and state reasoning.

### G.1 NOTEWORTHY BEHAVIORAL OBSERVATIONS

Beyond its quantitative performance, Claude's Sonnet models demonstrate several distinct behavioral patterns that surface through qualitative analysis of its responses across all five games and task tiers:

- **Explicit Acknowledgment of Uncertainty**: Claude frequently acknowledges when it is unable to parse or interpret aspects of input. In some cases, even when relevant visual information is clearly visible (e.g. opponents game state), the model notes its uncertainty or inability to "see" details, demonstrating a degree of calibrated self awareness. This is particularly noticeable in games like KingDomino where Claude often fails to commit to a final answer, explicitly stating confusion over victory conditions.

- **Strong Rule Retrieval**: Claude excels at retrieving relevant rulebook content, often more reliable than most models tested. It consistently references correct actions, turn/ round structures, and gameplay constraints in both tier 2 and tier 3 scenarios. The strength is evident even when the overall answer is incorrect, suggesting Claude understands a foundational logic even if the execution falters

- **Rulebook Based Example Referencing**: When uncertain Claude tends to stimulate the rulebook scenarios or example-based reasoning to justify its answers. Instead of directly answering based on observed state, it uses generalized examples or templates from the rulebook to simulate plausible scenarios. This reflects the model's fallback strategy when exact state resolution is not possible.

- **Thoughtful Educated Guesses**: In most cases, Claude offers answers that resemble well-reasoned guesses - clearly marked with assumptions and an explanation of its thought process. These responses reflect the model's attempt to reason through the situation using partial information, while still being transparent about limitations and uncertainty.

- **Inability to Simulate Intermediate States**: A key limitation observed across all games is Claude's difficulty in reasoning over evolving game states, particularly in multi-step scenarios. In Tier 3 optimization tasks, the model often fails to simulate the intermediate outcome of one action before evaluating the next, eventing it from forming a complete multi-action strategy

- **Occasional Unjustified Factual Errors**: While Claude's reasoning is most of the time explainable, there are instances where it outputs incorrect calculations (e.g., costs for a move) without any justification. These inconsistencies break from its otherwise transparent style and may point to retrieval or grounding failures.

### G.2 EDUCATED GUESSES DEMONSTRATION

Among the models we tested, Claude consistently produces some of the best knowledgeable guesses, especially in situations where full state understanding is difficult. While the model does not always

arrive at the correct final answer, it frequently demonstrates thoughtful reasoning, combining partial information with rulebook knowledge to make plausible approximations.

In the previous section, we outlined several key patterns in Claude's behavior, such as **explicit uncertainty acknowledgment, strong rule retrieval, example-based reasoning, and transparent educated guesses**.

In the following examples (Figures 24, 25, 26), we provide concrete demonstrations of these behaviors. Each table contains highlighted annotations to help visualize the model's reasoning patterns:

- Green: indicates correct rule retrieval and application.
- Blue: highlights correct game state interpretation.
- Red: marks failures to integrate game state with rules or hallucinated logic.
- Yellow: represents heuristic-driven educated guesses where Claude chooses a plausible path despite uncertainty.

These examples illustrate how Claude often balances between rule-based reasoning and heuristic guesswork, resulting in responses that are more interpretable and calibrated compared to other models. Additionally, they surface limitations such as the inability to simulate intermediate states or occasional factual errors.

*Note: All example demonstrations shown in the figures use Claude 3.7 Sonnet purely for presentation purposes, since its reasoning traces are more manageable in length; importantly, Claude 3.7 and Claude 4.5 exhibit similar reasoning patterns.*

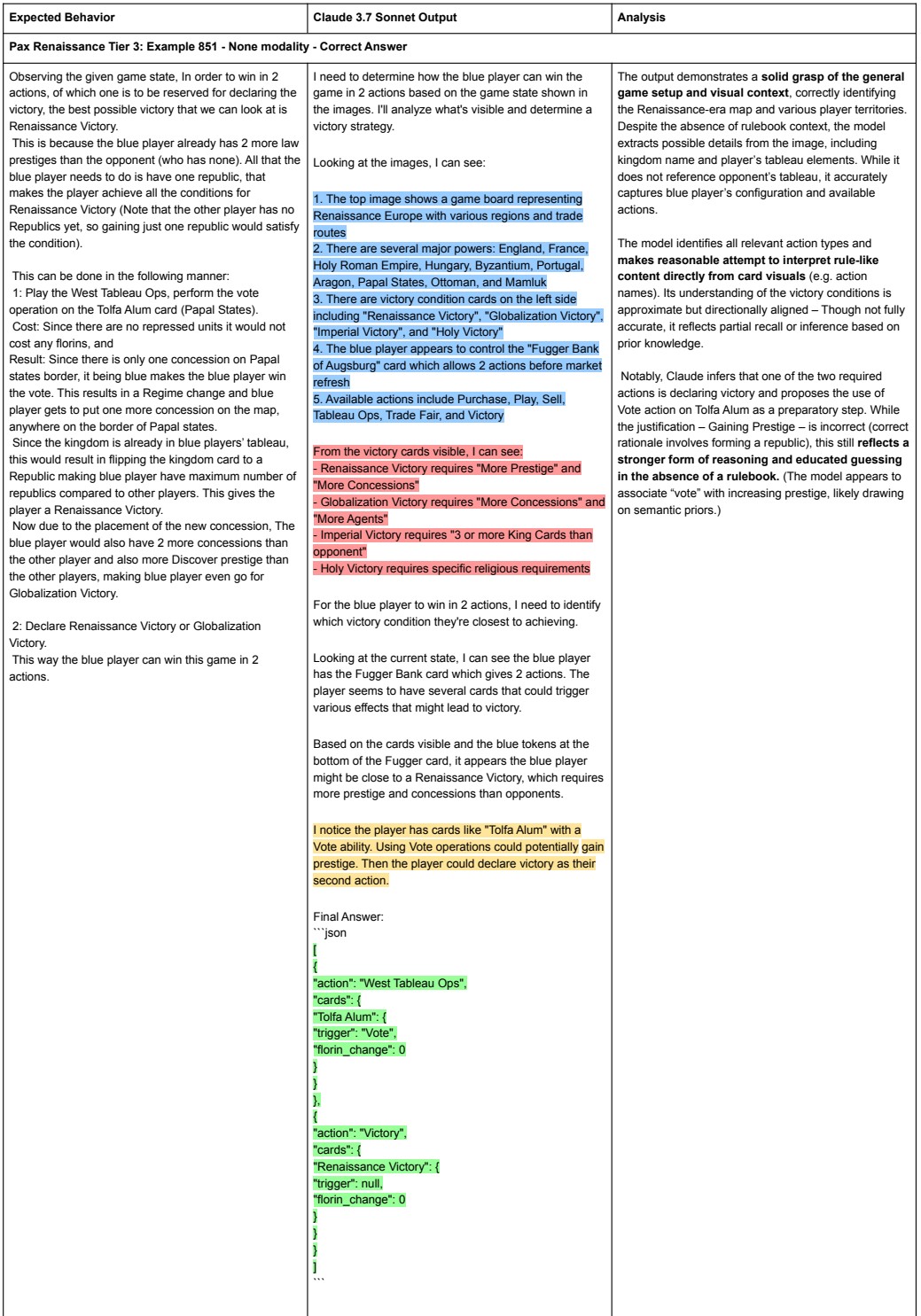

| Expected Behavior | Claude 3.7 Sonnet Output | Analysis |
|---|---|---|
| **Pax Renaissance Tier 3: Example 851 - None modality - Correct Answer** | | |
| Observing the given game state, In order to win in 2 actions, of which one is to be reserved for declaring the victory, the best possible victory that we can look at is Renaissance Victory.

This is because the blue player already has 2 more law prestiges than the opponent (who has none). All that the blue player needs to do is have one republic, that makes the player achieve all the conditions for Renaissance Victory (Note that the other player has no Republics yet, so gaining just one republic would satisfy the condition).

This can be done in the following manner:
1: Play the West Tableau Ops, perform the vote operation on the Tolfa Alum card (Papal States).
Cost: Since there are no repressed units it would not cost any florins, and
Result: Since there is only one concession on Papal states border, it being blue makes the blue player win the vote. This results in a Regime change and blue player gets to put one more concession on the map, anywhere on the border of Papal states.
Since the kingdom is already in blue players' tableau, this would result in flipping the kingdom card to a Republic making blue player have maximum number of republics compared to other players. This gives the player a Renaissance Victory.
Now due to the placement of the new concession, The blue player would also have 2 more concessions than the other player and also more Discover prestige than the other players, making blue player even go for Globalization Victory.

2: Declare Renaissance Victory or Globalization Victory.
This way the blue player can win this game in 2 actions. | I need to determine how the blue player can win the game in 2 actions based on the game state shown in the images. I'll analyze what's visible and determine a victory strategy.

Looking at the images, I can see:

1. The top image shows a game board representing Renaissance Europe with various regions and trade routes
2. There are several major powers: England, France, Holy Roman Empire, Hungary, Byzantium, Portugal, Aragon, Papal States, Ottoman, and Mamluk
3. There are victory condition cards on the left side including "Renaissance Victory", "Globalization Victory", "Imperial Victory", and "Holy Victory"
4. The blue player appears to control the "Fugger Bank of Augsburg" card which allows 2 actions before market refresh
5. Available actions include Purchase, Play, Sell, Tableau Ops, Trade Fair, and Victory

From the victory cards visible, I can see:
- Renaissance Victory requires "More Prestige" and "More Concessions"
- Globalization Victory requires "More Concessions" and "More Agents"
- Imperial Victory requires "3 or more King Cards than opponent"
- Holy Victory requires specific religious requirements

For the blue player to win in 2 actions, I need to identify which victory condition they're closest to achieving.

Looking at the current state, I can see the blue player has the Fugger Bank card which gives 2 actions. The player seems to have several cards that could trigger various effects that might lead to victory.

Based on the cards visible and the blue tokens at the bottom of the Fugger card, it appears the blue player might be close to a Renaissance Victory, which requires more prestige and concessions than opponents.

I notice the player has cards like "Tolfa Alum" with a Vote ability. Using Vote operations could potentially gain prestige. Then the player could declare victory as their second action.

Final Answer:
```json
[
{
"action": "West Tableau Ops",
"cards": {
"Tolfa Alum": {
"trigger": "Vote",
"florin_change": 0
}
}
},
{
"action": "Victory",
"cards": {
"Renaissance Victory": {
"trigger": null,
"florin_change": 0
}
}
}
]
``` | The output demonstrates a **solid grasp of the general game setup and visual context**, correctly identifying the Renaissance-era map and various player territories. Despite the absence of rulebook context, the model extracts possible details from the image, including kingdom name and player's tableau elements. While it does not reference opponent's tableau, it accurately captures blue player's configuration and available actions.

The model identifies all relevant action types and **makes reasonable attempt to interpret rule-like content directly from card visuals** (e.g. action names). Its understanding of the victory conditions is approximate but directionally aligned – Though not fully accurate, it reflects partial recall or inference based on prior knowledge.

Notably, Claude infers that one of the two required actions is declaring victory and proposes the use of Vote action on Tolfa Alum as a preparatory step. While the justification – Gaining Prestige – is incorrect (correct rationale involves forming a republic), this still **reflects a stronger form of reasoning and educated guessing in the absence of a rulebook.** (The model appears to associate "vote" with increasing prestige, likely drawing on semantic priors.) |

Figure 24: Claude Knowledgeable Guesses Part 1.

| Expected Behavior | Claude 3.7 Sonnet Output | Analysis |
|---|---|---|
| **Res Arcana Tier 3: Example 549 - Image Modality - Correct Answer** | | |
| In the given situation, The player has Reanimate magic item, Druid mage, Mermaid creature, Sunken Reef and Sacrificial Pit Places of power, along with magical card and elemental spring cards in hand.

The amount of victory points present are 2 from sacrificial pill and 2 (1x2 calm placed on card) from sunken reef. So there are a total of 4 victory points present.

We can add vp either by using action on the mermaid card and placing calm onto sunken reef, (or) by using the second action on Sacrificial pil and gaining gold essences to buy a monument one (2 actions), or by gaining 3 life essences and placing one death essence onto pil using first action (3 actions).

Now out of the possible ways, one could use the action on mermaid but that would only add one point and needs only one action leading to just one more vp, instead performing the second action on Sacrificial pill could result in purchasing the Temple monument (can be done in 2 actions) and gaining 2 points.

The actions to be taken are:
1: Perform the second action on the Sacrificial pill by turning the card, giving away one death essence, destroying the mermaid and gaining 4 gold essences.
2: Using the previously gained 4 gold essences purchase the Temple monument that would add 2 points. As a result the final amount of

victory points present are 4 + 2 = 6.

Final Answer:
Previous amount: 4
New amount: 6 | I'll analyze this game state to determine the maximum guaranteed VP gain in two actions. Let me assess the current state and available options.

**Current Game State:**
- Player 1 has a Druid mage card, Mermaid (Creature), and has claimed two Places of Power (Sunken Reef and Sacrificial Pit)
- In hand: Magical Shard and Elemental Spring artifacts
- There are two monuments visible in the shared area: Temple and Solomon's Mine
- The player has the Reanimate magic item

**Current VP Count:**
- Sunken Reef: 1 VP (visible at bottom of card)
- Sacrificial Pit: 2 VP + 1 VP (appears to show both a fixed 1 VP plus 1 VP for a Death essence on it)
- Total current VP: 4 VP

**Possible Actions:**
1. Claim a monument (Temple or Solomon's Mine)
2. Place an artifact from hand (Magical Shard or Elemental Spring)
3. Use a power on a straightened component
4. Discard a card for essences

Looking at the monuments:
- Temple provides 2 VP (shown at bottom)
- Solomon's Mine provides 1 VP (shown at bottom)

Since monuments require 4 Gold to claim (according to rules), and I don't see sufficient Gold in the player's pool, claiming a monument might not be immediately possible.

The Magical Shard has a cost of 0, so it can be placed for free. Elemental Spring has some essence cost.

Let's evaluate the best two actions:

**Action 1:** Place Magical Shard (costs 0) - this gives no immediate VP but can potentially help with essence collection

**Action 2:**
Looking at the available powers on the components, I see the Sacrificial Pit has a power that seems to provide resources based on destroying something. This might be useful if we can destroy the Magical Shard we just placed.

From what I can see, the guaranteed way to gain VP would be to claim the Temple monument if we can gather the needed resources, which would give +2 VP.

*However, the image doesn't clearly show all the player's essence pools, so I'm limited in determining if the player has enough resources to claim either monument.*

Based on visible information, claiming the Temple would provide the most VP (+2), bringing the total from 4 VP to 6 VP.

Final Answer: Previous amount: 4, New amount: 6 | In this example the model has access to the rule book in the image modality. The model correctly **identifies most elements of game state**, including the available cards, and also **retrieves full set of possible actions** it can perform. It demonstrates a clear understanding of strategic play: recognizing when actions are infeasible, planning sequential steps, and aiming to maximize rewards.

The model does **misinterpret key details** regarding card essences. It occasionally attributes essences to cards that lack them, while overlooking those that do. Moreover, it also calculates the victory incorrectly – even though in this case it matches the correct answer just by chance.

Despite these errors, the model shows **partial rule comprehension and strategic foresight**. For example, it partially identifies a power on Sacrificial Pit (destroy a card to gain essences) and proposes placing a card solely to destroy it in the next turn using this power—an approach that aligns with maximizing resource gain, even though it relies on an assumed card and partial rule understanding.

Crucially, the model does **explicitly acknowledge its limitations**. It clearly states that it cannot verify full player's resources and qualify its strategy accordingly ("based on visible information…"). This **self-aware reasoning**—balancing visible data with plausible extrapolation—makes its output **more trustworthy and interpretively grounded than many counterparts**. |

Figure 25: Claude Knowledgeable Guesses Part 2.

| Expected Behavior | Claude 3.7 Sonnet Output | Analysis |
|---|---|---|
| **Pax Renaissance Tier 2: Example 720 - Text modality - Correct Answer** | | |
| The Sophia Palaiologina card present in the market, with the coronation one shot.
According to the rules the coronation can be performed to any one of the kings listed on the card.
This card has 3 kingdome names: Hungary, Byzantium and Holy Roman Empire (HRE).
Out of the given 3 kingdoms, two kingdoms (HRE and Byzantium) are not prenent on the throne or in blue player's table, so only option left is: Hungary.
Therefore the answer is Hungary. | I need to analyze the Sophia Palaiologina card to determine which king she can marry through her coronation one-shot.

Looking at the images, I can see the Sophia Palaiologina card in the second-to-last row of cards. This is a coronation card (has a coronation one-shot). According to the rules for coronations (section I), when you play a Queen with a coronation one-shot, you must choose one of her suitors listed on the card to form a royal couple.

On Sophia Palaiologina's card, I can see she has "Hungary" as her location. *When I look at her card details in the image, I can see she has a list of suitors, though the specific countries are not entirely clear in the image.*

According to the rules in section I1, if you trigger a coronation One-shot, you form a royal couple by placing the King side of the chosen Empire Square on top of the Queen card in your Tableau.

Based on the card's location being Hungary, and following the typical pattern of coronation cards, Sophia Palaiologina would be able to marry the King of Hungary (Mátyás Corvinus, visible on the game board).

Final Answer: Hungary | The model accurately identifies the location and context of the card in question, recognizing that it is a Coronation card and **retrieving the relevant rules for coronation procedures directly from the rulebook.** The model exhibits a **clear understanding of game mechanics** and performs well in rule-grounded reasoning.
Importantly, the model explicitly states its inability to read the list of suitors from the image – **showing transparency about its perceptual limitations**. To compensate, it hallucinates that the card is located in Hungary, which is, coincidentally, the correct answer.

This hallucination appears to be influenced by a sample coronation scenario in the rulebook as shown below:

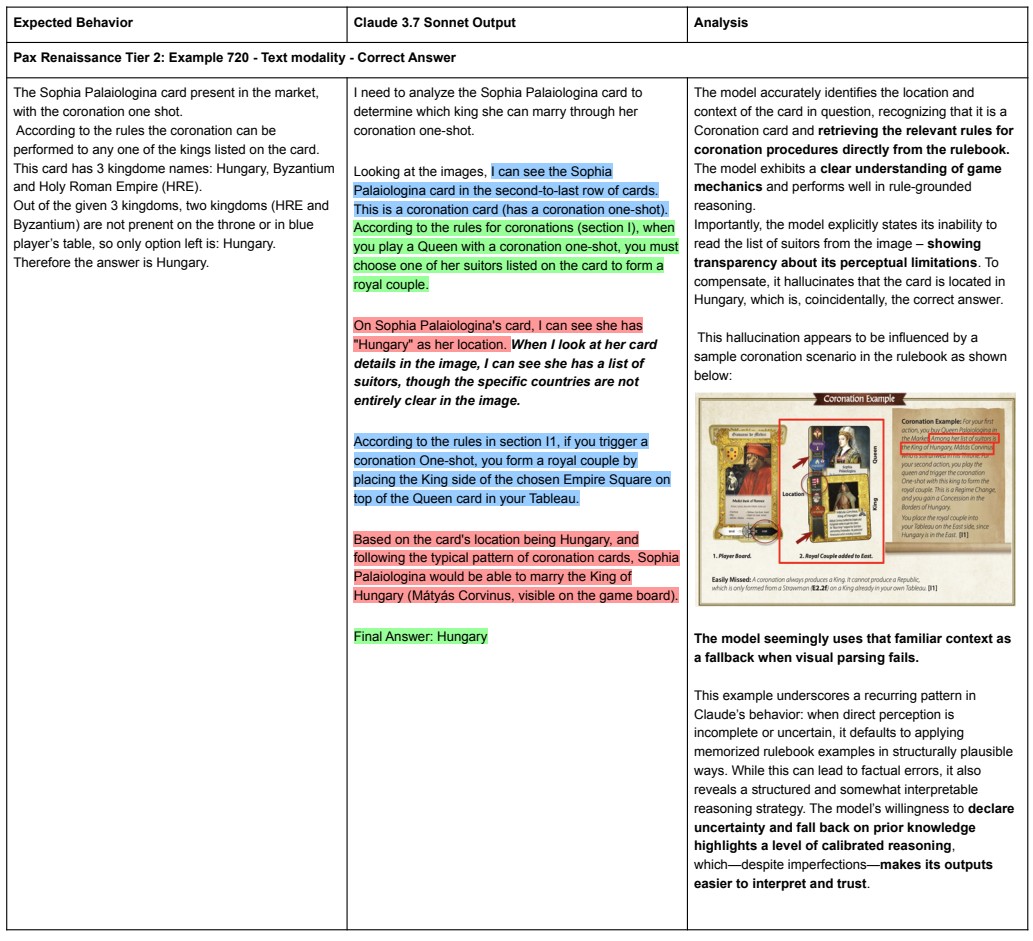

**The model seemingly uses that familiar context as a fallback when visual parsing fails.**

This example underscores a recurring pattern in Claude's behavior: when direct perception is incomplete or uncertain, it defaults to applying memorized rulebook examples in structurally plausible ways. While this can lead to factual errors, it also reveals a structured and somewhat interpretable reasoning strategy. The model's willingness to **declare uncertainty and fall back on prior knowledge highlights a level of calibrated reasoning**, which—despite imperfections—**makes its outputs easier to interpret and trust**. |

Figure 26: Claude Knowledgeable Guesses Part 3.

## H   o3 ANALYSIS

Despite outperforming prior baselines on our multimodal reasoning tasks, the o3 model still exhibits several critical shortcomings that highlight the limits of its current reasoning capabilities.

Overall, the o3 model performs better than existing baselines, but several key limitations still remain:

- **Overthinking and self-doubt**: A recurring issue is the model's tendency to overanalyze or second-guess itself. For example, even after correctly identifying a rule or game state element, it sometimes questions its own logic and changes course; ultimately arriving at an incorrect answer despite being initially on the right path.

- **State tracking and perception issues**: Across games and modalities, o3 struggles to accurately track intermediate states or parse visual/textual information correctly. This often leads to errors early in the reasoning chain, even when the overall logic is sound.

- **Shallow abstraction vs deep understanding**: In games like Pax Renaissance, even though the model is yielding the best performance, on further analysis we found that o3 can often generate high-level correct answers (e.g., it can suggest the correct high-level action types the player should take), but fails when more granular outputs are needed (e.g., if asked to specify the specific cards or concrete operations sequence required); Refer Fig. 27 for more details. This suggests it is able to grasp generic gameplay logic still lacks the ability to deeply integrate its knowledge with state-specific details.

**Example 804, 805, 806: Pax Ren 2e tier 3 question.**

**Question**: How can the Blue player to earn most florins without selling your cards on their turn? If there are multiple ways to earn the maximum amount of florins, give the solution that minimizes the florins earned by the opponents.

| Level of Detail | High level / Abstracted | Moderate level | Fine Grained |
|---|---|---|---|
| **Original Solution** | `[`
  `{`
  `'action': 'East Tableau Ops'`
  `},`
  `{`
   `'action': 'West Trade Fair'`
  `}`
`]` | `[`
  `{`
  `'action': 'East Tableau Ops',`
  `'cards': {`
    `'Phocoean Alum': {}`
    `}`
  `},`
  `{`
  `'action': 'West Trade Fair',`
  `'cards': {`
    `'West': {}`
    `}`
  `}`
`]` | `[`
  `{`
  `'action': 'East Tableau Ops',`
  `'cards': {`
    `'Phocoean Alum': {`
     `'trigger': 'Commerce',`
     `'florin_change': 1`
     `}`
    `}`
  `},`
  `{`
  `'action': 'West Trade Fair',`
  `'cards': {`
    `'West': {`
     `'trigger': None,`
     `'florin_change': 2`
     `}`
    `}`
  `}`
`]` |
| **O3 Predicted output** | ` ```json `
`[`
  `{`
  `"action": "East Tableau Ops"`
  `},`
  `{`
  `"action": "West Trade Fair"`
  `}`
`]`
` ``` ` | ` ```json `
`[`
  `{`
  `"action": "East Tableau Ops",`
  `"cards": {`
    `"Phocaean Alum": {}`
    `}`
  `},`
  `{`
  `"action": "East Trade Fair",`
  `"cards": {`
    `"East": {}`
    `}`
  `}`
`]`
` ``` ` | ` ```json `
`[`
  `{`
  `"action": "West Tableau Ops",`
  `"cards": {`
    `"Fugger Bank of Augsburg": {`
     `"trigger": "Commerce",`
     `"florin_change": 1`
     `}`
    `}`
  `},`
  `{`
  `"action": "West Trade Fair",`
  `"cards": {`
    `"West": {`
     `"trigger": null,`
     `"florin_change": 1`
     `}`
    `}`
  `}`
`]`
` ``` ` |
| **Inference** | The model correctly interprets the question and identifies the high-level sequence of actions the blue player should take to maximize florin gains. While the detailed reasoning tokens/ thought process is unavailable,, the model demonstrates a rough understanding of the goal and a plausible path to reach it for sure. | The model correctly identifies that the blue player can leverage a card in their tableau to gain additional florins and recognizes that initiating a trade fair is a viable path to do so. However, it fails to discern that the West Trade Fair yields a higher payoff (2 florins) compared to the East Trade Fair (1 florin). This oversight appears to stem from the model second-guessing its reasoning, likely due to overthinking or lack of confidence, despite having access to the relevant information. | The model correctly grasps the core logic that the blue player can activate a card in their tableau to gain additional florins, and it also identifies trade fairs as a valid mechanism to do so. In this instance, the model initially leans toward the West Trade Fair—rightly recognizing that it offers a higher florin yield—but ultimately fails to maintain an accurate count of the florins it would generate. This confusion likely stems from misinterpreting or conflating the East and West Trade Fair lines in the image. Additionally, the model mistakenly names the activating card, suggesting it may be overthinking or experiencing referential ambiguity during generation. |
| **Conclusion** | Across all three modalities, we observe that even frontier vision-language models exhibit consistent reasoning gaps. While they often grasp the high-level goal or action structure, they frequently fail at crucial details—such as tracking spatial relations, identifying the correct rule logic, or counting intermediate resource changes—leading to suboptimal or invalid predictions.

**Note**: The examples shown here are the final model predictions generated by GPT-o3 on a subset of our evaluation questions. While we can speculate on failure modes based on outputs, we cannot confirm the exact internal logic or missteps, as the model's intermediate "thinking tokens" are not surfaced via the API. The error analyses reflect the most likely interpretation based on model behavior and answer structure, and reasoning when models are prompted through frontend UI interface. | | |

Figure 27: o3's Granularity Analysis

# I TABLETOP SIMULATOR INTERFACE

We illustrate the Tabletop Simulator interface used to generate board images in Fig. 28. Annotators had full control over the scene configuration and camera angle, enabling natural, human-like viewpoints that reflect how players visually engage with the game during actual play.

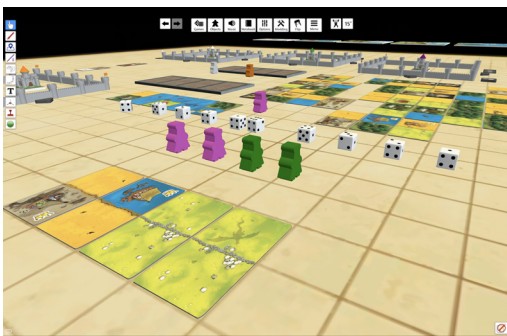 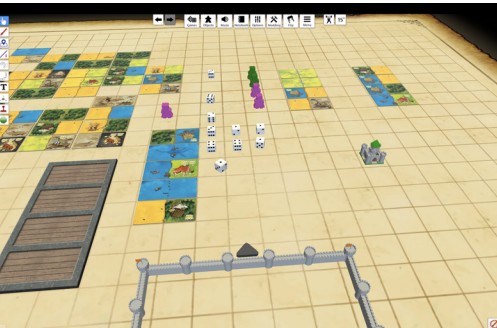

Figure 28: Illustration of the Tabletop Simulator interface with dynamic camera angles, allowing the generation of natural, human-like game state views closely matching real tabletop gameplay.

# J ANNOTATOR ONBOARDING AND TRAINING

To support high-quality, consistent annotations in our benchmark, we recruited three annotators with backgrounds in computer science and data science. Annotators were selected based on their ability to follow complex instructions, their interest in strategic and game reasoning. Their onboarding process was structured around three components, detailed below.

## J.1 GAME ONBOARDING PROCESS

Annotators were onboarded through a structured familiarization process designed to ensure basic fluency with the games and task format. This involved reviewing the official rulebooks and, where helpful, consulting tutorial or gameplay videos. Annotators also engaged in light gameplay practice to understand typical game dynamics. Before full-scale annotation, they completed a brief set of calibration examples to align on expectations and task clarity. The annotation interface (Fig. 29) allowed annotators to submit both their answers and short rationales in response to assigned questions.

## J.2 GAME-SPECIFIC LEARNING REFLECTIONS

**KingDomino:** Annotators were able to understand the rules of this game and master it in a single pass, successfully answering all Tier 1 and Tier 2 questions. However, despite the relative simplicity of the game, Tier 3 questions proved more challenging than expected due to the spatial awareness and perceptual reasoning required. In contrast, Res Arcana was more streamlined, with rules and game state that were easier to interpret even for Tier 3 tasks.

**Res Arcana:** Annotators were able to understand this game's rules with a single pass through the rulebook and tutorial videos. Gameplay clarified timing and card interactions, but the overall mechanics were intuitive and required less post-hoc clarification.

**Pax Renaissance:** This game required multiple passes through the rulebook, paired with video walkthroughs and extensive gameplay. Annotators reported that understanding the rules in isolation was difficult; gameplay was essential to grasping the card interactions and victory conditions. Group discussion played a major role in resolving rule questions that emerged during play.

**Catan:** This game involved interacting components: especially understanding trading strategies and multi-step action sequences, to maximize points. Tier 1 and Tier 2 questions were largely

answerable after a single pass through the rulebook. Tier 3 questions, however, required a small number of clarifications around which constraints were applicable (e.g., trading restrictions), how different trade mechanisms interact, and how resource limits affect what actions are feasible within a turn.

**Carcassonne:** This game was more advanced than Kingdomino in terms of reasoning about tile orientations and end-of-game scoring. Tier 1 and Tier 2 questions were generally easy to answer after a pass through the rulebook. Tier 3 questions, however, took more time, since annotators had to explore multiple placement options (including rotations) and verify several alternative scoring paths before selecting the optimal line.

Across all games, the onboarding process was interactive, involving both passive learning (videos, reading) and active learning (gameplay, discussion). This structure allowed annotators to develop the contextual knowledge required for precise, consistent labeling.

### J.3    ANNOTATION PROCEDURE AND VALIDATION

All example prompts were originally created by two expert contributors. A shared document containing each example, the expert-provided rationale, and the expert-labeled answer served as the basis for discussion and comparison. However, for validation, all examples were re-annotated from scratch by the three student annotators (without access to the expert answers) to ensure blind evaluation and eliminate potential bias.

To resolve ambiguity and improve agreement:

- Each annotator labeled their portion of the dataset independently.
- We extract all annotator answers and compare them to the expert answers.
- Disagreements were flagged for collaborative review and discussion in group meetings.
- A tracking spreadsheet containing expert rationales and annotations was revisited to compare rationales and refine interpretations where necessary.
- Final labels were only assigned after unanimous agreement among all three annotators and an expert was reached.

### J.4    ANALYSIS OF ANNOTATION CONFLICTS BETWEEN EXPERT AND ANNOTATORS

To evaluate student annotator consistency and identify common sources of disagreement in our ground truths, we analyze the results of the annotators' responses to the dataset questions, comparing them with the original 'expert' answers. In Table 9, we overview outcomes where annotators and experts answer differently, and in Table 10 we categorize these disagreements into common failure modes:

- **Rules Application**: Misapplication or misunderstanding of the formal game rules.
- **Math Error**: Mistake in arithmetic or numerical calculation steps.
- **Typo**: Minor transcription, spelling, or formatting error in the answer.
- **Planning Error**: Failure to consider a condition or case required for complete reasoning (i.e, player did not fully enumerate the action possibilities and therefore arrived at a suboptimal solution)
- **Comprehension Error**: Partial or incomplete reasoning due to incorrect question understanding.
- **Environmental Perception Error**: Error caused by incorrect visual or textual judgment of the game state (e.g., missing crowns on a new domino).
- **Ambiguous Question**: The question is vague, underspecified, or allows multiple valid interpretations.

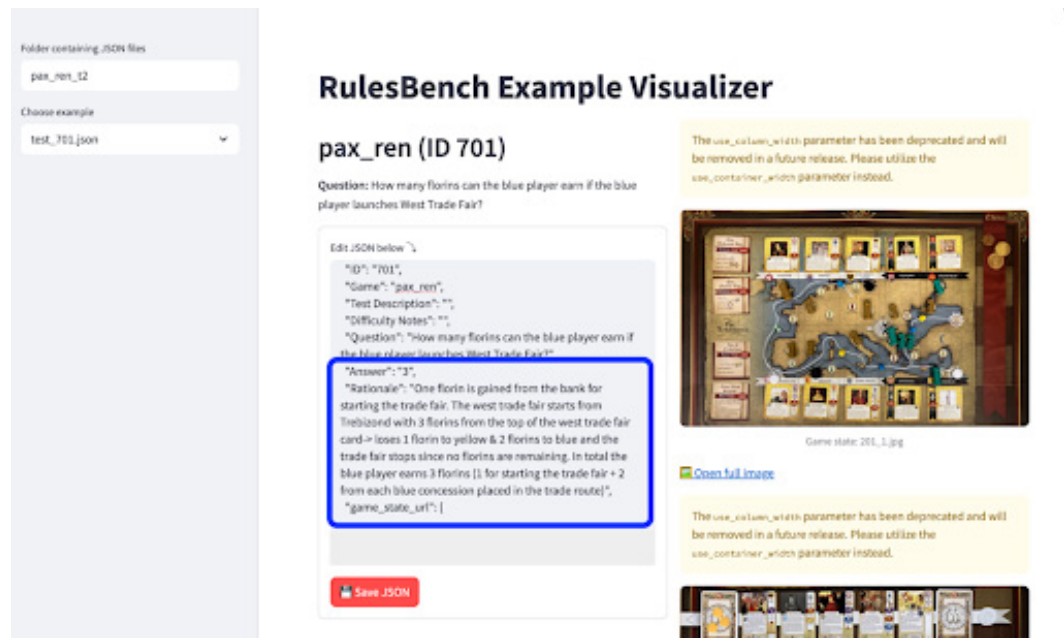

Figure 29: Annotation interface for LUDOBENCH game questions. The highlighted section indicates the field where the annotator can enter a final answer and supporting rationale.

| Tier | Expert Errors | Annotator Errors | Both Wrong | Expert OK | Annotator OK | Total |
|---|---|---|---|---|---|---|
| Tier 1 | 2 | 4 | 0 | 117 (98.3%) | 115 (96.7%) | 119 |
| Tier 2 | 3 | 13 | 0 | 116 (97.5%) | 106 (89.1%) | 119 |
| Tier 3 | 16 | 24 | 4 | 137 (87.2%) | 129 (82.2%) | 157 |

Table 9: **Overview of expert vs annotator correctness.** Overall we see very high agreement, with most conflicts resolved in favor of the expert.

### J.5 GAME RULEBOOK MATERIALS

- **KingDomino:** KingDomino Rulebook (2nd Edition)
- **Res Arcana:** Res Arcana Rulebook
- **Pax Renaissance 2e:** Pax Renaissance 2e Rulebook
- **Catan:** Catan Rulebook
- **Carcassonne:** Carcassonne Rulebook

### J.6 EXAMPLE GAME VIDEO TUTORIALS

- **KingDomino:** https://www.youtube.com/watch?v=smbwBPmP4Ms&ab_channel=WatchItPlayed

| Tier | Rules App. | Math Err. | Typo | Planning Err. | Comp. Err. | Percep. Err. | Ambig. Q. | Total / Size |
|---|---|---|---|---|---|---|---|---|
| Tier 1 | 0 | 0 | 2 | 0 | 2 | 1 | 1 | 6 / 119 |
| Tier 2 | 6 | 4 | 3 | 0 | 3 | 3 | 0 | 19 / 119 |
| Tier 3 | 7 | 7 | 6 | 10 | 0 | 7 | 4 | 38 / 157 |

Table 10: **Breakdown by failure modes in the inconsistent answers between expert and annotator.** Rules application and math errors dominate Tier 2, while planning and perceptual errors become more prominent in Tier 3.

- **Res Arcana:** `https://www.youtube.com/watch?v=smbwBPmP4Ms&ab_channel=WatchItPlayed`
- **Pax Renaissance 2e:** `https://youtu.be/JDMWjdHwPr0?si=Z1-u8RRRsOy_l8GA`
- **Catan:** `https://www.youtube.com/watch?v=oiQ6SgBzfqY`
- **Carcassonne:** `https://www.youtube.com/watch?v=R1qh-lhxy9s`

