# OpenReview forum: "LLMs as Rules Oracles: Exploring Real-World Multimodal Reasoning in Tabletop Strategy Game Environments"
_ICLR.cc/2026/Conference — ICLR 2026 Poster_

### Official Review · Reviewer_cEf5 · 2025-10-21

**Soundness:** 3
**Presentation:** 3
**Contribution:** 2
**Rating:** 4
**Confidence:** 3

**Summary:**

This paper examines the multimodal reasoning capabilities of LLMs within the complex, rule-driven setting of tabletop strategy games. The authors propose a three-tier framework of game understanding — environment perception, heterogeneous rule integration, and short-horizon optimization — and evaluate how well models perform across these layers. Using a newly constructed benchmark, LUDOBENCH, which includes three real-world board games of varying complexity, the study assesses eight state-of-the-art multimodal LLMs (including GPT-4.1, Gemini 2.5 Pro, Qwen-VL 2.5 32B, and Claude 3.7 Sonnet) under three rulebook input conditions: text, image, and none. The results reveal that even at the lowest tier—environment perception—performance is inconsistent, and it degrades sharply when rule retrieval and planning are required. Common failure modes include misreading cluttered board states, retrieving but misapplying rules, overlooking opponents’ states, and hallucinating illegal actions. Taken together, these findings indicate that current LLMs are not reliable “rule oracles” for analog tabletop play and underscore the need for stronger cross-modal grounding and faithful rule execution.

**Strengths:**

1. Clear, well-structured writing and figures make the method and setup easy to follow.
2. The paper’s three progressive competencies — environment perception, heterogeneous rules integration, and short-horizon optimization — mirror how humans build understanding step by step. This structuring makes results interpretable and helps localize failure modes.
3. The experiments show models often perform worse with visual inputs than with plain-text rules — the opposite of human behavior. This is an interesting, discussion-worthy result.
4.  The study employs a multi-stage annotation pipeline that includes checks for annotator–expert agreement, which enhances the reliability of the benchmark. This methodological rigor improves the transparency, reproducibility, and credibility of the reported results.

**Weaknesses:**

1. While tabletop games offer well-defined, structured rule systems, they are highly domain-specific and differ markedly from real-world reasoning contexts.  In this setting, rule comprehension can overshadow genuine reasoning, which may limit the benchmark’s ability to reflect broader inferential capability.  Consequently, it is unclear how LUDOBENCH performance should be interpreted as a measure of general reasoning: the paper provides no evidence that its scores correlate with wider reasoning skills or transfer beyond tabletop domains.
2.  Many tabletop rulebooks and gameplay descriptions are publicly available online. How do the authors ensure that the evaluated models had not already seen the same or similar rule texts during pretraining? If not, superior performance under text-based conditions might partially reflect memorization rather than genuine reasoning or rule integration.
3. The benchmark covers only three tabletop games with a steep difficulty jump (1.2 → 2.6 → 4.6) and narrow genre coverage. This constrains external validity and makes it hard to trace how performance scales with incremental complexity.

**Questions:**

1. The paper states that the setting “reflects the experience of real-world gamers during first-time gameplay,” and the conclusion emphasizes “one-shot” scenarios. However, the human baselines (expert / hobbyist) are clearly not first-time players. How do the authors reconcile this with the stated goal, and would a novice baseline alter the model–human gap?
2.  In some T1-Perception results (e.g., model o1 on KingDomino and Res Arcana), the no-rulebook condition outperforms having rules. What explains this inversion?
3.  Many cards in tabletop games contain stylized artwork that diverge from LLM pretraining data or real life. Given that even basic perception is challenging, the study should first separate perception from reasoning with a dedicated pretest (e.g., card/asset/symbol recognition) before tier-by-tier game reasoning.

---

> ### Author Response · Authors · 2025-12-01
> **Response to Reviewer cEf5 (1/3)**
>
> Thanks for your review!
>
> To summarize our review response and corresponding paper improvements:
> - We address your feedback relating to:
>     - Benchmark impact and whether it reflects real-world reasoning .
>     - Do models see these games during pretraining and what effect does this have?
>     - How does LudoBench ‘reflect the experience of real-world gamers during first-time gameplay’?
> - We have expanded the size of LudoBench, addressing questions about breadth and steep difficulty jump.
>     - We now cover five games with high genre coverage and ‘game mechanics’ diversity.
> - We clarify questions about T1 reasoning performance and perception pretests.

---

> ### Author Response · Authors · 2025-12-01
> **Response to Reviewer cEf5 (2/3)**
>
> ### **Re: Whether LudoBench is a measure of general reasoning.**
>
> We do not necessarily argue that LudoBench is a measure of general reasoning, however, we feel it probes a diverse and realistic set of very real world competencies; LudoBench simultaneously tests:
>
> 1) challenging scene parsing (extracting and grounding details from complex scenarios – beyond simple OCR or object recognition),
> 2) complex multimodal knowledge retrieval (interpreting dense text passages, and aligning them with visual and tabular information with perfect fidelity), and
> 3) real-world planning (simulating a complex environment internally with one’s own world model).
>
> Further, we would argue that situated tabletop game reasoning is not just some interesting proxy for the real world, but is intrinsically a highly-relevant, impactful domain of study; millions play tabletop games every year (the game *Catan* -- which we evaluate -- *alone has sold over 45M copies* since its release), and the challenges LudoBench presents are exactly those needed for enabling LMs to engage in messy, situated environments. Our work bridges the gap from conventional academic analysis of game reasoning, and instead focuses on assessing competence for the messy, real-world problems that humans face.
>
> ---
>
> ### **Re: Exposure to rulebooks during pretraining.**
>
> Great question, and it addresses a key topic in our benchmark framing. It is very reasonable for models to have seen our five studied games during pretraining; all have publicly-available rulebooks (often on the publisher website), and some (e.g. Catan) are quite prominent within the tabletop gaming zeitgeist. Additionally, all of the games we explore were published well before the training cut-off of all models we test. **Ultimately, our motivation is to assess whether models can solve real-world tasks that are relevant to humans, and it is entirely valid for models to leverage both intrinsic and extrinsic knowledge to solve these problems.**
>
> That said, we do see clear signs that intrinsic knowledge alone is not sufficient for reasoning, indicating the importance of long-context multimodal reasoning for rules integration:
> 1) Across the board, we see that Text-modality rulebooks consistently improve performance over None for Tier 2 and Tier 3 tasks (Tier 1 questions do not require rulebooks).
> 2) For recent thinking models (we have updated the paper draft with Gemini 3 Pro and GPT 5.1 results), we see that these models are powerful enough to leverage rulebook screenshots for multimodal rules retrieval, improving performance significantly. We see this in Pax Renaissance, a game whose 44 page ruleset is notoriously difficult to internalize. The avg. performance of the strongest thinking models (5.1, o3, Pro 3) is 0.08, 0.12, 0.21 for None/Text/Image modalities.
>
> Overall, we see through our newly-created, high-quality examples that **simply seeing rulebooks does not mean games can truly grasp and rehearse the core reasoning logics behind the rules**, and our evaluation results prove this point exactly.
>
> ---
>
> ### **Re: Dataset scope, difficulty jumps, and genre coverage.**
>
> Since the original submission, we have expanded LudoBench to include 600+ examples across five games, adding 100+ examples from the games Catan and Carcassonne – the two the most-favorited games on BoardGameGeek. These additions broaden thematic and mechanical coverage, and smooth the difficulty curve
>
> With respect to genre coverage, we intentionally choose games to cover a wide variety of genres and mechanisms. Notably, according to [BoardGameGeek](https://boardgamegeek.com), the current games cover 11 of the top genre [categories and themes](https://boardgamegeek.com/browse/boardgamecategory), and 30 of the top game design [mechanisms](https://boardgamegeek.com/browse/boardgamemechanic), a few of which are listed below:
> - Categories: *Abstract strategy, Card game, Economic, Puzzle, Territory building.*
> - Themes: *Medieval, Fantasy, Religious, and Abstract*.
> - Mechanisms: *Action Points, Chaining, Hand Management, Hexagon Grid, Modular Board, Multi-Use Cards, Network and Route Building, Pattern Building, Simulation, Tile Placement, Trading, and Turn Orders.*
>
> ---
>
> ### **Re: Does LudoBench reflect the experience of real-world gamers during first-time gameplay?**
>
> Good question. We would expect a complete gaming novice or non-gamer to fare quite poorly on these ‘rules oracles’ tasks, particularly for games with longer rulesets. In practice, most gaming groups rely on a designated player to both interactively teach a game to newcomers and also serve as a rules oracle during gameplay. This role is generally assumed by either a player with prior game experience (an ‘Expert’), or a dedicated gamer that has spent considerable time learning the game in advance (a ‘Hobbyist’). Thus, the questions LudoBench poses are directly in line with those that would be asked of such an expert or hobbyist-level human rules oracle.

---

> ### Author Response · Authors · 2025-12-01
> **Response to Reviewer cEf5 (3/3)**
>
> ### **Re: Tier 1 (Perception) – sometimes the no-rulebook setting outperforms having rules?**
>
> We kindly point out that Tier 1 is intentionally scoped as reference-free perception task (Section 3.1). Namely, these questions can be directly answered from the image with simple visual recognition and some familiarity with generic gaming vernacular (e.g. what are dice, what do dominos look like?) – without requiring *any* game-specific terminology. The visualizer link (https://anonymous-researcher-2026.github.io/anon-submission-2026/) demonstrates these examples clearly. As Tier 1 does not require rule grounding, it is possible that prepending the full rulebook (ie, the ‘Text’ and ‘Image’ variants) could simply distract the model from its perception task. For example, assume a Tier 1 example ask the color of a pictured pawn. If the rulebook has a specific definition for green-colored items (such as ‘Life pieces’), the model may inadvertently say ‘Life’ rather than ‘Green’.
>
> ---
>
> ### **Re: A pre-test (Tier 0) to assess cards/symbol comprehension.**
>
> Tier 1 examples, by design, are scoped as such. Tier 1 is a rule-agnostic perception tier by design: examples focus on identify token/symbol counts, icon/terrain identification, orientation/adjacency, and simple spatial relations, with no rule lookup required.

---

### Official Review · Reviewer_k6xk · 2025-10-30

**Soundness:** 3
**Presentation:** 4
**Contribution:** 2
**Rating:** 4
**Confidence:** 4

**Summary:**

The paper introduces LUDOBENCH, a new multimodal reasoning benchmark designed to evaluate the ability of VLMs to act as "rules oracles" for complex tabletop strategy games. The task challenges a model to, given a visual depiction of a game state and its corresponding rulebook, answer questions a new player might have. The benchmark is structured into three cumulative tiers of increasing difficulty: (1) Environment Perception (visual understanding of the game state), (2) Heterogeneous Rules Integration (applying rules from the rulebook to the state), and (3) Short-horizon Optimization (simple planning and tactical puzzles).

The authors curated a dataset of 400 question-answer examples across three popular tabletop games of varying complexity: Kingdomino, Res Arcana, and Pax Renaissance 2nd Edition. Evaluating eight state-of-the-art VLMs, the paper reveals significant shortcomings in current models. Even the best models achieve only ~68% accuracy on perception tasks and fall below 10% on optimization puzzles, in stark contrast to human hobbyist players who solve even the hardest puzzles with over 80% accuracy. The paper provides an extensive analysis of failure modes, identifying critical weaknesses in cross-modal grounding, spatial reasoning, state tracking, and the application of abstract rules to messy, unfamiliar environments.

**Strengths:**

-The paper addresses an important gap in AI research: while models have achieved superhuman performance in games like Chess and Go, this work presents the more realistic challenge of learning and reasoning in an environment where rules are not pre-encoded but must be acquired and grounded from multimodal sources.
- Game Selection: The choice of three games with a clear progression in rulebook length, component density, and mechanical complexity provides a strong basis for evaluating model scalability and robustness.
- Tiered Structure: The three-tiered evaluation (Perception, Integration, Optimization) is logical and effectively isolates different capabilities. This cumulative design allows for a clear diagnosis of where reasoning breaks down—a model cannot succeed at Tier 2 without competency in Tier 1.
- The findings are clear: the huge performance gap between SoTA models and humans is a valuable negative result for the community, highlighting how far these models are from human-like comprehension in this domain.
- The paper is well written and the figures are compelling.

**Weaknesses:**

- The provided link to access LudoBench does not work.
- The benchmark's scale is limited. The results, while conclusive for these specific games, may not generalize to the thousands of other tabletop games. The combinatorial nature of game states and rules is vast, and a larger, more diverse set of games and questions would further solidify the paper's claims.
- The paper presents results based on a fixed prompt structure. Given the significant impact of prompt engineering on LLM performance, the results could be sensitive to this choice. It is unclear if more sophisticated prompting techniques (for example breaking down the query into Chain-of-Thought steps for the model) could provide better performance. A more detailed ablation study on prompting would strengthen the conclusion that the failures are fundamental to the models, not just the evaluation setup.  The authors could also report the top-k accuracy to highlight if the models sometimes know the right answer (which would indicate that they could get way better at this task with some finetuning).

**Questions:**

- Why would this benchmark in particular be more valuable than all the others that already exist?

---

> ### Author Response · Authors · 2025-12-01
> **Response to Reviewer k6xk (1/4)**
>
> Thank you for your review! To quickly recap our response to your feedback:
>
> - We have added a link to a dataset visualizer for easy exploration of our dataset.
> - We have expanded LudoBench in both size and breadth of games covered.
> - We have conducted further top-k accuracy analyses to understand if repeated sampling improves the theoretical upper-bound of performance.
> - We discuss the novelty and expected impact of our benchmark.

---

> ### Author Response · Authors · 2025-12-01
> **Response to Reviewer k6xk (2/4)**
>
> ### **Re: Access to LudoBench data.**
>
> Thanks for noting this. The URL in the paper now links to an (anonymized) repository with a dataset visualizer: (https://anonymous-researcher-2026.github.io/anon-submission-2026/). With this you can see the game states and questions presented to the models. We would encourage you to take a look and try a few for yourself! In particular, the Tier 1 questions are very straightforward and answerable without game-specific knowledge.
>
> ---
>
> ### **Re: Dataset scale and generalization.**
>
> First, we have expanded LudoBench to 600+ examples across five games by adding Catan and Carcassonne. These games are the most-rated games according the site [BoardGameGeek](​​https://boardgamegeek.com/browse/boardgame?sort=numvoters&sortdir=desc)  Our expansion widens mechanics and difficulty while preserving our tiered design and annotation quality. The updated set of games cover 11 of the top genre [categories and themes](https://boardgamegeek.com/browse/boardgamecategory), and 30 of the top game design [mechanisms](https://boardgamegeek.com/browse/boardgamemechanic.)
>
> While our evaluation of these five varied games does not guarantee universality across thousands of published tabletop game, however, our benchmark surfaces consistent **capability gaps that are skill- and modality-level rather than title-specific**. In our analyses, we consistently see (i) misperceived state (miscounts, orientation/direction slips), (ii) weak rule-to-scenario grounding in spatial placements (e.g., inability to identify all valid tile place, dropped state updates), and (iii) overfitting to rulebook illustrations that induces hallucinations. We have added two popular, stylistically different games (Catan, Carcassonne) and observed many of the same signatures. Existing works ([1](https://arxiv.org/abs/2412.06394), [2](https://arxiv.org/abs/2505.15146), [3](https://arxiv.org/abs/2501.17186)) do not surface these challenges in real-world, situated settings. Our results highlight clear, transferable capabilities for the community to target.
>
> ---
>
> ### **Re: Top-K accuracy and an upper bound on individual model performance (Part 1/2).**
>
> Our evaluation inference setup is Top-1 inference with CoT prompting. We have explored the overall solvability of our dataset (Figure 7) through an oracle ‘best case’ setup, aggregating all of the answers from all models and rulebook input variants; per your feedback, we have additionally explored this in two other methods:
>
> 1. Top-5 Accuracy: Sampling 5 outputs, labeling an example as correct if any of the 5 outputs are correct.
> 2. Top-modality Accuracy: Sampling 1 output from each rulebook modality (None, Text, Image), labeling an example as correct if any of the 3 outputs are correct.
>
> We explore outputs from GPT4.1 and 5.1 on all tiers and modalities of Kingdomino. We note that Top-5 accuracy can be calculated for each model+modality combination, whereas Top-modality Accuracy is reported for each model... (continued in next comment)

---

> > ### Author Response · Authors · 2025-12-01
> > **Response to Reviewer k6xk (3/4)**
> >
> > ### **Re: Top-K accuracy and an upper bound on individual model performance (Part 2/2).**
> >
> > | Model: GPT 4.1 |          |                |                |                       | Model: GPT 5.1 |          |                |                |                       |
> > |:--------------:|:--------:|:--------------:|:--------------:|:---------------------:|:--------------:|:--------:|:--------------:|:--------------:|:---------------------:|
> > | Tier           | Modality | Top-1 Accuracy | Top-5 Accuracy | Top-Modality Accuracy | Tier           | Modality | Top-1 Accuracy | Top-5 Accuracy | Top-Modality Accuracy |
> > | Tier 1         | None     | 53.7%          | 77.5%          |                       | Tier 1         | None     | 75.0%          | 85.0%          |                       |
> > |                | Text     | 50.6%          | 75.0%          |                       |                | Text     | 71.0%          | 80.0%          |                       |
> > |                | Image    | 54.7%          | 75.0%          |                       |                | Image    | 70.0%          | 82.5%          |                       |
> > |                | Avg.     | 53.0%          | 75.8%          | 69.8%                 |                | Avg.     | 72.0%          | 82.5%          | 83.0%                 |
> > | Tier 2         | None     | 35.8%          | 47.3%          |                       | Tier 2         | None     | 34.7%          | 63.1%          |                       |
> > |                | Text     | 31.1%          | 57.8%          |                       |                | Text     | 40.0%          | 63.1%          |                       |
> > |                | Image    | 30.6%          | 63.1%          |                       |                | Image    | 31.6%          | 52.6%          |                       |
> > |                | Avg.     | 32.56%         | 56.1%          | 53.4%                 |                | Avg.     | 35.5%          | 59.6%          | 52.2%                 |
> > | Tier 3         | None     | 5.4%           | 20.8%          |                       | Tier 3         | None     | 11.2%          | 20.8%          |                       |
> > |                | Text     | 8.7%           | 29.1%          |                       |                | Text     | 6.6%           | 16.6%          |                       |
> > |                | Image    | 5.4%           | 20.8%          |                       |                | Image    | 7.9%           | 22.9%          |                       |
> > |                | Avg.     | 6.5%           | 23.6%          | 12.8%                 |                | Avg.     | 8.6%           | 20.1%          | 27.4%                 |
> > | Overall        | None     | 31.6%          | 48.5%          |                       | Overall        | None     | 40.3%          | 56.3%          |                       |
> > |                | Text     | 30.1%          | 54.0%          |                       |                | Text     | 39.2%          | 53.3%          |                       |
> > |                | Image    | 30.2%          | 52.9%          |                       |                | Image    | 36.5%          | 52.7%          |                       |
> > |                | Avg.     | 30.7%          | 51.8%          | 45.3%                 |                | Avg.     | 38.7%          | 54.1%          | 54.2%                 |
> >
> > We see Top-5 oracle accuracy, as expected, improves results across the board. However, its benefits are still limited on complex tasks (Tier 3), with this performance still under 25%. Interestingly, we see GPT 5.1 fares much better on Tier 3 Top-Modality analysis than 4.1. We theorize that perhaps varying the input rulebooks (none, text-based, image-based) yield more variance in ‘thinking model’ reasoning than 5 runs from the same modality.
> >
> > In addition to this Top-K analysis, we have qualitatively explored various prompts during the course of this project. In general, we found no discernible improvement in reasoning performance when comparing e.g. simple QA vs. CoT vs. Game Reasoning-specific prompts (e.g. “First confirm your perception of the game scene, then apply rules, and then plan a solution.”). We believe this boils down to the following two observations:
> > - Prompts don’t help weak models: Non-thinking models are not strong enough for the LudoBench task, therefore prompts have little effect.
> > - Strong models ignore specific prompts: Thinking models are strong enough to dynamically unroll the problem (independent of instructions), so generic reasoning prompts (e.g. “Think step by step…”) have no meaningful effect on game reasoning success.

---

> ### Author Response · Authors · 2025-12-01
> **Response to Reviewer k6xk (4/4)**
>
> ### **Re: Why is LudoBench valuable?**
>
> Unlike existing benchmarks which explore game reasoning in ‘lab environments’, we are the first to explore an ‘in the wild’ benchmark for a wildly popular and underexplored use case. As reviewer xQL531 mentioned, LudoBench presents a straightforwardly valuable dataset with direct applications; **millions play tabletop games every year**, and *the challenges LudoBench presents are exactly those needed for enabling LMs to engage in these analog, situated environments.*
>
> LudoBench mirrors a common use case: a player snaps a photo of their game state and asks for a rules clarification or advice on what to do next. These steps (rules lookup, simple strategizing) are often burdensome for humans (especially self-taught newcomers), so LM-assisted guidance is genuinely useful. **Our work bridges the gap from conventional academic analysis of game reasoning, and instead focuses on assessing competence for the messy, real-world problems that humans face.**

---

### Official Review · Reviewer_xQL5 · 2025-10-31

**Soundness:** 4
**Presentation:** 3
**Contribution:** 3
**Rating:** 6
**Confidence:** 4

**Summary:**

The authors introduce a benchmark, LudoBench, to evaluate the abilities of multimodal large language models on three categories of tasks—perceiving board game states, integrating rulebook knowledge to evaluate game states, and solving short-horizon optimization tasks (e.g. "which legal move would get the most points?"). The dataset consists of 400 carefully annotated examples across three board games in divers settings. The benchmark is far from saturated, with models scoring less than 70% on the easiest tier, and less than 10% on the hardest, while humans achieve scores in at least the 80s on all tiers. The authors detail their question-generation pipeline, and analyze interesting patterns across model performance (e.g., how models tend to over-update on images and therefore performance sometimes increases for text-only rulebooks).

**Strengths:**

This is a straightforwardly valuable dataset to the community; it bridges the gap between games-as-nice-theoretically-perfect-environments and "can models to real-world tasks" by looking at "understanding games in the real world". I specifically like the fact that all the questions are, on one hand, open-ended, but on the other hand have specific, well-defined, and unique answers. I especially appreciate the authors' efforts to check every item in the dataset for correctness and resolve ambiguities; this plagues most ML eval sets, and seems to be thoroughly avoided here! The detailed analyses of failure modes and ablations are well-executed.

**Weaknesses:**

I think 400 data points is on the smaller side for a dataset, though this is mitigated by the fact that the points are all high-quality. Also, having only three board games is an obvious limitation. Probably the most major current weakness is the fact that the models evaluated are all decently out-of-date; if accepted, I would encourage the authors to update the main tables with the most recent models available.

**Questions:**

I wonder to what extent the generation of these scenarios is automatable; for instance, if there are online implementations of these games that track board state, it might be easy, at the very least, to get a lot of tier 1 input-output pairs?

---

> ### Author Response · Authors · 2025-12-01
> **Response to Reviewer xQL5 (1/2)**
>
> Thanks for your review!
>
> To briefly recap our response + corresponding paper updates:
> - We have expanded the size of LudoBench to address concerns on size and breadth.
>     - We now cover five games (adding the popular games Catan and Carcassonne)
>     - Each reasoning tier now has 200+ examples (638 examples total)
> - We have added results on the newest frontier models (GPT-5.1, Gemini 3 Pro, Claude Sonnet 4.5) and discuss our observations.
> - We have added a link to an anonymized dataset visualizer if you’re curious to see more examples (https://anonymous-researcher-2026.github.io/anon-submission-2026/).
> - We share our thoughts on scenario automation and future directions for this work.

---

> ### Author Response · Authors · 2025-12-01
> **Response to Reviewer xQL5 (2/2)**
>
> First, we appreciate your pointing out the real-world significance of a dataset like this; tabletop board games are played by millions across the globe, and we feel they offer a vibrant and compelling medium in which to assess realistic complex reasoning.
> Even as recently as last week, the new Gemini 3 reveal advertised the reasoning capacity to parse a game image (chessboard) and create a playable, interactive game from it (https://x.com/GoogleAI/status/1990873697658314753). Anecdotally, we were unable to effectively reproduce this demoed functionality on game states from our benchmark, however, it’s clear that these types of tasks are on the near horizon, and motivates the need for real-world testbeds.
>
> ---
>
> ### **Re: Benchmark Size**
>
> We appreciate your feedback on dataset size. As you hinted at, we intentionally have kept the benchmark scope tightly focused; as no ‘in the wild’ benchmarks such as ours exist for this setting, our emphasis was first and foremost on quality and human validation. We have, however, conducted further annotation to expand both the quantity and breadth of the dataset.
>
> - We have annotated examples for the popular tabletop games Catan and Carcassonne (the two most-favorited games on the site BoardGameGeek.com).
> - We now have 600+ examples from five games that reflect a broader range of complexities and coverage of popular board games.
> - Each reasoning tier now contains 200+ examples.
>
> ---
>
> ### **Re: Model evaluations are out of date.**
>
> We are happy to update the draft with the latest models before publishing the version of this paper. To this end, we have explored **GPT-5.1**,  **Gemini Pro 3**, and **Claude 4.5 Sonnet**, which were released this month. The results are quite exciting, and we have updated the paper draft with these numbers. While still nowhere near close to human performance on Tier 2 (Rules Integration) and Tier 3 (Short-horizon Optimization) reasoning tasks, we observe significant performance improvements, particularly for Gemini Pro 3.
>
> Below, we provide the overall results of human performance compared with recent thinking models.
>
> | Model                      | Tier 1 | Tier 2 | Tier 3 |
> |----------------------------|--------|--------|--------|
> | Expert                     | 98.3%  | 97.5%  | 87.2%  |
> | Hobbyist                   | 96.7%  | 89.1%  | 82.2%  |
> | Gemini Pro 3 (newly added) | 75.8%  | 51.3%  | 12.6%  |
> | GPT-5.1 (newly added)      | 65.7%  | 34.8%  | 10.7%  |
> | o3                         | 62.5%  | 38.2%  | 11.7%  |
> | Claude 4.5 Sonnet (newly added) |  57.2% |  35.3% |  4.5%  |
>
> **Some highlights:**
> - Perception performance is significantly improved over other SoTA methods. Gemini Pro 3 sets the new perception SoTA at 75.8%.
> - Rules integration (Tier 2) is significantly improved, reaching 51.3% in Gemini Pro 3.
> - T3 reasoning is still massively challenging, however, we see trends that indicate frontier models have the capacity to utilize long multimodal rules information.
> - - [Pax Renaissance] In the (updated) Figure 4 of the paper, we see that while recent strong models struggle with only intrinsic knowledge (‘None’ modality), they are the first to effectively utilize the multi-modal, 44-page rulebooks of this game, netting significant performance improvements. For Gemini Pro 3, starting from 5% with no provided rules, text roughly doubles the accuracy (11%), and image modality pushes it to almost 4x higher (19%). GPT 5.1 and o3 similarly achieve 21% performance on this split. In contrast, the best ‘non-thinking’ models achieve only 9%.
>
> ---
>
> ### **Re: Is it possible to automate scenario generation from online implementations?**
>
> This is a great question. There are some mainstream digital platforms for playing tabletop games – including BoardGameArena, Tabletop Simulator, Tabletopia – however, there are no open-source online implementations that display a configurable game environment combined with board game state tracking.  Existing platforms either:
>
> 1. Have proprietary, rules-enforced game implementations and expose no functionality to programmatically configure game states, or
> 2. Are simple 3D sandbox environments which do not encode any game logic or game knowledge – they simply provide 3D objects and assets from the game (replicating the analog nature of a tabletop environment).
>
> The direction of scenario creation is an important area for future work. We note that A) it is very costly for tabletop game studios and designers to create digital implementations of their games (https://stonemaiergames.com/the-truth-about-digital-board-games), B) there is a world in which sufficiently-strong LMs could automatically create programmatic game implementations simply from a rulebook (https://www.arxiv.org/pdf/2508.16447), C) High-quality game reasoning examples are crucial for validating and improving LMs on this task. Our work focuses on C, to enable A and B in the future.

---

### Author Response · Authors · 2025-12-02
**Summary for AC**

We thank the reviewers for their feedback! This summary outlines LudoBench’s contributions and how we addressed the reviewer concerns.

### 1) **Benchmark Overview**

Our work introduces **LudoBench**, a benchmark for *situated* multimodal game comprehension in real tabletop settings.

LudoBench tasks models with:
1. Interpreting a situated game state from image(s) of a tabletop.
2. Ingesting the corresponding game rulebook (None / Text / Image variants).
3. Answering game reasoning questions on action legality, scoring, and short-horizon strategy.

Examples are organized into three cumulative tiers:
- **Tier 1 – Environment Perception:** rule-agnostic scene understanding (counts, icons, orientations).
- **Tier 2 – Rules Integration:** retrieving and grounding specific rules in the pictured state.
- **Tier 3 – Short-Horizon Optimization:** solving tactical puzzles that require correct rules application.

### 2) **Benchmark Novelty**

LudoBench is, to our knowledge, the first benchmark to evaluate situated multimodal game reasoning in tabletop settings. Unlike prior work focused on long-term mastery, text-only games, or toy rulesets, we study *mainstream tabletop games in the wild*, matching how humans play and ask questions. LudoBench features:
- **Realistic visual game states:** off-axis images of real 3D boards requiring human-like scene parsing.
- **Oracle-free, offline setup:** models are provided no rules-enforced game engine -- only images + rulebook.
- **Tiered diagnostics:** we assess perception, rules grounding, and planning.
- **Expert-crafted, human-validated QA:** each example has a single gold answer, confirmed via expert–hobbyist review.

### 3) **Key Findings**

Across human baselines and nine frontier models, we see:

- **Large human–LM performance gaps:**
  - Experts/hobbyists score ~90–98% on Tiers 1–2 and >80% on Tier 3.
  - The best thinking model, **Gemini Pro 3**, reaches 76/52/13% on T1/T2/T3, i.e., **70 points below hobbyists on Tier 3 (~82%)**.
- **Human-level scene parsing has not been achieved:** strong models still make basic perception errors (miscounts, missed pieces, orientation mistakes).
- **Rules integration is a major bottleneck:** Text/Image rulebooks reliably improve T2/T3 over None, but accuracy remains far from human.
- **Short-horizon optimization is extremely challenging:** the best models achieve <20% accuracy on game “puzzle” problems.
- **Dense visual rulebooks help only strong models:** some thinking models benefit from screenshots; for non-thinking models, text rulebooks generally outperform images.

### 4) **Summary of Responses to Reviewer Feedback**

| Reviewer Feedback Questions/Topics                      | Summary of Response |
|-------------------------------------------|-----------------------------|
| **Evaluation results should include newer models.** (xQL5)          | We added results for *GPT-5.1, Gemini Pro 3, and Claude Sonnet 4.5*. |
| **Benchmark is high-quality, but size and breadth could be expanded.** (xQL5, k6xk, cEf5) | We expanded from 400 to *638 QA examples* and from 3 to *5 games* (adding the popular Catan and Carcassonne), now with **200+ examples per tier**, covering diverse genres and mechanisms with high-quality, expert–curated examples. |
| **Provide access to the dataset.** (k6xk)                 | We provide a live, anonymized [dataset visualizer](https://anonymous-researcher-2026.github.io/anon-submission-2026/) for browsing games/tiers and inspecting images and gold answers. |
| **Any concerns with pretraining exposure to games/rulebooks?** (cEf5) | In our use case, models *should* use both intrinsic and extrinsic knowledge. Empirically, pretraining alone is insufficient: for longer games and harder tiers, rulebooks (Text/Image) improve performance but remain well below human accuracy. |
| **What is an approximate performance upper bound through repeated/varied prompting?** (k6xk)     | We add Top-k accuracy analysis (and also cross-modality oracle analysis). Top-5 improves T1/T2, but *Tier-3 accuracy stays <25%*, indicating fundamental capability gaps that further sampling cannot solve. |
| **Why LudoBench, and does it reflect general-purpose reasoning?** (k6xk, cEf5) | LudoBench is the first *in-the-wild* benchmark for a widely used yet underexplored use case. Millions play tabletop games, and LudoBench’s multi-faceted reasoning challenges match the skills needed for LMs to assist in analog, situated environments. |

### 5) **Conclusion for AC**

LudoBench introduces a high-quality, situated reasoning benchmark that bridges idealized engine-based game evaluations and the real-world challenge of understanding physical tabletop games. Our extensive evaluation across frontier models, together with modality ablations and tiered failure analyses, uncovers a large and persistent human–LM performance gap and provides a clear, diagnostic testbed for advancing grounded multimodal reasoning in realistic settings.

---

### Meta-Review · Area_Chair_M7NU · 2026-01-07

**Summary:**

This paper introduces LudoBench, a multimodal benchmark evaluating vision-enabled LLMs’ ability to comprehend real-world tabletop strategy games via three cumulative tiers: Environment Perception (Tier 1), Heterogeneous Rules Integration (Tier 2), and Short-horizon Optimization (Tier 3). The benchmark initially included 400 QA examples across 3 games, later expanded to 638 examples across 5 games (Kingdomino, Res Arcana, Pax Renaissance 2e, Catan, Carcassonne), with human hobbyists achieving >80% accuracy on all tiers. Evaluations of 12 frontier models (including GPT-5.1, Gemini Pro 3) reveal stark performance gaps: top models reach ~76% on Tier 1, ~51% on Tier 2, and <13% on Tier 3. Key findings highlight failures in cross-modal grounding, spatial reasoning, and rule application—core limitations for real-world situated reasoning.
Reviewers recognized the benchmark’s novelty, real-world relevance, and rigorous annotation pipeline. Core concerns centered on dataset scale, game diversity, model currency, and generalizability. The rebuttal effectively addressed most issues via dataset expansion, new model evaluations, and additional analyses, strengthening the work’s impact. Overall, the paper's strengths overwhelms its weaknesses and it is recommended for acceptance.

**Reviewer Concerns:**

Key Concerns Raised:
- Dataset scale and game diversity (xQL5, k6xk, cEf5): Initial 400 examples/3 games were too limited; narrow genre coverage constrained generalizability.
- Model currency (xQL5): Evaluated models were outdated; newer models (e.g., GPT-5.1, Gemini Pro 3) needed to be tested.
- Benchmark access and transparency (k6xk): Broken link to dataset; lack of visibility into examples.
- Pretraining exposure (cEf5): Risk of models memorizing public rulebooks, inflating text-modality performance.
- Prompt sensitivity and upper-bound performance (k6xk): Fixed prompt structure; no analysis of Top-k accuracy or advanced prompting (e.g., CoT).
- Generalizability beyond tabletop games (cEf5): Domain-specific tasks may not reflect broader reasoning capabilities.
- Perception-reasoning conflation (cEf5): No dedicated pretest to isolate visual comprehension from game reasoning.

Addressed in Rebuttal:
- Expanded benchmark to 638 examples across 5 games (adding Catan, Carcassonne) with broader genre/mechanism coverage (30+ game mechanics, 11+ genres).
- Added evaluations of latest models (GPT-5.1, Gemini Pro 3, Claude 4.5 Sonnet) and updated results.
- Provided an anonymized dataset visualizer for example browsing and transparency.
- Demonstrated that pretraining alone is insufficient: rulebooks (text/image) consistently improve Tier 2/T3 performance, indicating genuine reasoning rather than memorization.
- Conducted Top-5 and cross-modality oracle analyses; found Top-5 accuracy remains <25% on Tier 3, confirming fundamental capability gaps not solvable by prompting.
- Argued LUDOBENCH probes transferable real-world competencies (scene parsing, multimodal retrieval, planning) and highlights intrinsic relevance (millions play tabletop games).
- Clarified Tier 1 inherently serves as a perception pretest (rule-agnostic visual tasks like counting, icon recognition).

Outstanding/Partially Addressed:
- Generalizability to non-tabletop domains (cEf5): No evidence of performance correlation with broader reasoning tasks; remains domain-specific.
- Novice human baseline (cEf5): Human evaluations focus on hobbyists/expert—no data on first-time players (the target use case of "first-time gameplay comprehension").
- Inversion in Tier 1 performance (cEf5): Occasional cases where rulebook modalities underperform "None" (attributed to distraction) lack deeper analysis.

**Reviewer Scores:**

xQL5: 6 → 6 or 8

k6xk: 4 → 4 or 6

cEf5: 4 → 4 or 6

---

### Decision · Program_Chairs · 2026-01-26

Accept (Poster)